# A long-term study of cloud residuals from low-level Arctic clouds

Linn Karlsson[1,2], Radovan Krejci[1,2], Makoto Koike[3], Kerstin Ebell[4], and Paul Zieger[1,2]

[1]Department of Environmental Science, Stockholm University, Stockholm, Sweden
[2]Bolin Centre for Climate Research, Stockholm University, Stockholm, Sweden
[3]Department of Earth and Planetary Science, University of Tokyo, Tokyo, Japan
[4]Institute for Geophysics and Meteorology, University of Cologne, Cologne, Germany

**Correspondence:** Paul Zieger (paul.zieger@aces.su.se)

**Abstract.** To constrain uncertainties in radiative forcings associated with aerosol–cloud interactions, improved understanding of Arctic cloud formation is required, yet long-term measurements of the relevant cloud and aerosol properties remain sparse. We present the first long-term study of cloud residuals, i.e. particles that were involved in cloud formation and cloud processes, in Arctic low-level clouds measured at Zeppelin Observatory, Svalbard. To continuously sample cloud droplets and ice crystals and separate them from non-activated aerosol, a ground-based counter-flow virtual impactor inlet system (GCVI) was used. A detailed evaluation of the GCVI measurements, using concurrent cloud particle size distributions, meteorological parameters, and aerosol measurements, is presented for both warm and cold clouds and the potential contribution of sampling artefacts is discussed in detail. We find an excellent agreement of the GCVI sampling efficiency of liquid clouds using two independent approaches. The two-year data set of cloud residual size distributions and number concentrations reveals that the cloud residuals follow the typical seasonal cycle of Arctic aerosol, with a maximum concentration in spring and summer and a minimum concentration in the late autumn and winter months. We observed average activation diameters in the range 58–78 nm for updraft velocities below $1\,\mathrm{ms^{-1}}$. A cluster analysis also revealed cloud residual size distributions that were dominated by Aitken mode particles down to around 20–30 nm. During the winter months, some of these small particles may be the result of ice, snow or ice crystal shattering artefacts in the GCVI inlet; however, cloud residuals down to 20 nm in size were also observed during conditions when artefacts are less likely.

## 1   Introduction

Aerosols and clouds are important for climate, yet they remain one of the largest sources of uncertainty in climate projections (Boucher et al., 2013). Many of the parameters that govern cloud and aerosol formation are subject to change as the climate changes as well, which further obscures the picture. The Arctic is a region of particular interest, because it is warming more rapidly than the rest of the globe (Serreze and Francis, 2006; Serreze and Barry, 2011). In terms of aerosol particles, the Arctic is characterised by a distinct seasonal cycle with low natural background number concentrations for parts of the year (Willis et al., 2018). The low background concentration is especially true for late autumn and early winter when the absence of sunlight and direct particle sources inhibits natural emissions and the formation of new particles (Tunved et al., 2013). This means that small changes in Arctic aerosol particle concentrations, for example following sea ice loss and increased

natural marine emissions (Struthers et al., 2011) or altered transport and/or emissions of anthropogenic particles (Law and Stohl, 2007), can potentially cause large changes in cloud properties (Mauritsen et al., 2011). Crucially, the autumn and winter seasons are also when Arctic amplification is most pronounced (Serreze and Barry, 2011; Maturilli and Kayser, 2017). This, in combination with the low background particle concentrations, makes the Arctic autumn and winter seasons more likely to experience large relative changes in aerosol particle concentrations and, consequently, changes in cloud properties. Due to the sparsity of observations, we know less about cloud and aerosol processes in the Arctic than elsewhere. New long-term observations are thus essential for closing existing knowledge gaps.

Long-term observations of Arctic aerosol particles generally come from a relatively small number of permanent measurement stations. While there are differences in aerosol properties between the sites, it has been shown that they all share common features both in terms of particle number concentration and particle number size distribution (Freud et al., 2017). This characteristic seasonal cycle of Arctic aerosol properties has been demonstrated previously for individual sites (e.g. Ström et al., 2003; Tunved et al., 2013; Nguyen et al., 2016). During the transition from winter to springtime, the number concentration of accumulation mode particles (diameter $\gtrsim 60\,\mathrm{nm}$) increases due to long-range transport of polluted air masses – a phenomenon known as Arctic haze (Mitchell, 1956). In summer, changes in circulation and cloud cover lead to efficient nucleation scavenging of these particles, subsequently lowering their concentration (Tunved et al., 2013). Lower accumulation mode particle concentrations, together with increased biological activity and photochemistry, helps facilitate new particle formation leading to number size distributions dominated by the smaller, Aitken mode particles (diameter $\lesssim 60\,\mathrm{nm}$) in the Arctic summertime (Ström et al., 2003). During autumn, new particle formation in the Arctic is less common and aerosol removal processes (i.e. precipitation) are stronger compared to aerosol sources, resulting in gradually decreasing aerosol number concentrations across the particle size spectrum (Tunved et al., 2013).

Studies characterising Arctic cloud condensation nuclei (CCN) generally cover short time periods, and only a couple of studies exist that look at the seasonal cycle in the Arctic (Jung et al., 2018; Dall'Osto et al., 2017; Schmale et al., 2018). Jung et al. (2018) measured CCN on Svalbard and found that the seasonal variation in CCN concentrations correlated well with the variation in accumulation mode aerosol particle concentrations. They also identified new particle formation and subsequent particle growth as contributors to summertime CCN concentrations, in line with results from a previous long-term study (Dall'Osto et al., 2017) as well as shorter airborne and ground-based measurement campaigns (Leaitch et al., 2016; Zábori et al., 2015). CCN number concentrations in the Arctic have been found to range between a few tens and a couple of hundred particles cm$^{-3}$ (Jung et al., 2018), although concentrations vary spatially. Local concentrations of less than 1 and more than 1000 cm$^{-3}$ have been reported (Mauritsen et al., 2011; Moore et al., 2011). CCN are of course only part of the picture – in cold and mixed-phase clouds, ice nucleating particles (INP) are also important. INP are much rarer, with concentrations several orders of magnitude lower than typical CCN concentrations. In the Arctic, INP concentrations have been found to range between approximately $10^{-5}$ and $10^{-1}$ L$^{-1}$ (e.g., Wex et al., 2019; Tobo et al., 2019; Irish et al., 2019).

An important caveat is that all of the aforementioned studies measure CCN and INP concentrations by artificially activating aerosol particles. It is, however, possible to study CCN and INP properties directly inside clouds, by measuring the so-called cloud residuals that remain when cloud droplets and ice crystals (collectively termed cloud particles) are dried. This can be

achieved with a counterflow virtual impactor (CVI) inlet (Ogren et al., 1985; Noone et al., 1988), which separates cloud particles from unactivated aerosol particles on an inertial basis. Because the CVI can measure both water and ice particles, cloud residuals may correspond to either CCN, INP or results from in-cloud processes (e.g. impaction scavenging or secondary ice; Field et al., 2016). In the Arctic, CVI inlets have previously only been deployed during short, dedicated aircraft campaigns (McFarquhar et al., 2011; Wendisch et al., 2019) and, until now, no long-term observations of cloud residual properties have been carried out either in the Arctic or globally. Aircraft measurements using CVI inlets have the advantage of recording profiles of undisturbed and elevated clouds but are very expensive and limited in time, while ground-based CVI observations can cover longer time periods (seasons to years) but are potentially affected by the surrounding orography. Here, we present a unique 2-year data set of size-resolved cloud residual and total particle measurements recorded on Svalbard using a ground-based CVI (GCVI) inlet. These are the first continuous measurements of cloud residuals in the Arctic that cover the full annual cycle. Since this is the first long-term deployment of a GCVI inlet, globally and in the Arctic, emphasis will be put on the evaluation of the GCVI inlet sampling efficiency and a detailed discussion of the potential contribution of artefacts during mixed-phase cloud conditions. Our observations are accompanied by measurements of total aerosol particles (interstitial and activated aerosol particles) and cloud particle size distributions, meteorological parameters as well as remote sensing data, which, taken together, provide valuable new information about the elusive Arctic cloud nuclei.

## 2 Methods

We present total particle and cloud residual size distributions and number concentrations measured from 26 November 2015 to 4 February 2018 at Zeppelin Observatory using two different inlet systems. These measurements are complemented by measurements of cloud particle size distributions, temperature, wind parameters and remote sensing data which are described below. A schematic illustration of the experimental set-up and a photo of the inlet systems at Zeppelin Observatory are shown in Fig. 1. Tables 1 and S1 (in the supplementary material) give further details on the instrumentation and data coverage.

### 2.1 Site description

Zeppelin Observatory (78°54'N 11°53'E) is located on Svalbard in the high Arctic, approximately 2 km south of the research village Ny-Ålesund. Situated 480 m above sea level (inlet height) on the ridge of Mt. Zeppelin, the station is largely unaffected by local pollution sources and often in cloud ($\sim$16 % of the time in 2015–2018, where *in cloud* is defined in this work as visibility < 1 km for at least 5 min as measured by the visibility sensor, see below), making it well-suited for the study of Arctic aerosol particles and clouds. Note that the observed cloud occurrence may not exactly equal the annual mean cloud occurrence at the station, as we have a slightly uneven data coverage for the different months (cf. right panel in Fig. 7 as shown later in Sect. 3.2.3).

Two predominant wind directions are characteristic for the site: south and north-north-west with median horizontal and vertical wind speeds of 3.0 ms$^{-1}$ and 0.7 ms$^{-1}$, respectively, during periods of cloud occurrence (see Fig. S1 in supplementary material). The annual cycle of aerosol size distribution parameters is quite predictable for the site (Tunved et al., 2013), however,

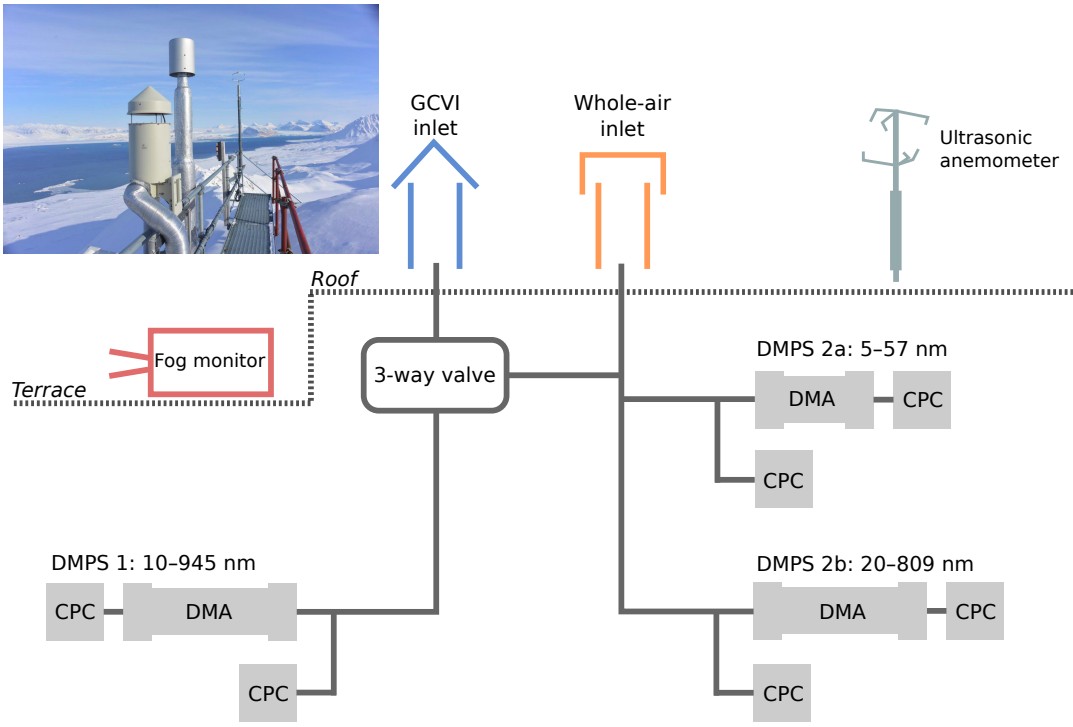

**Figure 1. Schematic illustration of the experimental set-up at Zeppelin Observatory.** The diagram shows how the whole-air inlet (orange) and the ground-based counterflow virtual impactor (GCVI) inlet (blue) are connected to the differential mobility analysers (DMAs) and condensation particle counters (CPCs). The 3-way valve switches the sample flow to the instruments on the left-hand side from the GCVI inlet to the whole-air inlet when there is no cloud to be sampled. Cloud sampling is activated if the visibility drops below 1 km (measured by a visibility sensor (not pictured) next to the GCVI inlet). Auxiliary measurements from a fog monitor and an ultrasonic anemometer have also been included in the data analysis.

the site can not be regarded as being representative for the entire Arctic. Freud et al. (2017) have shown that although certain similarities in aerosol size and concentration exist between the different permanent measurement sites in the Arctic, e.g. caused by similarities in transport patterns, particle formation or removal mechanisms, distinct differences were attributed to the proximity of aerosol sources, local meteorological effects or the influence of open ocean, land areas and sea ice.

In terms of cloud cover and cloud type, it is difficult to say how representative the measurements at Ny-Ålesund and Zeppelin Observatory are for the broader Arctic. Shupe et al. (2011) have analysed the occurrence and macro-physical properties of Arctic clouds at six observatories, including Ny-Ålesund, and found, for example, that clouds are more persistent at the far western Arctic sites. More detailed analyses of cloud radar observations from Ny-Ålesund (Nomokonova et al., 2019b; Ebell et al., 2020; Nomokonova et al., 2019a; Gierens et al., 2020) partly confirmed results of previous studies, e.g. high cloud occurrence at Ny-Ålesund in summer and autumn, but also revealed differences. For example, Nomokonova et al. (2019b) revealed a higher annual cloud occurrence at Ny-Ålesund (∼81%) than Shupe et al. (2011, ∼61%). Differences in the observed

cloud statistics are likely also due to different observing instruments and methods as well as different time periods analysed. Satellite data show that Ny-Ålesund is located in a region with highest cloud cover in the Arctic (Cesana et al.; Mioche et al., 2015). The previously reported cloud occurrences are much higher than what we observe, because we are observing at a fixed altitude and thus only measure low-level clouds. The representativeness of our observational conditions for the broader Arctic cloud cover (cf. Liu et al., 2012) is unclear.

Dahlke and Maturilli (2017) showed that the synoptic flow towards Ny-Ålesund represents typical Arctic climate during the summer months, while during the winter periods large scale advection from lower latitudes is dominating in recent decades, resulting in a more maritime climate. This transition will most likely affect also cloud properties with Ny-Ålesund probably becoming less representative of the sea-ice dominated Arctic.

## 2.2 Inlet systems

### 2.2.1 Whole-air inlet

The standard aerosol inlet is heated to a temperature of around 5–10°C to prevent freezing and fulfils the World Meteorological Organization (WMO)/Global Atmosphere Watch programme guidelines for aerosol sampling of whole-air (Kazadzis, 2016). The inlet follows the guidelines for whole-air inlets for extreme environments given by the World Calibration Centre for Aerosol Physics (WCCAP) at the Leibniz Institute for Tropospheric Research, Germany (https://www.wmo-gaw-wcc-aerosol-physics.org/recommen-dations.html). It has similar characteristics as the inlet described by Weingartner et al. (1999) which can sample cloud droplets up to $40\,\mu m$ at wind speeds up to $20\,\mathrm{m\,s^{-1}}$. It is placed on the roof of the station, and particle-laden air is brought into the lab where an isokinetic flow splitter directs the air to the different sampling instruments through quarter inch stainless steel tubing. The air is not actively dried, but the temperature difference between the outside and the inside of the lab causes a reduction in the relative humidity. During our sampling period, the relative humidity of the sample flow was always below 40 % (mean±std for our period: 13±7 %).

### 2.2.2 Ground-based counterflow virtual impactor inlet

For sampling of cloud residuals, we utilise a ground-based counterflow virtual impactor (GCVI; Brechtel Manufacturing Inc., USA, Model 1205) inlet, which is based on the working principles described in Noone et al. (1988). The inlet uses opposing air flows to separate out particles with low inertia (i.e. interstitial particles), so that only cloud particles (i.e. droplets and ice crystals) are sampled. A detailed technical description of the GCVI can be found in Shingler et al. (2012). Here, we outline the basic principles only.

The GCVI inlet at Zeppelin Observatory is mounted vertically on the north side of the station roof. During operation, cloudy air is accelerated onto the tip of the inlet with the help of a wind tunnel with typical airspeeds of around $120\,\mathrm{m\,s^{-1}}$ (monitored with a pitot tube). When the ambient air meets the dry counterflow within the GCVI, two stagnation planes are generated where only particles with sufficient inertia (i.e. cloud droplets or ice crystals) can pass through and enter the sample flow. The sample flow rate is set to $15\,\mathrm{L\,min^{-1}}$ by automatic mass flow controllers that take into account the actual sample flow of each

connected instrument. It should be noted that other instrumentation besides the ones used in this study, and listed in Table 1, were operated during the years 2015 and 2018 behind the GCVI. That is why we used the overall high sample flow. The lower cut-size (diameter at which 50% of particles are sampled) in the inlet is calculated by the instrument software and is determined by the different flow velocities and the distance between the stagnation planes. Shingler et al. (2012) compared experimentally determined cut-sizes to those predicted by the software and found good agreement. The cut-size was generally between 6 and 7 µm aerodynamic diameter during our sampling period. As the cloud particles travel through the inlet, they are dried until only the cloud residuals remain. The dew point of the dry counterflow produced by the dry air generator was -40°C. Cloud particles larger than approximately 40 µm in diameter are impacted in a particle trap inside the inlet due to their long evaporation times (Shingler et al., 2012).

The GCVI is only operated when there is a cloud at the station. The system is automated and uses a visibility sensor to determine whether or not a cloud is present. The GCVI is turned on when the visibility drops below 1 km, which is the WMO's definition of fog (see WMO, 2008, and https://cloudatlas.wmo.int/). This threshold corresponds to a large liquid water content (LWC) range of $0.0004\,\mathrm{gm^{-3}}$ to $0.10\,\mathrm{gm^{-3}}$ (5th and 95th percentile) with a median value at $0.01\,\mathrm{gm^{-3}}$ as measured by the FM-120 (see below) for liquid clouds (temperatures above $0°C$). Visibility is the only criterion used, so there is no discrimination between precipitating and non-precipitating clouds. When the visibility is above 1 km, instruments that normally sample behind the GCVI inlet instead receive their sample flow from the whole-air inlet. This is achieved with a three-way valve (installed in April 2017) between the two inlets and the instruments, and allows us to collect duplicate measurements of particle size and concentrations for quality assurance during non-cloud periods.

Particles that enter the wind tunnel are concentrated inside the tip of the CVI inlet, meaning the sampled air is effectively enriched in cloud particles relative to the ambient air. The concentrations observed behind the GCVI therefore have to be corrected by an enrichment factor (EF), which depends on the airspeed in the wind tunnel, the sample flow rate and the geometry of the inlet itself (Shingler et al., 2012). With our set-up, the EF was $11.9 \pm 1$ (median 12). It should be emphasised that even after correcting for the EF, the cloud residual concentrations measured behind the GCVI cannot be considered as absolute due to the transmission efficiency of the inlet. Because the transmission efficiency depends on the size of the cloud particles before they are dried, it cannot be corrected for. However, an estimate of the absolute cloud residual concentrations can be obtained by back-calculating from the cloud particle size distribution (as measured by a fog monitor) using the experimentally determined cloud particle size dependent transmission efficiency (Shingler et al., 2012) of the GCVI inlet. Shingler et al. (2012) measured the sampling efficiency of the CVI inlet in an aerosol/droplet wind tunnel, and it has been validated by over 30 000 in-flight droplet size distributions / cloud residual concentration intercomparisons. Computational fluid dynamics modelling and a separate GCVI characterisation project by the manufacturer in a small cloud chamber showed that the Shingler et al. (2012) sampling efficiency applies to the GCVI; agreement between the corrected droplet number concentrations above the GCVI cut-size and the cloud residual particle concentrations measured downstream of the GCVI by an MCPC were within experimental uncertainty, typically 25% (F. Brechtel, Oct. 2020, pers. comm.).

As will be shown below, we find that, on average, approximately half of the cloud particles make it into the GCVI sample flow, and cloud residual concentrations therefore have to be multiplied by correction factors derived from the observations from the fog monitor (see Sect. 3.1).

## 2.3 Instrumentation

See Fig. 1 for a schematic overview of the experimental set-up and Table 1 for a summary of the instruments used, parameters measured and their temporal and/or spatial resolutions.

**Table 1.** List of instruments, measured parameters and their temporal and/or spatial resolution.

| Instrument | Parameters | Resolution |
| --- | --- | --- |
| DMPS 1 | Aerosol particle number size distributions (diameters 10–945 nm) | 5–7 min |
| DMPS 2a–b | Aerosol particle number size distributions (diameters 5–809 nm) | 15–16 min |
| GCVI | Visibility (proxy for cloud presence) | 1 s |
| FM-120 | Cloud particle number size distributions (diameters 3–47 $\mu$m) | 10 s |
| Ultrasonic anemometer (METEK uSonic-3) | 3D wind field, virtual temperature | 1 s |
| Cloudnet | Target classification (in terms of occurrence of e.g. liquid droplets, ice crystals, drizzle, etc) | 30 s, 20 m altitude bins |

### 2.3.1 Differential mobility particle sizer

Particle number size distributions were measured with a differential mobility particle sizer (DMPS). The experimental set-up at Zeppelin Observatory has three DMPS instruments: one is behind the GCVI inlet (DMPS 1) and the other two are behind the whole-air inlet (DMPS 2a–b). DMPS 1 (sample flow 1 L min$^{-1}$, sheath-air flow 4.8 L min$^{-1}$) consists of a medium Vienna type differential mobility analyser (DMA; length 0.28 m, outer radius 0.033 m, inner radius 0.025 m) and a condensation particle counter (CPC; TSI Inc., USA, Model 3772). Another CPC (TSI Inc., USA, Model 3772) is used in parallel with the DMPS to measure the total particle number concentration. DMPS 1 is set to measure particles from 10 to approximately 945 nm in mobility diameter. A full DMPS 1 scan (small to large or large to small diameters) takes approximately 6 min to complete. For the number concentrations shown in the manuscript, we used the integrated and loss corrected particle number size distributions. However, when comparing the cloud residual number concentrations to the cloud particle concentrations, the total CPC (behind the GCVI) was used.

DMPS 2a and DMPS 2b measure different but overlapping size ranges. They are synchronised as one system (DMPS 2a–b) that runs on the same software. DMPS 2a (sample flow 1 L min$^{-1}$, sheath-air flow 9.9 L min$^{-1}$) measures at the smaller end of the particle size spectrum and has an extra small Vienna type DMA (length 0.053 m, outer radius 0.033 m, inner radius 0.025 m) to minimise diffusional losses, with a CPC (TSI Inc., USA, Model 3010) behind the DMA and a CPC (TSI Inc., USA, Model 3776) for measuring the total aerosol particle concentration. DMPS 2b (sample flow 1 L min$^{-1}$, sheath-air flow

5.2 L min$^{-1}$) measures the larger size particles and has a medium Vienna type DMA (length 0.28 m, outer radius 0.033 m, inner radius 0.025 m) with a CPC (TSI Inc., USA, Model 3772) behind the DMA and a CPC (TSI Inc., USA, Model 3010) for measuring the total aerosol particle concentration. With this set-up two total CPCs are available and the second CPC is used as back-up and quality assurance. Together, DMPS 2a–b span roughly the same size range as the DMPS 1, but the time resolution is approximately 15 min per full scan. In the overlapping size range, the size distributions from DMPS 2a and DMPS 2b were

combined by using the data from DMPS 2a in all overlapping bins except the last three. DMPS 2a data were preferred because DMPS 2a is shorter than DMPS 2b, and therefore suffers fewer losses. The last three bins, however, were not corrected for multiple charges, and therefore we used the data from DMPS 2b for those bins instead.

Figure S2 shows how the DMPS systems compare during non-cloud periods. The comparison is based on data collected from May 2017–February 2018 (after the installation of the three-way inlet valve, see above). In general, the instruments compare

well for large particle sizes while DMPS 2a–b shows consistently higher concentrations of small particles below around 30 nm in diameter. This is to be expected, since the diffusion losses are higher for DMPS 1 due to the instrument dimensions and longer sampling lines. Most of the differences originate from the lowest size bins between 10 and 15 nm, as can be seen in the scatter plots of Fig. S2 (panel c and d), where the integrated number concentrations of both DMPS 1 and DMPS 2a–b are shown. The slope of the orthogonal linear regression and the R$^2$-value (coefficient of determination) improve from 1.36 to

1.01 and 0.96 to 0.99, respectively, if particle number size distributions are integrated above 15 nm instead of 10 nm particle diameter.

### 2.3.2  Fog monitor

A fog monitor (Droplet Measurement Technologies Inc., USA, Model FM-120) was used to determine the cloud particle size and number concentration. It uses an optical method to size individual cloud particles at a flow rate of approximately

1000 L min$^{-1}$ (airspeed 12 m s$^{-1}$). The instrument is positioned facing south and measures cloud particle size distributions in the size range 3.5–46 μm optical diameter (bin midpoints). More details on the instrument at Zeppelin Observatory can be found in Koike et al. (2019). It should be noted that no loss correction has been applied to the fog monitor data because no clear signatures of particle loss were found in previous work by Koike et al. (2019), although significant sampling losses were suggested in other studies depending on, for example, the cloud particle diameter, and the wind speed and wind direction

relative to the fog monitor (Spiegel et al., 2012).

### 2.3.3  Ultrasonic anemometer

A uSonic-3 Omni (METEK GmbH) ultrasonic anemometer was used to monitor wind conditions at Zeppelin Observatory. The anemometer has 3 pairs of ultrasonic transducers arranged to form 3 paths along which the speed of sound is measured. From the difference in the travel time of sound along the 3 measuring paths, the 3D wind vector as well as the acoustic temperature

can be derived. The acoustic temperature is a close approximation of the virtual temperature, which depends on the ambient relative humidity and is generally 1–2 degrees higher than the true temperature (uSonic-3 Omni Ultrasonic anemometer user manual, Metek GmbH). In the Arctic, this temperature difference was larger. The GCVI inlet has its own temperature sensor,

but it was only working for a few months at the start of our measurement period. During the overlap period, the difference between the measured acoustic temperature and the ambient temperature measured by the GCVI temperature probe was around
$2.6°C$. Thus, we have subtracted $2.6°C$ from all temperatures measured by the ultrasonic anemometer.

### 2.3.4 Cloud remote sensing

The Cloudnet algorithm suite (Illingworth et al., 2007) has been applied to the Ny-Ålesund ground-based remote sensing observations from the French-German research station AWIPEV (Nomokonova et al., 2019b), which is located approximately 2 km north of Zeppelin Observatory. A standard product is the target classification which combines measurements from cloud
radar, ceilometer and microwave radiometer with output from a numerical weather prediction model. Each radar height bin is classified in terms of the occurrence of e.g. liquid droplets, ice particles, rain/drizzle, melting ice and a combination of those. More details on the product for Ny-Ålesund can be found in Nomokonova et al. (2019b). For comparison with the cloud residual data collected at Zeppelin Observatory, we selected Cloudnet height bins between 400 and 600 m. We only compared cases when the cloud base height at AWIPEV was between 300 m and 600 m to ensure that the classifications were likely to be
applicable also to the cloud at Zeppelin Observatory. It should be noted that this cloud base height criterion reduces the number of data points we can use, such that we only have Cloudnet data for approximately 30% of our in-cloud size distribution data.

### 2.4 GCVI and DMPS data treatment

The DMPS and GCVI data were processed in several steps. The logbooks from Zeppelin Observatory – which detail dates and times for visits, maintenance, instrumental issues and other observations – were examined, and data were removed when the
logbooks indicated that they may be affected by the activity at the station. Next, daily overview plots of all relevant parameters were made, and each daily plot was visually inspected. Outliers (e.g. sudden concentration spikes) and suspected pollution events (e.g. concentration peaks around mealtimes or flight times) were removed. Special attention was also given to data points around gaps in the time series, and if there appeared to be issues in the data leading up to the instrument failure or after reboot, the suspicious data points were removed. Finally, several numerical filters were applied to catch additional outliers that may
have been overlooked during the visual inspection. These filters looked for DMPS scans where the integrated number concentration was much higher ($>500$ cm$^{-3}$, e.g. caused by electrical sparking inside the DMA) than the concentration measured by the total CPC, data points that showed a much higher concentration than both neighbouring data points ($>1\,500$ cm$^{-3}$; kept this high so as not to accidentally cut out nucleation events), and scans where the majority of the concentration came from the highest or lowest size bin (indicating sparks in the DMA or possible pollution).
The GCVI system outputs status codes for the operation of each part. When switching on/off of the GCVI occurred during a DMPS 1 scan, that scan was removed (since it is neither in- or out of cloud, and the enrichment factor is not defined for this case). Occasionally, there were also issues with icing of the visibility sensor, which led to the GCVI turning on despite there not being a cloud at the station. These cases were found by comparing the visibility to the measured cloud residual concentration, and data points that seemed questionable (i.e. too low concentration with respect to the visibility) were further investigated. If

no cloud was detected by looking at webcam images from the station (Pedersen, 2013), or if the visibility was suspiciously constant (indication of icing of the sensor), the DMPS scan for those times were removed.

After the data screening, 1 729 hours of cloud residual number size distribution measurements remained. Different analyses were limited by the availability of concurrent data from the other instruments (DMPS 2a, DMPS 2b, the fog monitor, the ultrasonic anemometer, and the Cloudnet retrieval). Thus, slightly different subsets of the cloud residual data are used in the different figures. Table S1 shows how many hours of simultaneous measurements we have for different instrument combinations, and which figures the combinations are relevant for.

We have not applied any standard temperature and pressure normalisation or particle shape correction to the data presented here, but multiple-charge corrections have been applied to all measured size distributions. They have also been corrected for particle losses due to diffusion, impaction and sedimentation using the *Particle Loss Calculator* by von der Weiden et al. (2009), assuming a particle density of $1.5\,\mathrm{g\,cm^{-3}}$.

## 2.5 Cluster analysis

A cluster analysis was performed to identify cloud residual size distributions that were dominated by Aitken mode particles. We used $k$-means clustering, implemented in the *scikit-learn* (v. 0.20.2) Python package (Pedregosa et al., 2011), which is a method to categorise data into a pre-defined number of clusters, $k$, where members of a cluster are as similar to each other as possible while at the same time being as different to members of other clusters as possible. Each data point is assigned to the cluster with the nearest mean. We categorised cloud residual number size distributions based on their shape, so the size distributions were normalised by the integral before applying the $k$-means algorithm. We selected 5 clusters ($k = 5$) to separate out the cloud residual size distributions that were dominated by the very smallest particles. Choosing fewer clusters did not fully separate this distribution of interest, while more clusters led to a further splitting of the accumulation mode (see Fig. S3).

## 3 Results

Unless otherwise stated, all data presented in this section have been averaged to match the time resolution of the cloud residual size distributions measured by DMPS 1 (i.e. 5–7 min averaging time, cf. Tab. 1). When making simple comparisons to DMPS 2, which has a lower resolution than DMPS 1, we used all simultaneously measured data (i.e. overlapping DMPS scans, without repetition of data points). In the cases where a one-to-one data point comparison was necessary, both DMPS data sets were downsampled (usually to 30 min averages).

### 3.1 Determining the GCVI sampling efficiency

The cloud residual concentration measured downstream of the GCVI inlet cannot be considered absolute due to the transmission efficiency of the inlet. Thus, we need to know the cloud particle number size distribution to be able to derive correction factors for the cloud residual concentrations. Assuming that the cloud particle distribution measured by the FM-120 fog monitor is an accurate representation of the cloud particles that enter the GCVI inlet, we applied the experimentally determined

size-dependent transmission efficiency from Shingler et al. (2012) (linearly extrapolated to cover the full FM-120 cloud particle size range, see Fig. S4) to calculate the cloud particle concentration above the GCVI cut-size that would have made it into the sample flow. Here, it is important to note that the transmission efficiency was determined for hollow glass beads without using the inlet counterflow (Shingler et al., 2012). As such, it does not take into account potential evaporation of water from the
cloud particles in the different inlet segments. Within this work, we have only used the transmission efficiency determined for the first inlet segment, because we believe that the dry counterflow initiates evaporation which would make the transmission efficiency determined for subsequent sections an underestimation of the true transmission efficiency. This choice may result in an overestimation of the transmission efficiency (particularly of larger cloud droplets) since some losses are effectively ignored.

The cloud particle concentrations, multiplied by the GCVI sampling efficiency and integrated above the GCVI cut-size,
were compared to the cloud residual number concentrations measured behind the GCVI inlet (by the CPC), and the result can be seen in Fig. 2. Given the uncertainties involved, the instruments agree reasonably well in terms of the seasonal cycle and magnitude of cloud particle/cloud residual concentrations (Fig. 2a). A 2D histogram of the cloud particle concentrations versus cloud residual concentrations (Fig. 2b) shows that most of the data points lie on or around the 1:1 line. 65% of the data lie within a factor of 2, 88% within a factor of 5, and 92% within a factor of 10 from the 1:1 line. A linear orthogonal
distance regression (ODR) of cloud residual versus cloud particle number concentrations (Fig. 2b) returns a slope of 1.14 and a coefficient of determination, $R^2$, of 0.61. However, there is some scatter. Most notably, there is a group of data points below the 1:10 line ($\sim$ 7–8% of the data) that seems to be associated with colder temperatures at the sampling site (Fig. 2c). Due to this temperature dependent behaviour, we will discuss the correction factors for warm and cold clouds separately.

### 3.1.1 Correction factors

In Fig. 2, we corrected the cloud particle concentrations for the GCVI transmission efficiency. However, to be able to compare the cloud residual measurements to the aerosol particle measurements from the whole-air inlet, we need to apply the correction in the other direction. The integrated transmission efficiency of the GCVI inlet was estimated by comparing the integrated cloud particle number concentrations with and without taking into account the size-dependent transmission efficiency of Shingler et al. (2012) and the cut-size of the GCVI inlet. Figure 3a shows histograms of total integrated cloud particle number
concentration after correcting for the GCVI transmission efficiency (Shingler et al., 2020) and cut-size, divided by the total integrated cloud particle number concentration without corrections, for temperatures above and below 0°C (red and blue histograms, respectively). Both histograms show that, in the majority of cases, 40–50% of the total cloud particles were sampled by the GCVI. Figure 3b shows the corresponding 2D histogram (for warm and cold cases combined), together with an ODR fit which returned a best fit slope of 0.46 (note that fitting the cold and warm data separately gives the same slope).

For warm (liquid) clouds, the sampling efficiency could also be estimated by scaling the cloud residual size distribution to the total particle size distribution under the assumption that all accumulation mode particles activate into cloud droplets. We compared cloud residual and total particle concentrations (30 min mean values) integrated above different accumulation mode threshold diameters, $D_{cut}$, in the range 41–505 nm. Figure 4a shows an example scatter plot for $D_{cut} = 123$ nm, with the corresponding ODR fit parameters (grey data points correspond to cold clouds and are not included in the fit). Figure 4b–c

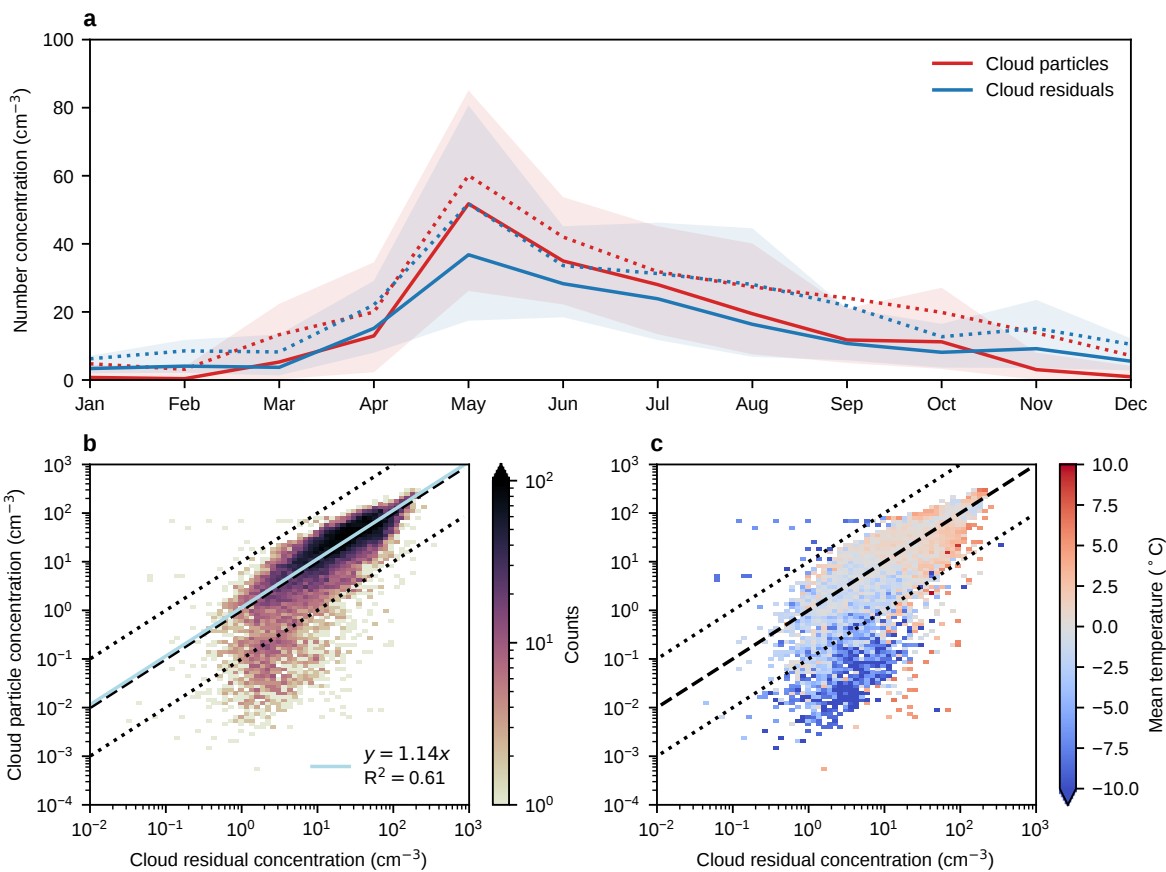

**Figure 2. Comparison of cloud residual and cloud particle number concentrations. a** Monthly averages of cloud residual number concentrations as measured behind the GCVI (blue) and corresponding cloud particle number concentrations derived from the FM-120 fog monitor measurements and the transmission efficiency of the GCVI inlet (red). Solid and dotted lines show median and mean values, respectively, and shaded areas indicate the 25[th] to 75[th] percentile ranges. **b** Density scatterplot of cloud residual versus cloud particle number concentrations, including an orthogonal distance linear regression (grey line). **c** The same as b, but colourcoded by the average temperature instead of the data point density. In b and c, the black dashed line represents the 1:1 line and the dotted lines represent 10:1 and 1:10 lines. The transmission efficiency and cut-size of the GCVI inlet (Shingler et al., 2012) has been included in the calculation of the cloud particle number concentration in all panels.

show ODR best fit slopes and coefficients of determination for all the different $D_{\text{cut}}$ diameters (results from cold cloud data are included for completeness, but not used since we then cannot assume liquid droplet activation). Above approximately 100 nm, the slope plateaus around 0.46, which is the same factor derived from Fig. 3, so these two independent methods agree remarkably well. This gives us confidence in the validity of correction factors derived based on the fog monitor data and based on the accumulation mode comparison.

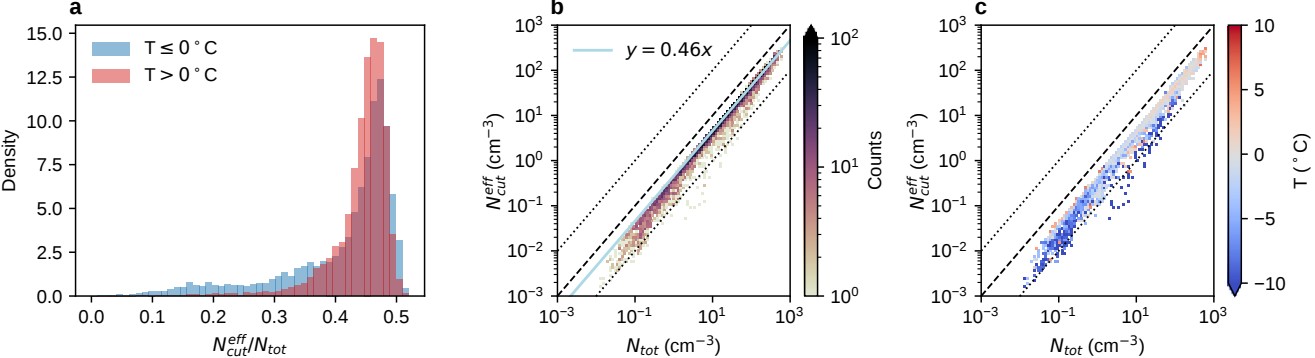

**Figure 3. Cloud particle number concentrations with and without taking the GCVI sampling efficiency and cut-size into account.**
Comparison of fog monitor total integrated cloud particle number concentration after correcting for the GCVI transmission efficiency (Shingler et al., 2020) and cut-size ($N_{cut}^{eff}$) divided by the total integrated cloud particle number concentration without correction ($N_{tot}$). **a** shows histograms of the ratio for temperatures $> 0°$C (red) and temperatures $\leq 0°$C (blue). **b** shows a density scatter plot of $N_{tot}$ versus $N_{cut}^{eff}$ for all data points (both histograms), together with an ODR best fit slope, **c** shows the same as b, but colour-coded by the average temperature instead of the data point density. In b and c, the black dashed line represents the 1:1 line and the dotted lines represent 10:1 and 1:10 lines.

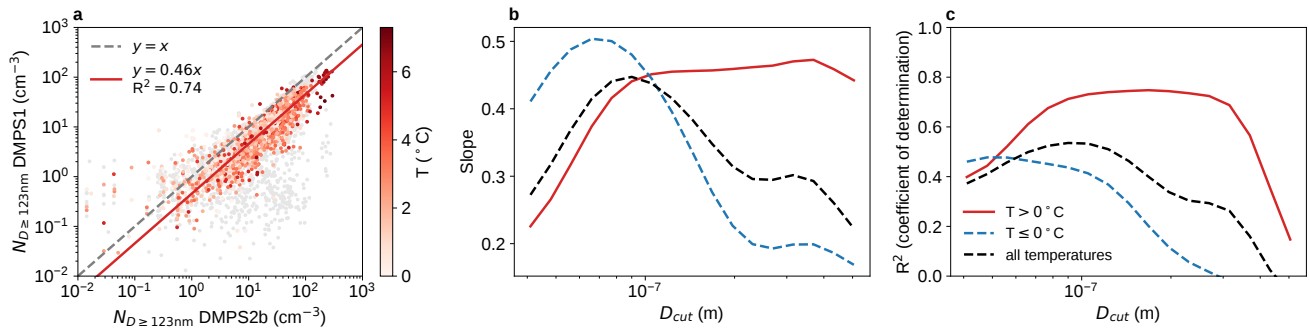

**Figure 4. Comparison of total and residual accumulation mode particle concentrations. a** scatterplot of accumulation mode (here: diameter $> 123$ nm) concentrations from DMPS 2b and DMPS 1 for temperatures $> 0°$C, together with an ODR best fit slope. Temperatures $\leq 0°$C are shown as grey points but are not included in the fit. **b** and **c** show ODR best fit slopes and corresponding coefficients of determination, respectively, for different definitions of the accumulation mode. The red lines show these parameters for temperatures $> 0°$C, and the dashed lines show the corresponding values for temperatures $\leq 0°$C (blue) and all data (black), included for completeness. 30 min mean values of DMPS 1 and DMPS 2b data were used for this analysis.

At temperatures below $0°$C, we have no independent way of verifying the correction factors. However, Fig. 3 shows that the behaviour is very similar to that at warm temperatures. The majority of the sampled cloud particles at temperatures $\leq 0°$C are likely to be supercooled droplets, which could explain the similarity in behaviour. The histogram for cold temperatures has a

slightly larger tail, related to the very coldest cases (cf. Fig. 3c). The correction factor is a function of the cloud particle number size distribution, which was relatively constant during the sampling period with the exception of the coldest temperatures

(cf. Fig. S4), so this explains why the derived correction factors would be different for the coldest cases. However, it should be noted that cold temperatures also complicate things because we could be sampling not only droplets but also ice crystals (e.g. in ice or mixed-phase clouds). Fig. 2b–c clearly showed that there are a few cases, mainly at very cold temperatures, where the assumed transmission efficiency (cf. Fig. S4) cannot reconcile the cloud particle and cloud residual concentrations measured. This will be discussed in Sect. 3.1.2 below.

Based on the analysis above, we derived an individual correction factor for each measured cloud residual size distribution using the simultaneous measurements of the fog monitor. The cloud residual data is corrected by dividing by the correction factors, $f$ (see Fig. 3a for the distribution of factors):

$$f = \frac{N_{\text{cut}}^{\text{eff}}}{N_{\text{tot}}} \tag{1}$$

where $N_{\text{tot}}$ is the total integrated cloud particle concentration and $N_{\text{cut}}^{\text{eff}}$ is the cloud particle concentration integrated above the

cut-size and with the Shingler et al. (2012) transmission efficiency applied.

These factors (Eq. 1) have been applied to all cloud residual data presented from Sect. 3.2.1 onwards assuming cloud residual size and cloud particle size are not correlated. Note that an individual correction factor using Eq. 1 assumes that the FM-120 gives a correct measure of the ambient cloud particle size distribution, which cannot always be assured (see example in Appendix A2).

### 3.1.2  Outliers at cold temperatures

Figure 2b showed that there were two groups of data points: one with good agreement between cloud residual and cloud particle concentrations, and one where the cloud residual concentrations exceeded the cloud particle concentrations by one to two orders of magnitude. The second group is associated with very low cloud particle concentrations ($1 \pm 3\,\text{cm}^{-3}$, total concentrations without cut-size and Shingler et al. (2012) correction) and the cloud particles are also fairly large in size ($11 \pm 4\,\mu\text{m}$ effective

radius). While this group is only a small minority of the data ($\sim 7$–$8\%$), its correlation with cold temperatures (Fig. 2c) means it warrants further investigation.

A discrepancy like this can have two basic causes:

a) The apparent concentration difference does not reflect the true difference but is a result of varying sampling efficiencies/issues of the FM-120 and the GCVI, or

b) The concentration difference is true, but can be a result of either real physical processes in the atmosphere or of spurious measurements caused by sampling artefacts

Determining the cause is not a trivial task. Both the GCVI and the FM-120 were calibrated using spherical standard particles, which makes the comparison especially difficult for cases when ice crystals are sampled. The true transmission efficiency of

the GCVI inlet is going to be different for non-spherical particles, i.e. ice crystals, which are not accurately represented by
glass beads. In addition, the concept of size becomes ambiguous when the sampled particles are not spherical, especially since the two instruments deal with different types of size. The optical size reported by the FM-120 is not necessarily the same as the Stokes equivalent size that determines how a crystal behaves inside the GCVI inlet, which means that the transmission efficiency we apply could be incorrect. However, the points below the 1:10 line in Fig. 2b–c still remain below the 1:1 line even if we compare the cloud residual concentration to the total, uncorrected cloud particle concentration (not shown), which
suggests that something other than errors in the assumed transmission efficiency is causing the difference.

A comparison of the measured visibility and the visibility calculated from the FM-120 data shows a reasonable agreement for the majority of data points, but again, as in Fig. 2c, there is a group of data points at predominantly cold temperatures where the agreement is much worse (Fig. S5a–b in supplementary material). The visibility was calculated using the Koschmieder formula (Seinfeld and Pandis, 2016), the measured cloud particle size distribution of the FM-120 and Mie theory (Python
package *PyMieScatt* (v. 1.7.5); Sumlin et al., 2018) by assuming spherical particles with the refractive index of water (1.33) and a wavelength of 880 nm (of the visibility sensor). We have already suggested that the assumption of spherical particles might not hold at cold temperatures, which could explain the differences in Fig. S5. However, the calculated visibility is sometimes several orders of magnitude higher than the measured one, and it is unlikely that non-sphericity would cause such large differences given that the overall sizing uncertainty from optical particle spectrometers only increases from $\pm 20\,\%$
to around $\pm 30\,\%$ for non-spherical particles (Baumgardner et al., 2017; Borrmann et al., 2000). One possibility is that the differences in visibility are caused by a loss of detected cloud particles within the FM-120 (e.g. turbulent deposition inside the contraction of the inlet) or that cloud particles (e.g. larger ice crystals or snow flakes) are larger than the last channel of the FM-120. However, in the latter case, the cloud particles are also likely to be too large to be sampled by the GCVI.

The FM-120 as well as the GCVI inlet sampling efficiency can also be affected by the wind speed and direction (Spiegel et al.,
2012), but this is not something we can easily correct for. The FM-120 faces south, i.e. into the prevailing winds at Zeppelin Observatory. The station experiences northerly winds approximately one third of the time during cloud events (see Fig. S1), and one might expect this to reduce the sampling efficiency of the FM-120. However, heatmaps similar to Fig. 2c for wind speed, updraft, and wind direction indicate no obvious correlation between wind parameters and deviations of concentrations from the 1:1 line (see Fig. S6). Interestingly, the two data groups show similar internal gradients of wind speed and updraft
velocity (Fig. S6a–b), which may suggest that the gradients are caused by the same physical process(es) but the second data group is just shifted by e.g. undercounting in the FM-120. In general, no clear influence of the prevailing wind direction in the comparison of residual and cloud particle concentration can be observed (see Fig. S7). Local effects of blowing snow could also affect the measured visibility, but there is no overall correlation between wind speed and differences in cloud particle and cloud residual concentrations (see Fig. S6). For both cases, one should also take into account that high wind speeds are only
rarely observed at Zeppelin Observatory (the median wind speed is approximately $3\,\mathrm{m\,s^{-1}}$, see Fig. S1).

The presence of precipitating particles may cause the measured visibility to be higher than the calculated one; however, precipitating particle concentrations at Zeppelin Observatory have previously been found to be mostly lower than $0.3\,\mathrm{cm^{-3}}$ (Koike et al., 2019). Hence, while the presence of such particles could explain the differences in visibility, they can only partially ex-

plain the differences in concentration in Fig. 2b unless significant ice crystal shattering (e.g. from snow flakes) occurs in the GCVI wind tunnel. The likelihood of shattering can be estimated by the non-dimensional Weber number, where fragments are expected to be produced under conditions with Weber numbers between 10 and 12 (Twohy et al., 2003). In the GCVI, these conditions occur at $100\,\mathrm{ms^{-1}}$ air speed and droplet diameters between 70 and $100\,\mu\mathrm{m}$ (F. Brechtel, pers. comm., Oct. 2020). Precipitating particles can exceed this size, and could thus produce fragments that are sampled if they are larger than the aerodynamic cut-size of the GCVI. However, it is likely that many of the fragments would not be aligned with the streamlines and therefore would not enter the GCVI.

Measurement artefacts during in-situ sampling of cloud droplets and ice crystals are a common and complex challenge (Baumgardner et al., 2017) contributing to both overestimation and underestimation of cloud residual (or cloud particle) concentration measurements (Pekour and Cziczo, 2011; Spiegel et al., 2012; Shingler et al., 2012). Particle capture by wake effects in the GCVI inlet is a possible explanation for cloud residual concentrations exceeding cloud particle concentrations; however, even at cloud particle and interstitial aerosol particle concentrations at the upper end of our observed values, only 1 % of the measured cloud residuals is estimated to be a potential artefact (Pekour and Cziczo, 2011). One would also expect to see the influence of the wake effect in the summer months but in our case the disagreement is mainly seen in winter months with generally low concentrations. Thus, this effect is likely not the major cause of the disparity between cloud residual and cloud particle concentrations that was observed.

Droplet or ice crystal shattering is another potential source of small particles. Shattering could either happen in the wind tunnel or after the stagnation plane within the CVI inlet, and this could also cause an overestimation of the cloud residual number concentration. If the particles were to shatter after the stagnation plane, this should be clearly seen as spikes in the cloud residual concentrations measured by the total CPC, and this was not observed. Regarding shattering in the wind tunnel, as stated above, the cloud particles need to exceed a certain size for this process to be likely. While this can happen when there is precipitation, it needs to be borne in mind that in most mixed-phase non-precipitating clouds, the concentration of large cloud particles is much lower than the total cloud particle concentration, and therefore particles that do shatter may need to far exceed the critical break-up diameter to produce enough fragments to significantly increase the measured cloud residual concentration (Twohy et al., 2003). Therefore, the magnitude of the concentration difference we observe suggests that precipitating particles (e.g. snow) are a more likely cause than large cloud droplets or ice crystals. If, on the other hand, we are sampling a fully glaciated cloud consisting of large ice crystals, then shattering artefacts could come from cloud ice as well as precipitating ice.

Riming or impaction scavenging of interstitial aerosol particles onto an ice crystal may be able to result in more than one cloud residual emerging from the crystal as it dries inside the GCVI inlet. If more than one residual could be released through this process without the need for crystal break-up, it could be an alternative explanation that is consistent with the difference in cloud residual and cloud particle number concentrations. However, this is speculative since no strong experimental evidence exists on how rimed particles would behave inside the CVI flow regime.

In summary, the relatively small amount of outliers discussed in this section could be a result of snow or ice crystal shattering, but there are also possible alternative physical explanations. In the following sections, we continue to segregate the cloud

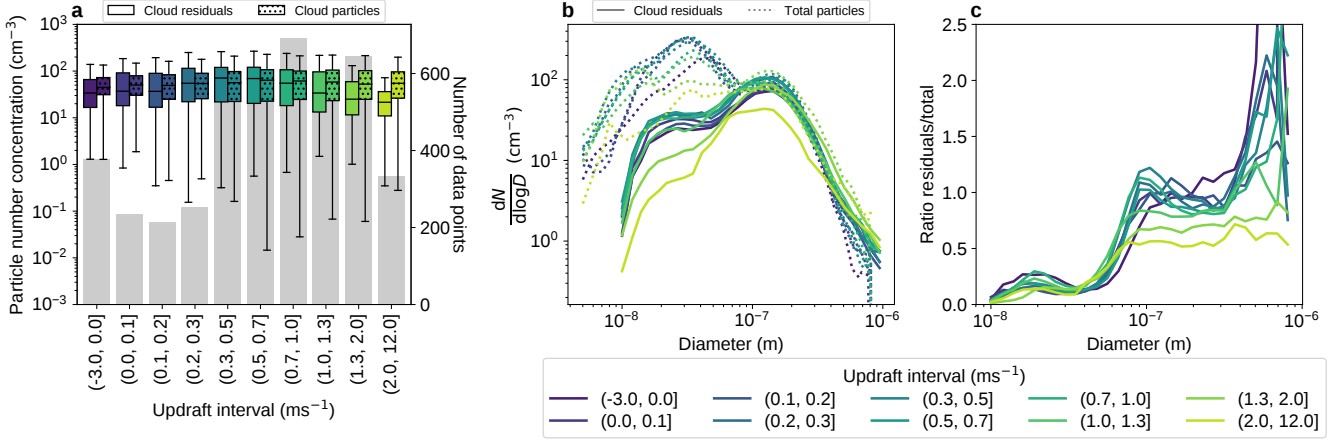

**Figure 5. In-cloud data binned by updraft. a** Box plot of cloud residual (solid) and cloud particle (hatched) number concentrations for different updraft intervals (see legend). The whiskers extend to the farthest points that are within 1.5 times the interquartile range from the nearest quartile. Points that fall outside the whiskers are not shown. The grey bars in the background indicate the number of cloud residual data points (right y-axis) per updraft bin. **b** Mean particle number size distributions of cloud residuals (solid) and total particles (dotted) for different updraft intervals (see legend). **c** Ratio of the size distributions in b, i.e. cloud residual concentrations divided by total particle concentrations. Only data collected at temperatures $> 0°C$ are shown in this figure.

residual data based on temperature and will later use cluster analysis of the shape of the cloud residual size distributions and further investigate the potential role of artefacts and ice/snow within the clouds we observed.

430 ## 3.2 Two years of cloud residual size distributions

### 3.2.1 Influence of updraft

This subsection only deals with data collected at temperatures $> 0°C$.

In liquid droplet activation, the updraft velocity is an important driver (alongside the aerosol particle size distribution and the particle composition) since it controls the supersaturation. We will therefore study how the cloud residual size distribution

435 and number concentration varies with varying updraft velocity. Because Zeppelin Observatory is a mountain site, a closer look at the updraft is also warranted to investigate potential orographic effects.

Figure 5 shows concurrent cloud particle, cloud residual and total aerosol particle data binned by updraft velocity. Panel a shows box plots of cloud residual concentrations (corrected for the GCVI transmission efficiency, see above) and cloud particle concentrations (now without corrections). The concentrations generally agree well, but there seems to be a tendency for the

440 cloud residual number concentrations to be underestimated at higher updraft, starting approximately above $1$–$1.3\,\mathrm{ms^{-1}}$. Panels b and c show a similar pattern, with cloud residual concentrations decreasing in the last two or three updraft bins. This pattern is not observed in the total aerosol particles, except for in the highest updraft bin (Fig. 5b).

Figure 5c shows the ratio between the distributions in panel b, i.e. cloud residual concentrations divided by total particle concentrations. The curves are more or less sigmoidal, like typical CCN-activated particle fraction curves. Most of the curves level out around a ratio of 1, indicating that most of the total aerosol particles larger than ∼100 nm are in fact CCN. Despite all the uncertainties and assumptions being made (e.g. GCVI sampling efficiency), it is encouraging to see that overall ratios are in the range of expected values giving further faith in our observations. The ratios are occasionally above 1 which, at the upper end of the particle size range, could be caused by small number statistics (i.e. ratios of small numbers). In the mid-size range, the ratio fluctuations could be the result of small uncertainties in sizing, concentration and losses of the two DMPS systems, causing the size modes to not be perfectly aligned.

The curves in Fig. 5c systematically level out at lower ratios with higher updrafts (for the last two or three bins), which could either mean that not all accumulation mode particles are CCN under these conditions, or that the GCVI inlet fails to sample all cloud particles at high updrafts (e.g. if the winds make it more difficult for the cloud particles to enter the wind tunnel). Taken together with the previous panels, it seems likely that the GCVI inlet sampling efficiency is negatively affected by high updraft velocities (or indeed high wind speeds in general, as these parameters tend to be correlated at Zeppelin Observatory). The sampling efficiency of the FM-120 fog monitor can in theory also be adversely affected by high wind speeds (Spiegel et al., 2012), but this seems to happen to a lesser extent than for the GCVI inlet based on Fig. 5a. One should bear in mind that the wind speeds and updrafts are generally lower near the FM-120 as it is positioned at a lower altitude than the GCVI inlet (∼5 m below).

The $D_{50\%}$, defined as the diameter where the ratio is 0.5, ranges between approximately 57 and 75 nm in Fig. 5c. One can attempt to account for the aforementioned sampling issues by normalising the plateau of the ratios to 1. If this is done, the $D_{50\%}$ ranges from 50 to 78 nm and shows a decreasing trend with increasing updraft (Fig. S8). This behaviour is expected from a cloud physics point of view, where higher updraft velocities can produce higher supersaturation levels which, in turn, allows smaller particles to activate. Updraft velocities in marine stratiform clouds are typically below $1\,\text{ms}^{-1}$ (Zheng et al., 2016) and hence higher updraft velocities could be indicative of local orographic effects and may not be representative for Arctic clouds in other areas. Excluding the bins with updraft $> 1\,\text{ms}^{-1}$ gives a $D_{50\%}$ range of 58–78 nm (Fig. S8).

### 3.2.2 Influence of temperature

Figure 6 shows concurrent cloud particle, cloud residual and total aerosol particle data, this time binned by ambient temperature instead of updraft velocity. The box plots in the first panel show that the cloud residual and cloud particle number concentrations agree well down to about -4 to -6°C, where the cloud particle concentrations drop below the cloud residual concentrations. These bins contain relatively few data points (bar plot in Fig. 6a), but they more or less follow the general trend of decreasing cloud particle/cloud residual concentrations with decreasing temperature.

Figure 6b shows the mean cloud residual and total aerosol particle size distributions for the same temperature bins. Both the Aitken and the accumulation mode are present in both cloud residuals and total particles, but the total particle size distributions generally show higher particle concentrations, particularly of Aitken mode particles. At temperatures above approximately -4°C, the ratio curves in Fig. 6c have a similar shape as for the pure liquid clouds (cf. Fig. 5). This could be an indication that

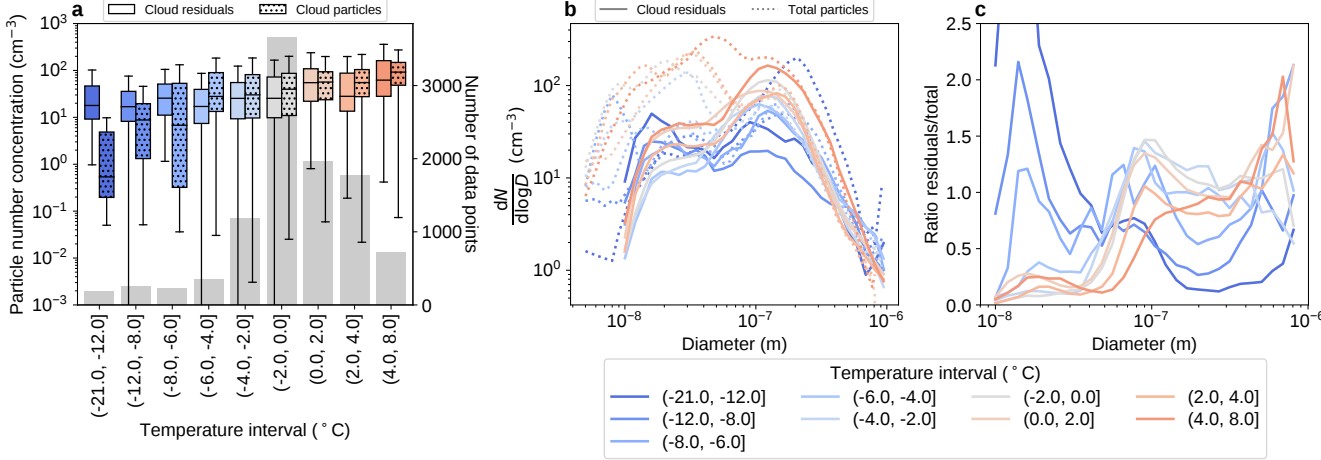

**Figure 6. In-cloud data binned by temperature. a** Box plot of cloud residual (solid) and cloud particle (hatched) number concentrations for different temperature intervals (see legend). The whiskers extend to the farthest points that are within 1.5 times the interquartile range from the nearest quartile. Points that fall outside the whiskers are not shown. The grey bars in the background indicate the number of cloud residual data points (right y-axis) per temperature bin. **b** Mean particle number size distributions of cloud residuals (solid) and total particles (dotted) for different temperature intervals (see legend). **c** Ratio of the size distributions in b, i.e. cloud residual concentrations divided by total particle concentrations.

in the temperature range -4–0°C we are mostly sampling supercooled droplets. The apparent $D_{50\%}$ decreases with decreasing temperature, which indicates an increase in cloud supersaturation with decreasing temperature. If the meteorological conditions are otherwise the same, this could be caused by an increase in updraft velocity or by a decrease in particle concentration (less

480    competition for water vapour allows smaller particles to be activated). The latter is consistent with the general decrease in particle concentrations with temperature seen in the first two panels of Fig. 6.

At temperatures below -4°C (approximately 11 % of this subset of data), however, the curves in Fig. 6c look very different. Instead of an S-shape, the curves are relatively flatter with a maximum appearing at lower sizes, with the coldest temperature bins even showing a peak below ∼20 nm particle diameter. Assuming that the measured cloud residuals directly correspond to

485    CCN or INP, this behaviour implies that many of the ambient accumulation mode particles have not activated, while now an increased contribution of Aitken mode particles served as cloud seeds. However, the peak ratio exceeds 2 for the two coldest bins. These clouds most likely contain ice particles and the question arises if the small particles could potentially be caused by sampling artefacts inside the GCVI system (see Sect. 3.1.2 above) or if a real physical atmospheric process is underlying this observation. This will be further discussed in Sect. 3.2.3.

 ### 3.2.3 Annual cycle

Our observations cover two full years and allow us to study seasonality effects, which are expected for this remote Arctic site, where aerosol properties follow recurring patterns (e.g., Tunved et al., 2013) that will also influence the seasonality of cloud residual properties. We will follow two approaches to study the seasonality of cloud residuals: (a) grouping the observations in liquid and mixed-phase clouds and (b) using a cluster analysis of the shape of the cloud residual size distributions. The latter
 approach will allow us to identify and quantify the impact of potential measurement artefacts or the contribution of ice which will be discussed in detail below.

**Separation between liquid and mixed-phase clouds**

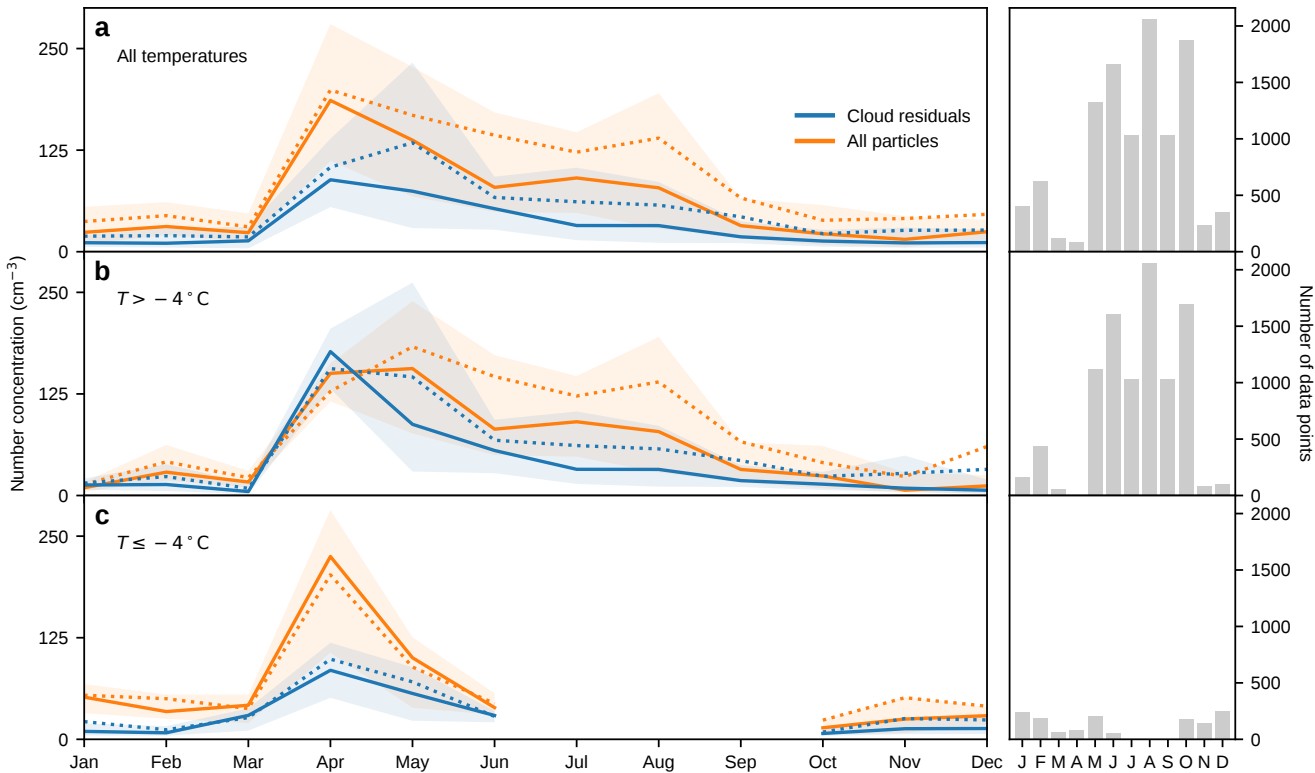

**Figure 7. Annual cycle of total and cloud residual number concentrations.** Monthly averages of total (orange) and residual (blue) particle number concentrations integrated above 20 nm, measured at Zeppelin Observatory, Svalbard during the period Nov 2015–Feb 2018. Solid and dotted lines show median and mean values, respectively, and shaded areas indicate the 25$^\text{th}$ to 75$^\text{th}$ percentile ranges. Data have been segregated based on the temperature at the station, $T$. The panels show data for **a** all $T$, **b** $T > -4°$C, **c** $T \leq -4°$C, and the corresponding bar charts show the number of data points per month.

Figure 7 shows concurrent monthly averages of total particle and cloud residual number concentrations integrated above 20 nm diameter. Panel a shows all data, whereas panels b and c show data for temperatures above and below $-4°C$, respectively. This boundary was chosen based on the different behaviours seen in Fig. 6c, where cloud residual size distributions below $-4°C$ are more likely to be influenced by ice, while cloud residual distributions above $-4°C$ are mostly liquid clouds. The bar charts next to each panel indicate the number of data points per month.

The observed total particle number concentrations follow the typical seasonal cycle of Arctic aerosol. We recognise the characteristic maxima in number concentration due to Arctic haze in spring and new particle formation in summer, and the low, relatively stable concentrations during the rest of the year. There are some differences compared to previous measurements at Zeppelin Observatory related to the natural variability of aerosols, for example in terms of when peak concentrations were observed (e.g., Ström et al., 2003; Tunved et al., 2013; Freud et al., 2017). Such differences could be due to annual variability, which has previously been shown to be significant (Freud et al., 2017). In addition, it should also be kept in mind that we present number concentrations exclusively during cloud events and concentrations shown are for particles above 20 nm diameter, in contrast to previous studies.

The cloud residual number concentrations, while lower than the total particle concentrations, display a similar seasonal behaviour. As seen in Fig. 2a, the shape and the magnitude of the cloud residual annual cycle agree nicely with the cloud particle annual cycle (note that Fig. 7a shows a slightly different subset of data and only includes cloud residuals above 20 nm).

In Fig. 7a, the overall cloud residual number concentrations range between 10 and 71 $cm^{-3}$ (25$^{th}$ and 75$^{th}$ percentiles), with a median of 26 $cm^{-3}$ (mean $\pm$ standard deviation: $56 \pm 77$ $cm^{-3}$). The corresponding total particle concentration during these cloud events is generally higher, ranging from 22 to 127 $cm^{-3}$ (25$^{th}$ and 75$^{th}$ percentiles) with a median of 55 $cm^{-3}$ (mean $\pm$ standard deviation: $101 \pm 143$ $cm^{-3}$). These numbers do not change appreciably when only clouds at temperatures above $-4°C$ (Fig. 7b) are considered, which shows that the annual cycle of cloud residual number concentrations is driven by mostly liquid clouds.

There are only a few long-term CCN data sets from the Arctic (Jung et al., 2018; Dall'Osto et al., 2017; Schmale et al., 2018) that we can compare our measurements to, and they are based on a different measurement technique. Cloud residual measurements differ from standard CCN measurements in that instead of attempting to replicate in-cloud conditions inside the instrument – most notably fixed supersaturation bands in place of dynamic ambient conditions – we extract cloud particles from the air, dry them and subsequently count and size the cloud residuals. Since these other techniques cannot measure INP, we will only compare them to our liquid cloud-dominated data.

Jung et al. (2018) found that CCN concentrations correlated well with concentrations of accumulation mode particles at Zeppelin Observatory, and that median CCN concentrations peaked in March at most supersaturation levels. This is slightly different from our measured cloud residual concentrations, which peak in April (Fig. 7b). However, one should keep in mind that Jung et al. (2018) considered different years (2007–2013) and did not differentiate between in- or out-of-cloud periods. In addition, Jung et al. (2018) observed for most of the year higher CCN concentrations than our cloud residual concentrations, particularly in winter, highlighting the differences between measurement techniques. Studies where particles are artificially activated, i.e. at a fixed supersaturation, are independent of the ambient meteorology and atmospheric dynamics, whereas our

study inherently takes the ambient conditions into account by sampling the actual cloud droplets or ice crystals. Therefore, the differences between our observed concentrations could either be because the actual ambient supersaturations are lower than what is used in CCN counters, or because the data in Fig. 7b still include some ice processes that can influence the droplet concentration (e.g. Wegener–Bergeron–Findeisen) while CCN counters only consider liquid droplet activation.

**Cluster analysis**

While Fig. 6c indicated that the cloud residuals behave differently, on average, below $-4°C$, using a strict temperature cut introduces some problems with data availability (as seen in Fig. 7c) and interpretation – is there really a physical reason to expect different behaviours on either side of this specific temperature boundary? Cluster analysis provides a means of sorting the data based on the shape of the cloud residual size distributions, which allows us to study the annual cycle of behaviour without relying on an external parameter for grouping the data. For completeness, monthly average cloud residual size distributions above and below $-4°C$ are shown in Fig. S9.

We performed $k$-means clustering on the cloud residual number size distributions (normalised by the corresponding total cloud residual number concentration). The results when 5 clusters are used are presented in Fig. 8. The clusters are numbered from 1 to 5 according to increasing modal diameters of the cluster average size distributions (the approximate modal diameters are 15, 30, 65, 100, and 150 nm). This order is also reflected in the number concentration of both cloud residuals, cloud particles, and total particles (cf. Fig. 9d–e). Cluster 2 is the most frequent cluster (27 % of the time) spread throughout the year but less in spring and early summer. This is followed by Clusters 5 and 4 (26 % and 25 %, respectively) which are more dominant in spring and summer. Cluster 3 (14 % of the time) occurs more in late summer and autumn, while Cluster 1 is the least frequent (8 % of the time) and occurs mostly during winter.

*Clusters 3–5*

Clusters 3–5 show almost identical cloud particle number size distributions with a mode around 12 $\mu$m cloud particle diameter. The cloud residual number size distributions are all dominated by the accumulation mode and have a similar shape, although with different modal diameters as mentioned above. These clusters all agree quite well with the concentrations from the FM-120, with 95% of the residual concentrations being within a factor of 5 from the cloud particle concentrations (Fig. S10c–e). While all clusters occur throughout the year, these three clusters occur mainly during the warmer months, and thus correspond mainly to liquid clouds. Comparing the target classification of Cloudnet above Ny-Ålesund, at the altitude around Zeppelin Observatory, to the cluster analysis indeed shows a lower ratio of ice to liquid occurrence (Cloudnet category 4 divided by category 1) for these clusters compared to the other two (Fig. 10 and Fig. S11). Note that the Cloudnet comparison could only be made for a subset of the data (see Sect. 2.3.4 and Table S1), but the relative cluster occurrence in this subset is similar to that in the full data set. It should be noted that all Cloudnet classification categories appear in each of the clusters (Fig. S11), so the cloud residual size distribution shapes cannot be solely attributed to one cloud particle type. However, even for mixed-phase clouds the majority of the particle concentration is likely to be made up of supercooled droplets rather than ice crystals. A case study of a cluster 5 cloud event is shown in Fig. A1 and described in Appendix A1.

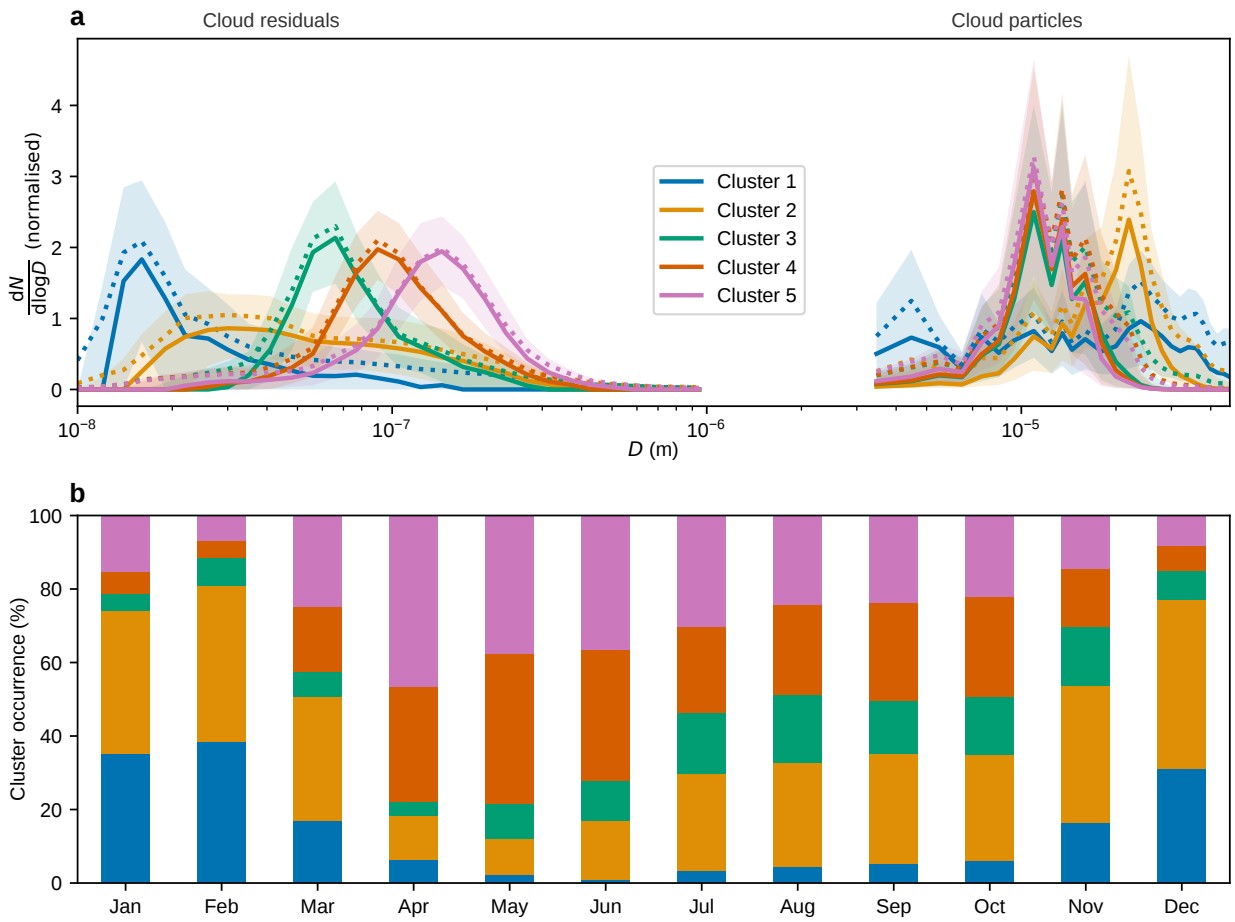

**Figure 8. Results of $k$-means clustering of cloud residual number size distributions using 5 clusters. a** Normalised cloud residual number size distributions for each cluster (left), and normalised number size distributions for the corresponding cloud particle population (right). Solid and dotted lines show median and mean values, respectively, and shaded areas indicate the 25[th] to 75[th] percentile ranges. **b** Monthly frequency of occurrence of each cluster.

The ratios between the cluster average cloud residual size distributions and corresponding average total particle size distributions, which for liquid clouds would correspond to the activation ratio, are shown in Fig. 11. Indeed, for clusters 3–5, the ratios are more or less S-shaped (albeit more so for cluster 5 than the other two), as would be expected from the classical Köhler theory of liquid droplet activation (assuming a size-independent chemical composition). In the accumulation mode size range, 570   the ratios show a wavy behaviour where, instead of levelling out around 1, they drop and then increase again. As stated in Sect. 3.2.1, this could be a result of the cloud residual and total particle size distributions not being perfectly aligned (e.g. due to uncertainties in sizing, losses, or perhaps evaporation of volatile material from the residuals).

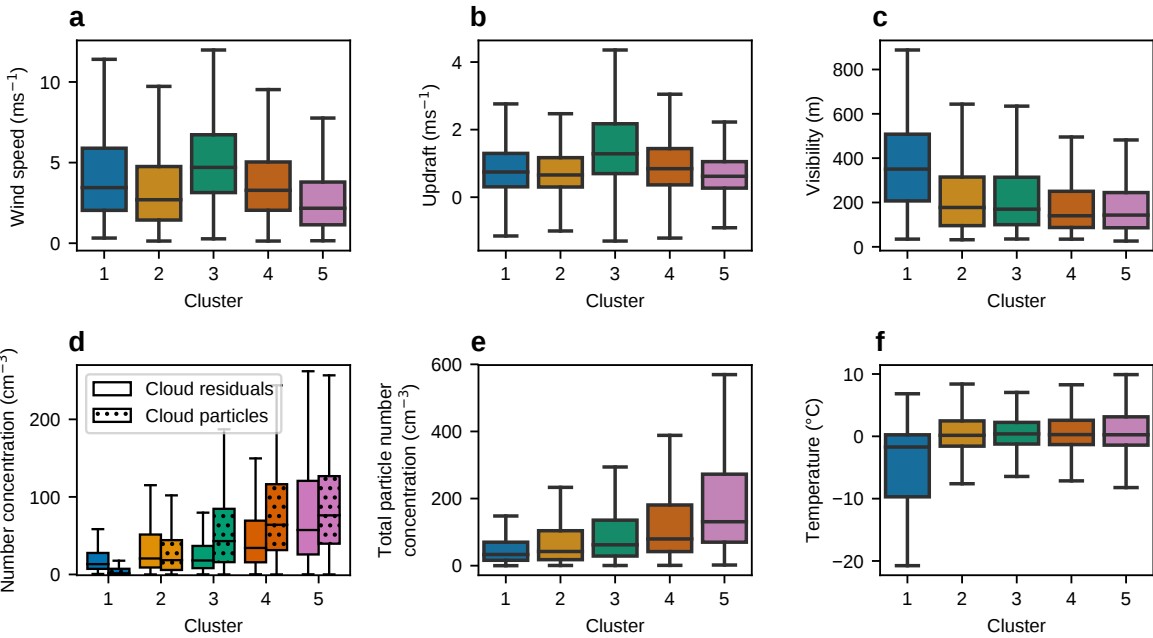

**Figure 9. Additional parameters for the cluster analysis from Figure 8.** The panels show the distribution per cluster of **a** wind speed, **b** updraft, **c** visibility, **d** cloud residual (solid; corrected for GCVI sampling efficiency) and cloud particle (hatched; without any correction factors) number concentrations, **e** total particle number concentration and **f** temperature. The whiskers of the box plots extend to the farthest points that are within 1.5 times the interquartile range from the nearest quartile. Points that fall outside the whiskers are not shown.

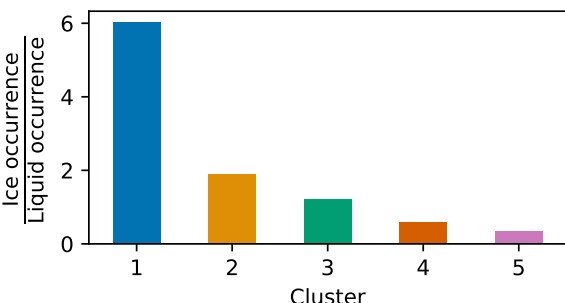

**Figure 10. Ratio of ice to liquid occurrence per cluster based on Cloudnet retrieval.** The bar chart shows, for each cloud residual size distribution cluster, the average ratio of pure ice to pure liquid occurrence, i.e. Cloudnet category 4 divided by Cloudnet category 1, around the altitude of Zeppelin Observatory. See Fig. S11 for a more detailed view on the average Cloudnet retrieval target classifications for each cluster.

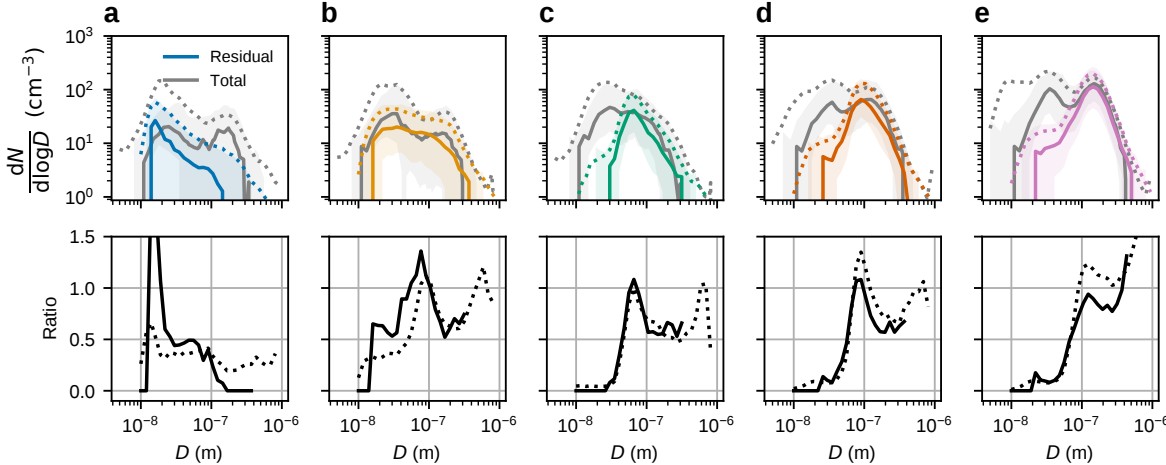

**Figure 11. Size distributions and activation ratios for the clusters from Figure 8.** The top row shows cloud residual size distributions (in colour) and the corresponding total particle size distributions (grey). Solid and dotted lines show median and mean values, respectively, and shaded areas indicate the $25^{\text{th}}$ to $75^{\text{th}}$ percentile ranges. The bottom row shows activation ratios calculated from the median (solid) and mean (dotted) distributions. Columns **a–e** show Clusters 1–5, respectively.

The modal diameters of clusters 3–5 in Fig. 8a and the $D_{50\%}$ in Fig. 11c–e correlate with the total particle number concentration (Fig. 9e) and anticorrelate with updraft velocity (Fig. 9b). A relationship between updraft velocity and $D_{50\%}$ was also seen in Sect. 3.2.1. As previously stated, both an increased updraft velocity and decreased particle concentrations can allow smaller particles to activate (assuming the meteorological conditions are otherwise the same).

Clusters 4 and 5 together make up just over half of all data, and they are also associated with the highest total particle number concentrations (cf. Fig. 9d–e). Thus, these clusters have a large influence on the overall annual cycle of particle number concentrations. From Fig. 8b, it can be seen that the occurrence of cluster 5 peaks in April, when total particle concentrations peak (cf. Fig. 7). This indicates that perhaps cluster 5 represents a typical cloud residual size distribution associated with the Arctic Haze.

### *Clusters 1 and 2*

Clusters 1 and 2 clearly deviate from the other three clusters, both in terms of the shape of the cloud residual size distribution and the cloud particle size distribution. They are also very different from each other, but what they have in common is that they show cloud residual size distributions dominated by Aitken mode particles. While cluster 1 consists almost entirely of Aitken mode particles, cluster 2 is bimodal with two broad size modes (i.e. it contains both Aitken and accumulation mode particles). Cluster 1 is not related to any clear cloud particle size mode but, in relative terms, it is associated with larger cloud particles than clusters 3–5 (Fig. 8a). Cluster 2 is also associated with larger cloud particles with a mode around $23\,\mu$m (cf. $12\,\mu$m for clusters 3–5). Clusters 1 and 2 also have the lowest cloud particle and cloud residual number concentrations of

the clusters (Fig. 9d). These factors suggest they represent thin clouds with few, large droplets and/or ice crystals. This is further corroborated by the visibility distribution (Fig. 9c) which shows high values, in particular for cluster 1. No clear relationship between these two clusters and wind speed or updraft was found (Fig. 9a–b).

Cluster 1 is the one that stands out the most. The shape of the cloud residual size distribution is peculiar, with peak concentrations occurring below 20 nm particle diameter. Cluster 1 occurs primarily during the winter months, and at considerably higher visibility and lower temperature and cloud particle concentrations than the other clusters (Fig. 9). These conditions would be consistent with ice or mixed-phase clouds in the winter. The Cloudnet analysis shows that cluster 1 has by far the highest occurrence of cases with ice crystals compared to cases with liquid droplets (Figs. 10 and S11a). Interestingly, the ratio of ice to liquid cases decreases from cluster 1 to cluster 5, which is consistent with the activation ratios in Fig. 11 which appear more like classical Köhler activation (of homogeneously mixed particles) when moving from cluster 1 to cluster 5.

The shape of the cloud residual size distribution of cluster 1 compared to the ambient particle size distribution (cf. Fig. 11a) reveals that the accumulation mode particles do not activate. In mixed-phase clouds, supercooled droplets outnumber ice crystals, often by orders of magnitude (e.g., Young et al., 2016), so we should be seeing accumulation mode cloud residuals stemming from the supercooled droplets in addition to the small particles. Verheggen et al. (2007) observed a decreased activated fraction (of particles larger than 100 nm) with decreasing temperature similar to our observations (see Fig. S12), which they attributed to the Wegener-Bergeron-Findeisen process, i.e. evaporation of liquid droplets promoting ice crystal growth. This might be an explanation for the missing accumulation mode in cluster 1; however, it does not explain the peak activated fraction we observe around 20 nm (which was not observed in Verheggen et al. (2007)). Part of cluster 1 is associated with very low cloud particle concentrations ($< 1\,\mathrm{cm}^{-3}$) and cold temperatures (down to $-21°C$), which may also be consistent with pure ice clouds. However, some of the cloud residuals we have measured, in particular those in cluster 1, are much smaller than typical INP (Hoose and Möhler, 2012; DeMott et al., 2010) (or indeed CCN). Residual size distributions with a very similar shape as the one of cluster 1 have previously been observed for ice particles in mixed-phase clouds measured with an Ice-CVI (Mertes et al., 2007), as well as for cirrus clouds using an airborne CVI (Seifert et al., 2003). Although those studies used different techniques and sampled different cloud types, it could be an indication that cluster 1 is the result of ice particles sampled by the GCVI. It should, however, be noted that the total aerosol size distributions in the aforementioned studies look different than in the present study, and consequently their activated fractions do not show the same behaviour as our cluster 1.

The question arises whether these small particles are really cloud residuals, or if they are measurement artefacts. As discussed in Sect. 3.1.2, artefacts in the form of ice crystal shattering cannot be fully ruled out. Large crystals are expected to be more prone to shattering, and indeed clusters 1 and 2 are related to larger cloud particles than the other clusters (Fig. 8a). Cloudnet does not distinguish between cloud ice and precipitating ice, so we could also be dealing with snowflakes. The average cloud particle size distribution associated with cluster 1 is rather flat with no obvious mode (Fig. 8a), and it is also associated with very low cloud particle concentrations (Fig. 9d). This could just be noisy measurements in the fog monitor during snowfall, and would indicate that cluster 1 is influenced by snow. Cluster 2, on the other hand, has a clear cloud particle size mode, although at a larger diameter compared to cluster 3–5 (Fig. 8a), and the cloud residuals are therefore much less likely to stem solely from precipitation.

The median cloud residual concentration is larger than the median total particle concentration for cluster 1 around 20 nm (Fig. 11a), which would suggest that there is a risk of crystal shattering artefacts. As shown above, Figure 2b revealed two groups of data, where one showed a discrepancy between measured cloud residual and cloud particle concentrations as would be expected with this type of artefact. The same figure separated by cluster (Fig. S10) shows that this group of data is over-represented in clusters 1 and 2, which speaks in favour of the crystal shattering hypothesis as well. However, cloud residual size distributions with modal diameters similar to those of clusters 1 and 2 still appear in a cluster analysis where all the data outside the 10:1 and 1:10 lines in Fig. 2b are excluded (Fig S13a; note, results do not change if we are even stricter, i.e. within 1:2 and 2:1). This stricter cluster analysis also shows a cloud particle mode for the small cloud residual cluster (Fig S13a), as opposed to the flat cloud particle size distribution for cluster 1 in Fig. 8, indicating a decreased relative influence from snow in the stricter analysis. The fact that cloud residual size distributions similar to cluster 1 still appear in Fig S13 suggests that while ice crystal shattering is certainly a possibility, it is not necessarily the only explanation for the shape of the size distributions we observe.

The presence of 20 nm particles is also observed in the whole-air inlet for the same times as cluster 1 occurs (Fig. 11a). The whole-air inlet is very different from the GCVI inlet (e.g. in terms of flows, velocities and how particles move inside the inlets), so it is unlikely that both inlets would produce artefacts with the same frequency and particle size. Yet, if the small particles are real, where do they come from? In the Arctic and marine boundary layer, the presence of particles below ∼50 nm is most often associated with new particle formation (Ström et al., 2003; Tunved et al., 2013) or even primary emissions of sea spray particles (Ovadnevaite et al., 2011). However, these sources are unlikely to explain the presence of small particles during winter, when there is reduced or no sunlight (i.e. no photochemistry), less biological production and most of the sea surface is covered by ice (Dall'Osto et al., 2017; Sharma et al., 2012). Other potential sources can be long-range transport, but the lifetime of Aitken mode aerosol particles in the boundary layer is rather limited, or entrainment from the free troposphere. However, this is purely speculative and future studies are needed to investigate the exact sources and chemical nature of these small particles.

At this point, it is important to point out that, even barring artefacts, a cloud residual does not necessarily correspond directly to a CCN or INP. Cloud residuals can also be nuclei that have undergone processing inside the cloud (be that chemical or physical), and can contain material from e.g. riming or aerosol particles that have been scavenged by the cloud particles. Unfortunately, we have no way of distinguishing between these particle types, especially since the FM-120 cannot differentiate between cloud droplets and ice crystals. It could potentially be that the crystals we measure are the result of secondary ice formation processes (Field et al., 2016), which has been suggested in a model study to be important for Arctic stratocumulus clouds (Sotiropoulou et al., 2020). In ice crystals formed by droplet freezing, solute material from CCN or scavenged particles could be built into the crystal structure, and that material could then be distributed across the splinters when secondary ice formation happens. In other words, the cloud residuals we measure may be remnants of CCN and/or scavenged particles, which would explain their small size.

Since secondary ice formation happens before the cloud particles enter our inlets, these particles should also be seen by the fog monitor. This is often not the case for cluster 1, as seen in Fig. S10, which means that secondary ice particles cannot be the

only reason for the small residuals we observe (unless the ice crystals are undersampled by the fog monitor, see Sect. 3.1). Cloud observations at mountain-top stations such as Zeppelin Observatory may also be influenced by surface processes (e.g. blowing snow) that could increase the ice crystal concentrations (Beck et al., 2018), but this, too, should be seen by the fog monitor and as discussed earlier, no clear dependency on wind-speed has been observed. As stated earlier, it is not possible to translate the cloud residual data to CCN, INP, etc without further detailed information on cloud phase, structure and origin. The cloud

phase is an important parameter and should as such be added in future studies.

In summary, it seems likely that cluster 1 is significantly influenced by snow and ice. It is difficult to say to what extent the signal is caused by crystal shattering artefacts as compared to other processes, but cold temperature outliers (cf. Sect. 3.1.2) make up roughly 45% of cluster 1 (cf. Fig. S10a). These data should be treated with caution, but there are some plausible physical explanations for the presence of small cloud residuals when the agreement with the fog monitor is better, e.g. secondary

ice processes, yet further measurements would be needed to verify this. The possibility that such processes would show a signal similar to shattering artefacts is an important consideration when analysing GCVI data from ice or mixed-phase cloud conditions. However, from an aerosol activation perspective, it is irrelevant whether the snow or ice crystals shatter before or after they enter the inlet – in both cases, the resulting cloud residuals do not represent CCN or INP. A case study of a cluster 1 cloud event is shown in Fig. A3, which is an example that illustrates the difficulties with interpreting this cluster (see

Appendix A2).

Cluster 2 is also Aitken mode dominated, and occurs throughout the year (27 % of the time, or 13 % of the time if we only consider $T > 0°$C). The cluster is bimodal, but relatively broad and flat. While the same approximate size modes are present in the total particle size distribution, the average cluster 2 cloud residual size distribution has a less pronounced minimum and slightly lower concentration of accumulation mode particles than the total particle distribution. The shape of cluster 2 could

perhaps also be influenced by ice processes (i.e. not all cloud residuals correspond to CCN), or the shape might be affected by evaporation of volatile compounds from the accumulation mode particles. However, this cannot be confirmed without size-resolved chemical composition or volatility measurements, which were not available for our period.

Unlike cluster 1, cluster 2 is much less likely to be affected by snow artefacts. While the exact contribution is difficult to quantify, cold temperature outliers only make up about 13% of cluster 2 (cf. Fig. S10b), i.e. there is a significantly better

agreement with the fog monitor than for cluster 1. Cluster 2 is also different from clusters 3–5 but, in contrast to cluster 1, it was observed more homogeneously throughout the year. In further contrast to cluster 1, the meteorological parameters related to cluster 2 are not distinctly different from those related to clusters 3–5 (cf. Fig.9). This means that cluster 2 was also observed during sampling conditions when we can safely rule out the influence of mixed-phase clouds and ice crystals. Many of the caveats listed above related to cluster 1 thus do not apply to cluster 2 to the same extent. In addition, cluster 2 does

not show the lack of accumulation mode particle activation that complicated the interpretation of cluster 1. Hence, the Aitken mode cloud residuals in cluster 2 very likely contain activated aerosol particles. Similar findings were reported in previous CVI measurements (Schwarzenboeck et al., 2000), although not in the Arctic. In the Arctic, activation of Aitken mode aerosol particles has been shown by indirect means and model studies (e.g., Korhonen et al., 2008; Leaitch et al., 2016; Koike et al.,

2019; Bulatovic et al., 2021). A case study from our dataset of a cloud event that is a mixture of clusters 1 and 2 can be found
in Beck et al. (2021, Fig. S8, supporting information).

## 4  Conclusions

Results presented in this paper are the first direct long-term measurements of size-resolved cloud residual number concentrations of Arctic low-level clouds. It is also the first cloud residual data set that covers more than a full annual cycle, in the Arctic and globally.

We conducted a thorough evaluation of the GCVI measurements by comparing them to cloud particle size distributions as measured by an FM-120 fog monitor, as well as to total particle size distributions measured behind a whole-air inlet. We derived correction factors for the cloud residual measurements based on the cloud particle data and the experimentally determined sampling efficiency of the GCVI. For warm clouds, we could also derive a correction factor by comparing cloud residual and whole-air accumulation mode particle concentrations (under the assumption of liquid droplet activation with no

size-dependent chemical composition), and we found that both methods agreed remarkably well. Our data set includes the winter months, when Arctic warming is most pronounced and clouds are hypothesised to play a key role. However, as it turns out, the winter months are not entirely straight-forward to analyse. We identified a group of data at cold temperatures where the cloud residual and cloud particle concentrations did not agree well. It is likely that this is a result of snow or ice crystal shattering artefacts. However, these points are a small percentage ($\sim$7–8%) of the total data and the majority of the data are

not affected by potential sampling artefacts.

Our measured cloud residual number concentrations generally follow the typical annual aerosol cycle previously reported for this site. For pure liquid clouds (T$>$ 0$^\circ$C), we observed activation diameters ($D_{50\%}$) in the range of 58–78 nm (for updraft velocities below $1\,\mathrm{ms}^{-1}$), where smaller activation diameters were associated with higher updraft velocities. A relationship between a decreasing total particle number concentration and a decrease in $D_{50\%}$ could also be inferred from the cloud residual

size distributions binned by temperature. Both a change in updraft velocity and a change in particle number concentration can affect the supersaturation, but we cannot clearly disentangle the influence of these parameters. The cluster analysis of cloud residual size distributions for liquid clouds (clusters 3–5) also showed that smaller cloud residuals were associated with higher updraft velocities and lower particle number concentrations.

From late spring to early autumn, the cloud residual size distributions at Zeppelin Observatory are dominated for most of the

time by the accumulation mode with clouds consisting mostly of liquid droplets (clusters 3–5). In late autumn to early spring, we found, in relative terms, a significant contribution of Aitken mode particles to the cloud residual number concentration (clusters 1 and 2). However, the presence of the ice phase and snow complicates matters. The mode of smallest particles we observed (cluster 1) is most likely due to artefacts of crystal shattering within the wind tunnel of the GCVI or potentially caused by fragments of CCN or scavenged particles created through secondary ice multiplication processes. With our instrumental set

up, the contribution of these different processes cannot be confidently quantified. As far as cluster 2 is concerned, while artefacts and ice processes cannot be completely ruled out, we believe that the majority of the signal is real and shows new experimental

evidence of the activation of aerosol particles down to $\sim 20$–$30\,\mathrm{nm}$ in the Arctic, confirming results from previous experimental and modelling studies.

In-situ sampling of cloud droplets and ice crystals are a complex challenge. Detailed cloud phase measurements, i.e. the
ratio of ice crystals to liquid droplets within the cloud and close to the GCVI, using a more sophisticated cloud probe will be
needed to better understand the relative importance of CCN / INP and the importance of other related in-cloud processes. To
study ice and liquid cloud particles separately, it would also be desirable to deploy ice-selective inlets (e.g., Mertes et al., 2007;
Kupiszewski et al., 2015; Hiranuma et al., 2016) at Zeppelin Observatory in the future; however, long-term deployment and
potential artefacts remain a challenge. In addition, detailed and size-resolved chemical composition and volatility measure-
ments of the sampled cloud residuals and the contribution of supermicron particles would help to better understand the sources
and processes related to low-level Arctic cloud formation.

*Data availability.* The data of this study will be available on the Bolin Centre Database (DOI and link will be added later). The Cloudnet
data are available on the Cloudnet website (http://devcloudnet.fmi.fi/)

*Author contributions.* LK analysed the data and wrote the manuscript. PZ and RK designed the study, carried out the observations and
contributed to writing the manuscript. MK provided the cloud droplet distribution data. KE provided Cloudnet data. All authors discussed
the results, read and commented on the manuscript.

*Competing interests.* PZ and RK are currently acting as co-editors in ACP. The authors declare no further competing interests.

*Acknowledgements.* We would like to thank research engineers Tabea Henning, Ondrej Tesar and Birgitta Noone from ACES and the staff
from the Norwegian Polar Institute (NPI) for their on-site support. NPI is also acknowledged for substantial long-term support in maintaining
the measurements at Zeppelin Observatory. We thank Fred Brechtel, Johan Ström, Urs Baltensperger, Joel Thornton, Samuel Lowe, Heike
Wex, Ilona Riipinen, Darrel Baumgardner, Ernest Weingartner, and Annica Ekman for valuable discussions.

This work was financially supported by the Knut-and-Alice-Wallenberg Foundation within the ACAS project (Arctic Climate Across
Scales, project no. 2016.0024), the Swedish EPA's (Naturvårdsverket) Environmental monitoring program (Miljöövervakning), and the
Swedish Research Council FORMAS (Project "Interplay between water, clouds and Aerosols in the Arctic", # 2016-01427)

We gratefully acknowledge the funding by the Deutsche Forschungsgemeinschaft (DFG, German Research Foundation) - project number
268020496 - TRR 172, within the Transregional Collaborative Research Center "ArctiC Amplification: Climate Relevant Atmospheric and
SurfaCe Processes, and Feedback Mechanisms (AC)[3]".

This study was partly supported by the Ministry of Education, Culture, Sports, Science, and Technology (MEXT ArCS Project) and the Environment Research and Technology Development Fund (2-1703 and 2-2003) of Environmental Restoration and Conservation Agency in Japan.


## Appendix A: Case studies

### A1 Case I: Cluster 5 cloud event

The first case study is a cloud event on the 3rd of October 2017 which is entirely dominated by cluster 5, a cluster with a cloud
residual mean modal diameter of around 150 nm. Figure A1 shows the evolution of various cloud, aerosol, and meteorological
parameters before, during, and after the cloud event. The cloud itself is a thick, most likely liquid cloud – the temperature
during the event is around -2–0°C, and the visibility is low and even below 100 m towards the end of the event (Fig. A1h).
There is a clear cloud droplet mode, with a modal diameter that increases from 10 to just under 20 $\mu$m during the course of the
event (Fig. A1c). This case is an example of when our instrumentation and analysis is expected to work best: a cloud with a
well defined (and not overly large) cloud droplet mode, easily measured by both the GCVI and the FM-120, during conditions
when the cloud is likely to be purely liquid, thereby limiting the probability of any crystal shattering artefact influence.

The total particle number size distribution is relatively constant with a broad mode around 100–200 nm, with slightly decreas-
ing particle concentrations during and after the cloud (Fig. A1a,e–g). The concentration decrease can be due to wet removal,
but could also partly be due to a change in air mass since the wind direction changes completely towards the end of the cloud
event (Fig. A1i).
The cloud residual number size distribution looks very similar to the total particle size distribution (Fig. A1b,f) and shows
the typical behaviour one would expect from liquid droplet activation, i.e. activation from large to small diameters. In this case,
the entire accumulation mode and a small fraction of the Aitken mode is activated into cloud droplets (Fig. A1f). We observe
good closure between cloud residual, total particle, and cloud particle concentrations (see also Fig. A2a below).

### A2 Case II: Cluster 1 cloud event

The second case, from the 14th of February 2017, is an example of a cloud event that, at least during the first two thirds of the
event, is dominated by cluster 1. This cloud is much thinner and colder than the cloud in Sect. A1, with ambient temperatures
between around -10 and -15°C and visibility oscillating between a kilometre and a few hundred metres (Fig. A3h). These are
conditions during which both measurements and analysis are more complicated to carry out, as has been discussed in the main
text (see e.g. Sect. 3.1.2).
As can be seen in Fig. A3c, the fog monitor detects almost no cloud particles during the part of the event that is dominated
by cluster 1, despite the fact that the visibility is relatively low. In the middle of the event, when the visibility is around 200 m,
one can see hints of a large cloud particle mode appearing in Fig. A3c, but it is clear that these particles are fewer than the
cloud residuals (cf. Fig. A3b, see also Fig. A2b). These data points belong to the group of outliers discussed in Sect. 3.1.2, and
it is unclear whether the disagreement is a result of issues with the FM-120 sampling, issues or artefacts within the GCVI, or
a combination of both. Snow flake or ice crystal shattering has been mentioned as a potential cause, and it is one plausible
explanation in this case as well since the webcam image from the middle of the cloud event shows that it was snowing for at
least part of the event (Fig. A3h).

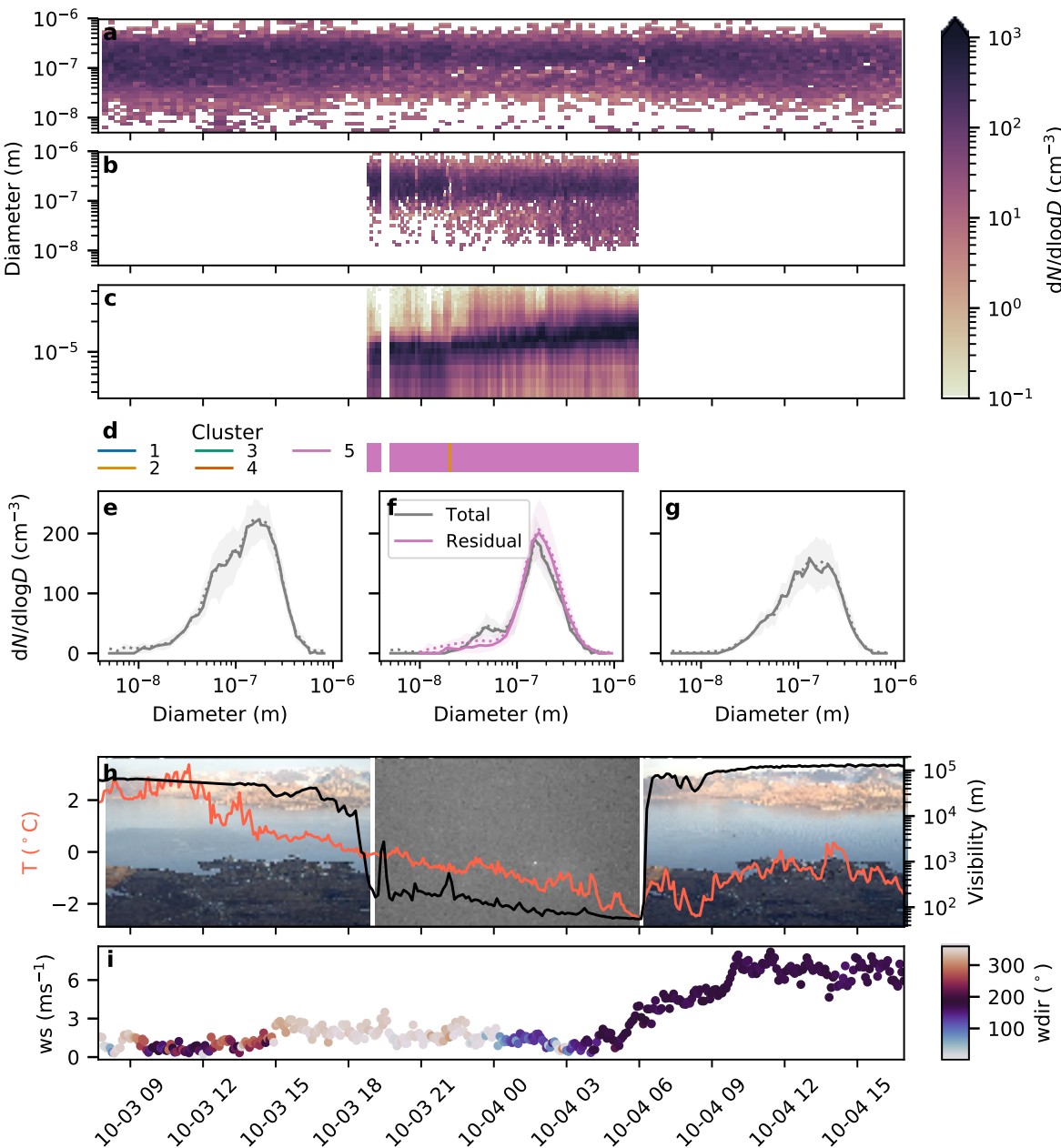

**Figure A1. Case study I: Cluster 5 cloud event on 2017-10-03.** The first three panels show **a** total, **b** cloud residual, and **c** cloud particle number size distributions and their evolution over time. **d** shows the cluster classification during the cloud event. Panels **e–g** show the average total particle size distributions **e** before, **f** during, and **g** after the cloud event. Solid and dotted lines show median and mean values, respectively, and shaded areas indicate the $25^{th}$ to $75^{th}$ percentile ranges. **f** also includes the cloud residual size distribution data in colour. **h** shows temperature (red, left) and visibility (black, right) over time, with background images of one webcam photo from Zeppelin Observatory (Pedersen, 2013) for each of the three periods. **i** shows wind speed, colour-coded by wind direction.

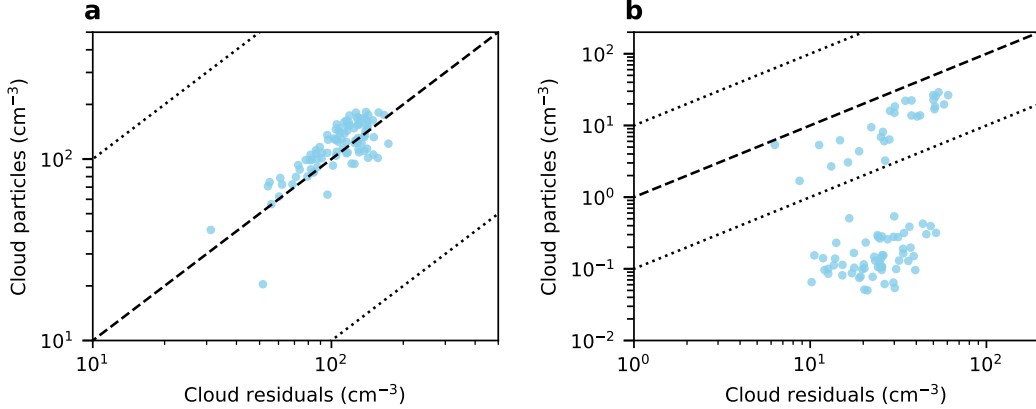

**Figure A2.** Scatter plots of cloud residual concentrations (corrected with individual factors, see Sect. 3.1.1) versus total cloud particle concentrations for **a** Case I (2017-10-03) and **b** Case II (2017-02-14). In both panels, the black dashed line represents the 1:1 line and the dotted lines represent 10:1 and 1:10 lines.

Comparing the cloud residual and total particle size distributions, the concentration of Aitken mode cloud residuals exceeds the total number of Aitken mode particles (Fig. A3f). This would also point to some form of sampling artefact being involved.
However, it is important to note that the Aitken mode is present in the total particle size distributions both before, during and after the event although at a lower concentration and with slightly different particle diameters (Fig. A3a, e–g), so there must also be some source of small particles that is not only related to sampling artefacts in the GCVI (note that the end of the cloud event is associated with a change in wind direction, so the particle size distribution after the event may be related to a different air mass, but the reasoning still stands for the before-cloud distribution).
Figure A3f also includes the median cloud residual size distribution obtained if we would assume the average transmission efficiency, i.e. multiplying by a factor of 2.2 instead of by an individual correction factor for each scan (see Sect. 3.1.1), to demonstrate the effect of correcting with an individual correction factor during cloud events when the fog monitor did not detect sufficient cloud particles. In this case, the wind was coming from the south with low wind speeds, when the fog monitor should be less affected by losses (see panel c, first half and panel i), while the visibility was low (see also visibility closure in
Fig. S5). As mentioned above, this could be due to the contribution of snow or large ice crystals that are not detected by the fog monitor. In the case of a constant correction factor, while the cloud residual concentrations still exceed the total Aitken particle concentrations, the disagreement is less severe. This example generally highlights the uncertainties that come with the assumptions made about instrument sampling efficiencies that involves other potentially biased cloud probes (e.g., with limited size range or own cloud particle loss issues). In this case, using an individual correction factor artificially increases the
contribution of cloud residual size distributions that are potentially affected by sampling artefacts resulting from snow or large ice crystals. The data for thin, ice and/or mixed-phase clouds are particularly difficult to interpret, and measurements such as ours should be complemented by additional data on cloud phase and cloud residual chemical composition in future work.

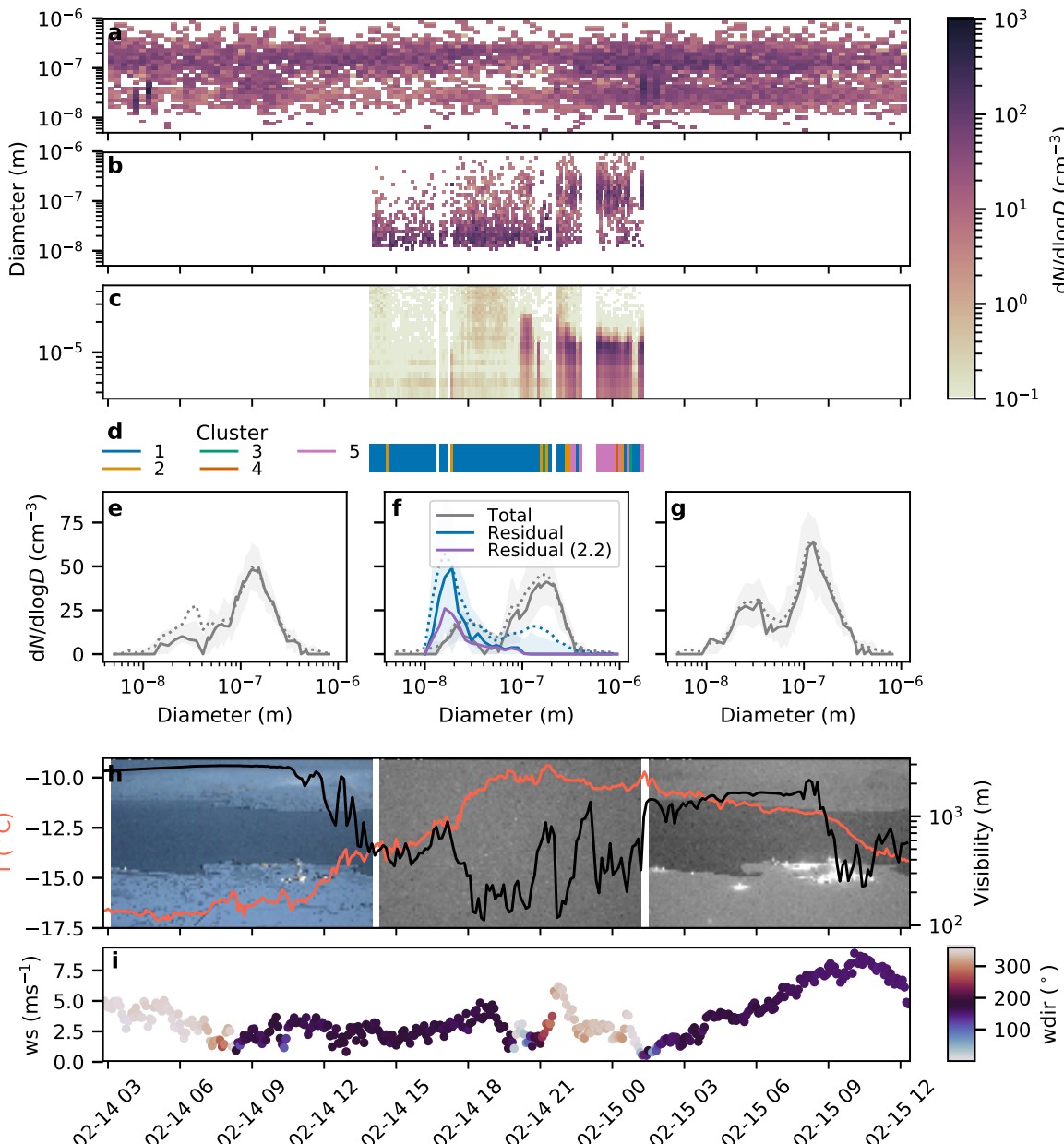

**Figure A3. Case study II: Cluster 1 cloud event on 2017-02-14.** The first three panels show **a** total, **b** cloud residual, and **c** cloud particle number size distributions and their evolution over time. **d** shows the cluster classification during the cloud event. Panels **e–g** show the average total particle size distributions **e** before, **f** during, and **g** after the cloud event. Solid and dotted lines show median and mean values, respectively, and shaded areas indicate the 25<sup>th</sup> to 75<sup>th</sup> percentile ranges. **f** also includes the cloud residual size distribution data in colour (blue). The median cloud residual size distribution if one would use an average correction factor instead of individual correction factors is shown in purple, for comparison. **h** shows temperature (red, left) and visibility (black, right) over time, with background images of one webcam photo from Zeppelin Observatory (Pedersen, 2013) for each of the three periods. **i** shows wind speed, colour-coded by wind direction.

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
