# Peer review of "A long-term study of cloud residuals from low-level Arctic clouds"

_Atmospheric Chemistry and Physics, 2020_

## Referee Comment (RC1) · Anonymous Referee #2 · 18 Jun 2020

Main general comments:

The authors present long term, ground-based measured cloud particle residual number concentration and size distribution results in combination with other aerosol and cloud particle and meteorological data. Like the authors point out, this is indeed a remarkable data set, helping to improve the knowledge about arctic cloud formation.

But:

A. According to the diverse uncertainties (representativity of the measurement location, GCVI and FM-120 sampling efficiency, unassignable assumptions in certain parts of the data interpretation), which are honestly admitted in the manuscript, the conclusions are much too high-flown. Examples are "The reported measurements...provide a new basis...for developing robust parameterizations of mixed-phase clouds in Earth system models" (line 7-9) or "The direct measurements...provide a valuable new perspective

on Arctic CCN and INP" (line 551-552), although not one INP size distribution or INP concentration is presented in the manuscript. The ball should be kept low with regard to these concluding statements.

B. The main and crucial point of criticism is the discussion and evaluation of the small residual particles (

D. It is known that the results of many airborne CVI measurements in mixed-phase clouds suffer from artefacts by large cloud particle shattering and that beside this effect, the results are hard to interpret with respect to CCN and INP, because such a CVI exactly like the used GCVI cannot differentiate between droplets and ice particles. This was the motivation to develop at least ground-based CVI systems that are able to separate the liquid and iced cloud phases, like the Ice-CVI (Mertes et al., 2007; Kupiszewski et al., JGR, 2016), the Ice Selective Inlet ISI (Kupiszewski et al., ACP, 2015) or the ice selecting pumped CVI IS-PCVI (Hiranuma et al., AMT, 2016). With those systems the residuals of ice particles can be separated and measured in mixed-phase clouds and would be thus needed for the Mt. Zeppelin measurements when mixed-phase clouds prevail.

E. Another drawback is the choice of the cloud particle sensor, which is the major reference instrument for the GCVI in this study. Due to its fixed position its detection efficiency is not clear and moreover it is also not able to distinguish between droplet and ice particles. But for the wintertime measurements and the discussion that the Aitken mode particles are related to ice formation in the cloud a ice particle sensor is crucially needed. It would also shed light into the question whether the measurements are real or mostly caused by artefacts.

Concluding, the manuscript has to be refused in the actual form including the discussion of the wintertime, most likely mixed-phase clouds residual measurements, where the Aitken mode residuals are assumed to be involved in ice particle formation. Even major revisions would not help there.

On the other hand, when the authors could force themselves to restrict their study to the presentation and discussion of pure liquid clouds, their work would indeed provide valuable progress in the understanding of cloud formation in the Arctic.

So, the authors should feel encouraged to change their manuscript in this appropriate way. I do not know if this can be done at this stage of revision or needs a complete new

**ACPD**
submission. In hopes that the authors go in this direction, I made a complete review hereafter with minor general comments and specific remarks. Many points are related to the Aitken mode residuals in order to indicate in detail the large doubts of the actual interpretation, but this would then be irrelevant for a revised as requested manuscript.

Minor general comments:

1. Many different number concentrations are elaborated in this study. The authors should think about to use an abbreviation for each of them to make life easier for the reader and themselves.

2. To confirm rather important statements or analysis approaches only references are provided that do not really help the reader. Here the explicit plot in the reference should be given or a plot in the supplement would be highly appreciated. Moreover, some few studies are cited that did not really support the given statement or observation and thus needs intensively revised. These text passages are all mentioned in the "specific remarks".

3. It is totally unclear why and how the sampling efficiency of an airborne, forward looking CVI are applied to an upward looking ground-based CVI. Since this the basis for the statement that the residual particle concentration agrees rather well with the cloud particle concentration, the applicability of this approach must be justified in the manuscript, which is totally missing.

4. The order of plots in the supplement is not the same as their mention in the manuscript, which should be harmonized.

5. It is unclear whether chapter 3.1 and 3.2 are not so much part of the assessment of the GCVI sampling efficiency but rather much more part of the scientific cloud analysis, maybe except the observation that the GCVI sampling efficiency is reduced at stronger upwind velocities. The authors should think about that.

6. In Addition, it is incomprehensible why the existing ambient aerosol particle data set

**ACPD**
(out of cloud) is not used at all in the analysis. Using short periods before and after cloud appearance, and in comparison with the total aerosol particle data set (inside cloud), the quantitative functionality of the whole air inlet inside cloud could be evaluated. When this is confirmed it would be another approach to derive a GCVI sampling efficiency by scaling the large part in the residual size distribution (> 100 nm) with the total aerosol particle size distribution assuming that those particles are activated with 100%. This result could be again compared with the approaches applied already in this study.

Specific remarks

L.3: This statement is not correct. There are aerosol particles contained in cloud particles due to impaction scavenging or secondary ice formation, which are not involved in cloud formation. So the authors must be more precise and need to reword their description here.

L.4: There is a too drastic change from the research description to the used instrument. This should be done more smoothly, especially not every reader will know a "ground-based counter-flow virtual impactor inlet system" and its peculiarities.

L.6-7: That not only liquid but also mixed-phase clouds exist in the Arctic is known and not only suggested by prior work. Hence, this statement is trivial and should be reworded.

L.16: The reference should be in brackets.

L.20-22: It looks like that something is missing in this sentence. If not, the statement is not totally clear and should be formulated more clearly. What is the cause what is effect?

L.29 and L. 33: What should be the meaning of "typically"? Should be deleted.

L.30: Which scavenging is meant by the authors? Nucleation or impaction scavenging? This is an important aspect and should be mentioned.

**ACPD**
L.34: Which particle sinks are meant? The needs to be explicitly listed here for a better understanding.

L.50-51: delete "for example".

L.55: "performed" should be replaced by "carried out" or something else, since only actors "perform" on stage.

L. 56: not only number concentrations but add also number size distributions.

L.59: it is obvious that the cloud particles are ambient, so this term should be omitted in the manuscript, especially since it could come to confusion with ambient particle size distributions where the term "ambient" is needed.

L.63-64: This listing is redundant to the listing in L.58-59, so the authors should do it completely at one text passage and not two times.

L.116: According to Fig.1 and chapter 2.3.1 only 2 L/min sample flow were indeed used to take the residual particle measurements. Thus, 13 L/min sample flow were not used at all. This is a pity, because when the sample flow would be reduced to the required 2 L/min, the CVI enrichment would be substantially enhanced, which would significantly improve the measurement statistics, which is especially important in the arctic clouds with less cloud particles. The increase in enrichment would increase from about 12 (L.133) to about 90! So, the authors should explain why they did not use this option or if this it is not possible to operate the commercial GCVI with this flexibility.

L.121-122: How save is this trap? Could it be that larger crystals and graupel could break-up in this trap, so that the smaller fragments become resuspended in the system and deliver residual particles in DMPS1? The authors should comment on this and best of all provide a short text passage in the manuscript.

L.124: For the WMO fog definition a solid reference should be given. Moreover, for a reader engaged in cloud physics it would be pretty much appreciated when this value would be transferred and provided as a minimum LWC, which gives a much better

**ACPD**
impression about the definition of a cloud in this study.

L. 158-159: Why are two CPC used behind the whole air inlet to measure the total aerosol particle concentration? And the data of which of the two CPC are used in the study?

L. 164: The used particle density is the one of liquid water, but the residual particles are measured at very dry conditions, so the effective particle density is 1.7 g/cm3 or even higher. The same argument holds for the total particles, but due to an expected higher RH in the sampling line, the effective particle density might be 1.5 g/cm-3. The particle density has a rather large influence in particle loss calculations, i.e. these correction calculations has to be repeated and applied to the presented data.

L.165: What is meant by contaminations? The authors need to mention those sources of error explicitly to give the reader a better expression, which situations have been excluded from the study and which not.

L.175: The consequence of the sentence is not quite clear. Does it mean the minor measurement efficiency of DMPS-1 for small diameters was not corrected at all or only not if the concentration in the respective bins was zero. In the presented data, the residual particle size distributions obtained by DMPS-1 often show non-zero concentration down to 10 nm, so that a diffusion loss correction could be in principle applied and it would be also possible to derive a correction from Fig.S2b, which would in both cases better than no correction. Therefore, the authors should say something about this.

L. 183: From Fig.S1 it is obvious that about 2/3 of the cloud sampling was carried out at south wind and 1/3 at north wind conditions. This implies that for 2/3 of the cloud sampling the aspiration efficiency of the FM-120 was very good and for 1/3 of the cloud sampling the aspiration efficiency was pretty bad (since the cloud particles have to make a U-turn to get measured, which will be additionally modulated by the wind speed). Consequently, the first idea would be to do the analysis with the 2/3
south wind cloud cases. The authors should therefore try to go in this direction or present very good arguments, supported by a data evaluation in this respect, why the measurements from all wind directions are equally suited. The mention of only a reference (Koike et al., 2019) is definitely not sufficient.

L. 194: It would be more meaningful to write the METEK user manual explicitly into the brackets and not treat it as a reference citation.

L. 197: Why does the authors did not correct the temperature according to this non-correct temperature difference?

L. 220-222: See comment for L. 183. It should be easy to carry out a FM-120 and GCVI comparison for the 2/3 south wind and the 1/3 north wind conditions to demonstrate whether there is a wind direction or not. This is much better than to "assume" that the fog monitor provides an accurate representation of the cloud particles entering the GCVI. This should be additionally restricted to droplets, since the FM-120 will not correctly measure ice particles.

L. 222-224: The complete GCVI sampling efficiency correction procedure is a very important part of the paper, which the reader must follow more easily. Thus, the mentioned transmission efficiency has to be explicitly shown as a graph in the supplement as only provide the Shingler et al. (2012) reference, especially because the original size dependent efficiency was extended.

L. 227: How dry is the counterflow? This is not mentioned anywhere else.

L. 230: The expression "no correction is preferable to an invalid correction" is pretty unscientific. Should that be a general statement or only related to the concrete approach? If the latter is the case, the authors should somehow quantify why the error is smaller with no correction or should simply remove this expression.

L. 231: The expression "corrected cloud particle concentrations" is badly chosen and misleading. "The GCVI sampled cloud particle concentrations according to the trans-
mission correction" or something similar would make it clearer.

L. 236-238: For the data points below the 1:10 line, it would be very important to know the measured FM-120 cloud particle concentration in order to evaluate how thick the clouds were during these measurements.

L. 247-252: At least the reference of Mertes et al. (2007) is used here in a wrong manner. When there is more than one aerosol particle in or on a ice crystal they will not be emerged as single particles during the drying process in a CVI. These particles will remain on the more and more shrinking ice particle until they will lump together and are released as one particle. Only if the original ice crystal breaks-up the scavenged aerosol particles will be counted individually. This could happen in the cloud (secondary ice) or by hitting any surface of the GCVI. The difference here is that secondary ice would have counted by the FM-120 too, whereas the latter would only be seen as increased concentration in the GCVI, as a shattering artefact, like it is described here.

L. 253: Again a citation, this time Lauber et al. (2018) is used in a wrong sense, because it does not show or even treat the topic of a INP break-up, which indeed is not possible in nature.

L. 263: What is the quantity of "sometimes"? Better change "that sometimes occurs" into "that was observed".

L. 264: The acceleration and deceleration zones are not the only possible locations. Since larger cloud particles could not follow the streamlines, they will simply hit inlet surfaces and walls where the shattering occurs. This needs to be added here.

L. 265-267: Not the concentration but the size of the particles is the important parameter.

L 279-280: The precipitating particles could easily explain the difference in concentrations, since they could also easily shatter at the GCVI surfaces and their fragments are Interactive comment

entering the GCVI and their residuals are counted.

L. 284: Spiegel et al. (2012) only studied liquid droplets and not ice crystals, i.e. this citation is incorrectly used here.

L. 285-293: What is this about now? First the FM-120 was supposed to be correct in order to evaluate the GCVI sampling, now the GCVI concentration results are used to scale FM-120 concentration measurements, and this only when the first is higher than the latter, to account for a FM-120 undercounting. This is dubious and a bit helpless. The much better approach would be to separate the cases with and w/o the presence of cloud ice by an appropriate sensor to prove all these assumptions. Since the amount of ice particles in a mixed-phase cloud is much smaller than the amount of droplets the ice undercounting of the FM-120 would not have much effect.

L. 294-298: This is really a funny conclusion: I cannot really quantify the sources of error in my instruments, therefore I use all the data!

L. 303: The size-dependent transmission efficiency must be shown in the supplement or at least the exact plot in Shingler et al. (2012) must be provided.

L. 309: I would call it "for the sake of convenience".

L. 324: All particles measured behind the GCVI are residuals. Most likely the authors wanted to say "are in fact CCN".

L. 324-328: The decisive parameter for this observation at unaltered background aerosol is an increase in cloud supersaturation. This needs to be explicitly mentioned here. A decrease in particle concentration is only a possible cause, just like for example a higher updraft velocity.

L.330-333: a) there should be no uncertainty in the CVI enrichment factor. If so, the authors need to explain this in the manuscript here or before when the working principle of the GCVI is explained. b) The uncertainty of the sampling efficiency would explain a level of the ratio below or above 1 but could not explain the variations of the ratio.
Maybe the counting statistics is not sufficient. The authors, should check for the real reasons for these variations.

L. 339-340: This statement is not true, because when the same size modes would be present in the whole air inlet there would be no peak at 20 nm in Fig.3c. So, Fig.S7 is not a good proof for the given statement, in addition because Fig.S7a also shows this peak.

L.340-341: The statement in L.339-340 (beside the point that it is not really the truth) is not an argument that there are no reasons to expect droplet and even more ice crystal shattering here.

L.343-362: It is totally surprising that there are no residual particle size distributions and ratios residual/total present that show this "ice present effect" of small residuals observed in chapter 3.1. Why is that? These "cold cases" should also be found when the size distributions are sorted by updraft. An explanation by the authors is definitely needed here.

L. 367: Why is no 25th percentile given here, like it is two lines before for the residual number concentration?

L. 375-376: This sentence is not comprehensible. What is meant by "in terms of when peak saturation occurs"? And what is meant by "some differences to previous measurements" in detail?

L.381: The residual particle annual cycle cannot be "confirmed" by the cloud particle measurements. But one could say, that it is "closely related to the cloud particle annual cycle.

L. 393: It is not convincingly proven that this "clear seasonality" is not caused by the artefacts of ice particle shattering. Therefore, I would not use this expression here.

L. 430: What is meant with "total number concentration" in Fig.S11?
L. 436-440: This argumentation supports to some extend the option of large cloud particle shattering as an explanation for the occurrence of the Aitken mode residuals. It is really a pity that no ice particle sensor was operated to bring more light in this problem.

L. 441: Do the authors indeed mean "ambient" particle size distributions (w/o cloud) or is it more likely "total" particle size distributions (within cloud)?

L. 434-448: In contrast to the discussion in the manuscript both Cluster 1 and 2 are most likely an indication for an artefact sampling due to ice particle and large droplet shattering. Cluster 1 are very small Aitken mode particles, but related to a broad size range of cloud particles. The larger cloud particles most likely experience shattering at the GCVI and the very small ones would release only one residual. Maybe this very small fraction might be due to secondary ice formation, but this is not further discussed in detail in this study.

The flat shape of Cluster 2 is a clear indication of large cloud particle shattering, realistically indicating that in this way all particle sizes between 20 and 200 nm occur with more or less the same frequency. At least there is no other mechanism known that could account for such a shape. The existence of artefact measurement by shattering is strongly supported by the fact that this cluster is related to the largest cloud particles measured.

The connection of both clusters with low temperatures and the existence of ice particles further support the occurrence of ice particle shattering artefacts.

L. 464-466: It is very good to point out the difference of CCN and residual particle measurements, but this make only sense when the residuals of liquid clouds are subject of the discussion. Once ice particles occur there is of course a difference since no INPs can be measured in a CCN counter.

L. 467: it is clearer to write "long-term CCN data sets".
L.479-482: Despite all the caution the authors has exercised in the data analysis by leaving it open whether the occurrence of Aitken mode is an artefact or not, they now claim that the observed Aitken mode particles play an important role for Arctic clouds. But the data situation does not allow for this.

L.486-490: This explanation is incomplete. It explains why smaller particles could become activated, but it does not mention that still the larger ones become activated first. Thus, the accumulation mode particles should be seen much more pronounced, especially because they are activated with 100% and, the 20-30 nm particles only with few % (e.g. Schwarzenboeck et al., 2000).

L. 495-499: Again, the argumentation her is only partly true. Findings from mixedphase and liquid clouds are mentioned in one sentence, so it is not clear to which cloud type the authors attribute the small Aitken mode particles to. Mertes et al. (2007) indeed measured similar residual size distribution in mixed-phase clouds but with an ice selective CVI, which is not the case in this study. There should be a difference in the measurements since the droplet residuals should be involved here but not are not in the cited study. Consequently, the number of residuals is much less in Mertes et al. (2007), because the CVI was designed to remove large cloud particles that are mostly responsible for artefact measurements. Moreover, the activated fraction showed a strong increase with size, whereas in this study this is not seen (cf. Fig.S7a, Fig.3c). Schwarzenboeck et al. (2000) found droplet activation down to 25 nm particles but at the same time an increase of the activation fraction to 1 above 100 nm, which is totally different to the Aitken mode particle observation in this study and is thus not a confirmation of the findings here.

L. 500-507: It is interesting that the authors do not find an explanation for the occurrence of Aitken mode particles in the Arctic winter. So how should these particles then play an important role in wintertime cloud formation?

L. 508-513: Secondary ice might be possible and would create small residuals, but

**ACPD**
the number concentration of ice particles with respect to cloud droplets in mixed-phase clouds is still very low (at least 1 order of magnitude), so that these residuals would not dominate the size distribution. And the Fm-120 would see these particles too, which seemed not to be the case as it was pointed out by the authors. Once more, only CCN but not the insoluble INPs would undergo fragmentation.

L.513-524: Many speculations which are not supported by additional measurements that would have been essential for such a study or using one of the ice selective CVI inlets.

L. 525-528: What are the conditions that are denoted as GCVI malfunctioning and how are they recognized?

L. 529-530: A more scientific procedure would be to remove all data points that could have been biased by potential artefacts and use the "unsuspicious" measurements/results only. Much stronger and first of all justified statements would have been possible.

L.533: Even if one would assume that the small residuals stem from non-shattered cloud particles, the authors always leave it open if these residuals are released from droplets or ice particles. If they are originating from secondary ice (which is to some extend the most likely explanation), these residuals do definitely not contribute to the formation of mixed-phase clouds. The used expression is even in this case a total exaggeration.

L. 538-539: This is a trivial statement that is valid for all clouds around the globe and could be therefore deleted.

L. 540: Before in this study, the separation of Aitken mode and accumulation mode was defined at a diameter of 60 nm, now the authors introduce a size of 100 nm as the lower limit for accumulation mode particles. This is not consistent and has to be harmonized. Since Cluster 1 and 2 are more suspected to present artefact sampling,

ACPD
the remaining Clusters 3,4,5 are accumulation mode particles according to the first definition, although they are closely to the left edge of the size distribution of the accumulation mode. Hence, it might be advantageous to speak of particles smaller 100 nm instead of declaring that these are Aitken or accumulation mode particles.

L. 543: It puzzles me that the authors claim that the may see the features of a CCN limited regime in their dark period data. Looking at Fig.S7a or Fig.4b it is obvious that the are many particles present in the total aerosol (interstitial + residual) that are not activated. But CCN limited means that there are no more or very few particles to activate, i.e. that there are hardly any interstitial particles left, which is not the case here.

L.544-547: This description is a possible scenario but certainly not a conclusion of this study. Thus, it should be removed or considerably reworded to bring it in the right context to the own conclusions.

Fig.2: Here mean and median are the solid and dotted lines whereas it is vice versa in all other figures. This should be made consistent.

Fig.2: according to the text it is not the measured FM-120 concentration but the GCVI sampled cloud particle concentration derived from the GCVI transmission efficiency, correct? At least that is the explanation in the corresponding text. If so, the figure caption needs to be corrected

Fig. 3 and Fig.4: brackets are use in this way "(]". In case this has no special meaning, this should be made consistent. Furthermore: The mean, median, percentiles are not mentioned in the figure caption which should be added.

Fig.S4: figure caption: meet instead of meed.

Figures in supplement: Several times the straight lines are named "full", whereas "solid" is the more common term and used in all other figures (supplement and manuscript). This should be made consistent.

---

## Referee Comment (RC2) · Anonymous Referee #1 · 4 Jul 2020

This manuscript presents a set of multi-year measurements of total particle size distributions and cloud residual size distributions at Zeppelin Observatory on Svalbard. This is an impressive and important data set from the Arctic region, with the potential to help better constrain our understanding of aerosol-cloud interactions in Arctic regions. The authors observe sub-100nm (Aitken mode) cloud residuals with some frequency, particularly at cold temperatures in the poorly characterized winter season, and make the claim that these Aitken mode particles play an important role as cloud nuclei in Arctic regions in the coldest seasons.

This is a perplexing and somewhat intriguing result; however, two very major issues arise with this manuscript. First, the impact of measurement artifacts cannot be dismissed in the work. The potential of cloud particle shattering leading to spurious results is discussed in the manuscript (e.g., L251-252, 254-255), and then is almost entirely discounted as a driving factor for the observations of the sub-100nm cloud residuals.

[Figure]

Given the significant uncertainties in these observations, the authors overstate implications of their observations (e.g., L7-9 in the abstract). Second, the manuscript focuses almost exclusively on the small cloud residuals at the expense of other observations, which are also surely valuable and are not given much interpretation. These two issues are elaborated further in the major and specific comments below.

It is clear that while the authors have thought in depth about the possible impact of CVI measurement artifacts, they have not been able to come to any strong conclusions about their impact, and ultimately make the choice to keep all their data in the analysis. My overall suggestion for this manuscript is for the authors to reconsider their focus, and to remove or soften their assertion that sub-100nm particles are important CCN and INP in Arctic winter. This could be accomplished by broadening the scope of the analysis, and particularly the interpretation, to better highlight their observations throughout the year. The authors could take an approach where they first include only data in which they have the highest confidence, and discuss what is learned about aerosol-cloud interactions from those data (i.e., mostly the data collected at warmer temperatures when ice crystal shattering may be less of an issue). The authors could then include the entirety of their data set, in a separate discussion where they lay out the evidence for and against these sub-100nm cloud residuals truly representing the cloud nuclei distribution, making it extremely clear that they cannot rule out measurement artifacts, and providing motivation for future measurements.

Major Comments:

1. The authors provide considerable evidence that their cloud residual Clusters 1 and 2 are associated with ice processes, and use this to suggest that very small particles may be somehow driving ice nucleation. This evidence includes: (1) occurrence at colder temperatures in January and February, (2) high ice occurrence from Cloudnet, (3) association with larger cloud particles, (4) association with times when the cloud residual and cloud particle measurements did not agree well. While all of these are indeed evidence for the presence of ice, they are also evidence for increasing importance

of ice crystal shattering in the CVI, which is a well known issue with this type of cloud residual measurement. Indeed, these measurement artifacts are a partial motivation for developing ice selective inlets (e.g., https://www.atmos-meas-tech.net/8/3087/2015/ and https://www.atmos-meas-tech.net/9/3817/2016/). Further, Cluster 2 is the most frequent, but it shows the most resemblance to a residual distribution you would expect from shattering i.e., nearly uniform across all sizes, bearing little resemblance to the total particle distribution. For this reason, I strongly suggest (as described above) that the authors re-consider the scope of their manuscript to not focus entirely on these smallest cloud residuals.

2. The authors explain their observation of Aitken mode cloud residuals with the possibility of secondary ice formation. However, two issues arise with this interpretation. First, the number of supercooled liquid droplets in a mixed phase cloud should far exceed the number of ice crystals. So, in addition to the small residuals from secondary ice formation there should also be accumulation mode residuals present from supercooled droplets. This does not appear to be the case. Second, INPs are generally not thought to be soluble and are generally larger than a micron (e.g.: https://www.atmos-chem-phys.net/16/1637/2016/ which includes some Arctic data) , and so INP material is unlikely to become fully distributed among secondary ice particles. Another possibility is coagulation scavenging of Aitken mode particles with ice crystals, which could lead to the observed residual size distributions upon shattering and/or evaporation in the CVI. Given these possibilities, the conclusion that Aitken mode particles driven cloud formation in Arctic winter is not supported.

3. An intriguing observation is shown in Figure S7: the comparison between clustered cloud residual size distributions and the total particle size distributions. These demonstrate that when the smallest cloud residuals are present, the total particle size distribution resembles that of the cloud residuals, particularly for for cluster 1. This could be construed as evidence in favour of the author's hypothesis. But, are the total particle sizes measured coming from interstitial particles? i.e., those measured within

a cloud? If so, are these valid particle size distributions, or are they impacted in some way by sampling of cloud particles into the whole air inlet? Do the authors get the same results if they sort the before-cloud and after-cloud size distributions based on the cloud residual clusters? If this result is robust, then I would expect something comparable. Overall, the out of cloud size distributions should be incorporated into this analysis to lend potential support to the conclusions. Further, the data shown in Figure S7 should be shown in the main paper, perhaps combined in some way with Figure 8.

4. Throughout the manuscript the authors appear to confuse the concepts of cloud residuals and cloud nuclei (e.g., L3: "cloud residuals, i.e. particles that were involved in cloud formation", L49-50, the paragraph beginning at L464, L532-535). Cloud residuals are a combination of cloud nuclei, particles that have been effectively scavenged in cloud by droplets and ice crystals, and scavenged particles or cloud nuclei that have been chemically processed within cloud. The one-to-one connection between residuals and nuclei cannot be made. This has direct bearing on the way in which the authors interpret their results.

5. Related to (4) above: the authors also at times appear to misconstrue CCN and INP, which may come from the fact that they cannot distinguish the type of cloud particle they measure. This issue is most prevalent in the interpretation of the results. For example, at L479-480, the authors cite previous studies that have shown small particles can be important CCN in Arctic regions (i.e., when particle numbers are low and supersaturations are high). Given that their observations of small cloud residuals occur in winter when ice processes are important or even potentially dominant, it is unclear how these prior studies directly support their conclusions. Later at lines 486-490 the authors discuss the concept of a CCN-limited cloud-aerosol regime, which is not related to ice formation, and was originally proposed using summertime Arctic observations. While the authors acknowledge this fact in the following sentence, it is not entirely clear how this discussion of the prior literature supports their observations.

6. The authors make frequent assumptions in their analysis that do not appear to have

been arrived at in a quantitative manner. For example on line 230 stating "no correction is preferable to an invalid correction," and the discussion around keeping all data in the analysis at L294-298. Some quantitative assessment of the uncertainty introduced in a correction, versus leaving out the correction, or the impact on the data interpretation when keeping a removing certain suspicious sections of data should be made to build a logical argument for making such decisions. I acknowledge here that making this type of measurement and accounting for all errors is very challenging, and at the same time I hope that the authors will consider this comment thoughtfully when refocusing this manuscript.

7. The monthly average total particle size distributions (orange curves in Figure S8) appear to have a larger dominance of Aitken mode particles in January and February than have been observed in previous multi-year measurements in Zeppelin (e.g., https://www.atmos-chem-phys.net/16/3665/2016/ and https://www.atmos-chem-phys.net/17/8101/2017/). Do the out of cloud, or just before cloud, total particle size distributions look the same as these on a monthly basis? If not, this would suggest that using total particle size distributions measured during cloud events may not be representative of the actual ambient particle populations. How is this observation impacted by the particle loss corrections and density assumptions?

8. Section 2.4: What metrics were used to select the appropriate and physically meaningful number of clusters? Mean euclidean distance? Or any other objective way of looking at optimizing cluster number to explain variability in the data with the smallest number of possible clusters? Four clusters groups the two distributions that contain small particles — what evidence is there that these are physically distinct clusters? How was the total particle data incorporated? i.e., the grey size distributions in Figure S7, were they grouped based on the cloud residual clusters? Or clustered separately? If you cluster the out of cloud (I.e., before or after cloud) particle size distributions on their own, do the same five clusters come out?

9. The abstract and conclusions sections contain several statements that are not direct

conclusions from this study. In particular, the conclusions at L 545-547 is not something that can be concluded from observations in this study.

Specific Comments:

L83-93: Observations from satellite are also relevant here, e.g.,: https://agupubs.onlinelibrary.wiley.com/doi/full/10.1029/2012GL053385

L135-140: Is the CVI transmission experimentally determined here explicitly? If so, it should be shown as a function of size in the SI.

L145-179: How well do DMPS 2a and 2b agree in their overlapping size range? How sensitive is the particle loss calculation to the chosen density? A density of 1g/cm3 is likely much lower than the true value.

L165: "Manual screen for outliers and contamination" should be elaborated

L262-267: Would these very low particle concentrations not be consistent with ice clouds in winter? Also, it should not be the absolute amount of particles that matters, but the difference between the two measurements, which is up to 2 orders of magnitude.

L285-293: This appears to be circular logic, and the inability to distinguish between droplets and ice crystals seems to cause a lot of problems here.

L324-327: This could be the case, or could be indicative of a change in supersaturation.

L330-332: Do you expect values up to 2? Does this not give further evidence of shattering, especially at smaller particle sizes? It would be best to include shading around these means to show the range of uncertainty

Figures 3a and 4a: what do the error bars represent?

Figure 4c: why do the authors think that the data for small cloud residuals is not apparent in this analysis?

Figure 6, 7 and 8: These figures are showing a lot of the same information in different ways. This could be focused in such a way to make clear what is most important for the reader to see.

---

## Author Comment (AC1) · 8 Dec 2020

**Reply to reviewers of the manuscript "The role of nanoparticles in Arctic cloud formation"**

Karlsson et al.

December 8, 2020

We thank both reviewers for their very detailed and constructive comments. Both reviewers had concerns about the potential influence of measurement artefacts and both suggested that we should restructure our manuscript with a more pronounced focus on the actual cloud phase (liquid vs. mixed-phase). Overall, we agree with these major remarks and have therefore substantially restructured our manuscript (mainly the result and discussion part) and added additional analysis and text. In summary, these major changes include:

1. We have separated the analysis and the presentation of the results between liquid and mixed-phase clouds. This separation is done in the result part for both (a) the GCVI sampling efficiency and (b) the presentation of the two year data set of cloud residual size distribution measurements. Figures and text were adapted accordingly.

2. We now give detailed information on the GCVI and DMPS data treatment (new subsection in the method part) and provide further technical specifications in the main manuscript as well as in the supplementary information (SI).

3. We have added additional analysis with regard to the CVI sampling efficiency by comparing the accumulation mode concentrations of residual and total particle size distribution for liquid clouds, as suggested by the reviewer. The result of a sampling efficiency of around 0.5 is similar to the value determined from independent cloud particle measurements, which further supports our findings and the overall reliability of our data.

4. We have toned down the interpretation and implications with regards to the sub-100nm-particles, which was also needed after the restructuring and addition of new analysis. We now give more weight to the entire set of observations. We have also changed the title of our manuscript to a more general title ("A long-term study of cloud residuals from low-level Arctic clouds") describing the actual research performed.

5. Related to the point above, we are now more cautious in our conclusions with regards to the influence of sampling artefacts as well. Further discussions among the coauthors, sparked by comments from both reviewers, have led us to believe that a small amount of the Aitken mode cloud residuals in the winter may be the result of ice or snowflake shattering artefacts. Additional analysis and a more balanced discussion about this and other potential causes has been added in the revised manuscript.

Reviewer 2 insisted on entirely removing the data with the potential influence of mixed-phase clouds. Although the reviewer suggests that these data points are suspicious, we believe we

have shown that there is still a lot that can be learned from them. As outlined above, we have chosen to present the results for liquid and mixed-phase clouds separately instead of discarding any data. We believe that the new structure makes it clear to the reader which data points are likely to be affected by artefacts and which are not, and that the toning down of the conclusions relating to those data points now makes the paper well balanced, thus resolving the issues raised by the reviewer. As suggested by both reviewers, we tested the approach of excluding data where no agreement between the residual and cloud particle number concentration was found. Fortunately, the overall results did not change, which gave further confidence in the validity of our results.

We see no clear evidence of artefacts originating from the sampling of ice crystals inside inside the CVI but we do discuss the potential break-up of ice crystals before the CVI (artefacts generated within the wind tunnel) or in the atmosphere (secondary ice). By using the cluster analysis, we directly assess the contribution of residual size distributions that do not follow the expected classical behaviour for liquid droplet activation and by doing so, we deliberately investigate the overall contribution, the temporal evolution and the trustworthiness of those ice-influenced size distributions. We would also like to emphasise that similar cloud residual size distributions have been observed before using similar techniques (see e.g., Seifert et al., 2003, although for cirrus clouds) or (Mertes et al., 2007, using an ice-CVI), thus there is no reason to fully remove this data. One important addition to the revised manuscript relates to the findings of Verheggen et al. (2007), a study on mixed-phase clouds at the high-alpine site Jungfraujoch. Although Verheggen et al. (2007) used a slightly different inlet system set-up, the temperature-dependence of their activation ratios agrees surprisingly well with our data from the Arctic, giving further confidence in the overall validity of our work.

Further, we would like to stress that the same type of CVI has been previously thoroughly evaluated (Shingler et al., 2012). The CVI has been used extensively by other research groups on various aircraft campaigns (e.g., Modini et al., 2015; Sanchez et al., 2016; Hossein Mardi et al., 2019). Since the CVI was installed within a wind tunnel, we added information on how its performance within the wind tunnel (GCVI) was evaluated by the manufacturer. In addition, we would like to mention that the same GCVI has been successfully deployed within short-term campaigns by our and other research groups and, like for the aircraft studies, no major artefact production has been observed within the sampling line of the CVI (Zhang et al., 2017; Lin et al., 2019a,b; Graham et al., 2020; Baccarini et al., 2020).

Summarising, we are convinced that by adding additional analysis, information and discussion and by restructuring the result section, our manuscript has significantly improved in quality. We thank the reviewers again for their great effort and patience.

We will provide more detailed replies to their comments below. Our comments are given in blue, new text within the revised manuscript is given in light blue. Because the revisions are so substantial, we have not been able to include every single minor change in this document. We have included the main ones in response to the relevant comments, and refer to the diff document for the complete set of changes. We have not included the new figures within this reply letter but rather refer to the revised manuscript and the SI.

**1    Anonymous Referee #1**

This manuscript presents a set of multi-year measurements of total particle size distributions

and cloud residual size distributions at Zeppelin Observatory on Svalbard. This is an impressive and important data set from the Arctic region, with the potential to help better constrain our understanding of aerosol-cloud interactions in Arctic regions. The authors observe sub-100nm (Aitken mode) cloud residuals with some frequency, particularly at cold temperatures in the poorly characterized winter season, and make the claim that these Aitken mode particles play an important role as cloud nuclei in Arctic regions in the coldest seasons.

This is a perplexing and somewhat intriguing result; however, two very major issues arise with this manuscript. First, the impact of measurement artifacts cannot be dismissed in the work. The potential of cloud particle shattering leading to spurious results is discussed in the manuscript (e.g., L251-252, 254-255), and then is almost entirely discounted as a driving factor for the observations of the sub-100nm cloud residuals.

Given the significant uncertainties in these observations, the authors overstate implications of their observations (e.g., L7-9 in the abstract). Second, the manuscript focuses almost exclusively on the small cloud residuals at the expense of other observations, which are also surely valuable and are not given much interpretation. These two issues are elaborated further in the major and specific comments below.

It is clear that while the authors have thought in depth about the possible impact of CVI measurement artifacts, they have not been able to come to any strong conclusions about their impact, and ultimately make the choice to keep all their data in the analysis. My overall suggestion for this manuscript is for the authors to reconsider their focus, and to remove or soften their assertion that sub-100nm particles are important CCN and INP in Arctic winter. This could be accomplished by broadening the scope of the analysis, and particularly the interpretation, to better highlight their observations throughout the year. The authors could take an approach where they first include only data in which they have the highest confidence, and discuss what is learned about aerosol-cloud interactions from those data (i.e., mostly the data collected at warmer temperatures when ice crystal shattering may be less of an issue). The authors could then include the entirety of their data set, in a separate discussion where they lay out the evidence for and against these sub-100nm cloud residuals truly representing the cloud nuclei distribution, making it extremely clear that they cannot rule out measurement artifacts, and providing motivation for future measurements.

We thank reviewer 1 for their detailed and helpful comments. We have followed the advice to start the discussion of our findings with the data of liquid clouds and finishing the discussion for mixed-phase clouds. We emphasise that artefacts cannot be ruled out, and have expanded the discussion (including additional figures) to try to quantify when artefacts are most likely, when they are less likely, and what bearing they have on the results. Further details are given below. This is the new structure of the manuscript (new headings/headings with changed names are in cyan instead of blue):

1 Introduction

2 Methods

  2.1 Site description

  2.2 Inlet systems

    2.2.1 Whole-air inlet

    2.2.2 Ground-based counterflow virtual impactor inlet

  2.3 Instrumentation

**1.1 Major Comments:**

**1.** The authors provide considerable evidence that their cloud residual Clusters 1 and 2 are associated with ice processes, and use this to suggest that very small particles may be somehow driving ice nucleation. This evidence includes: (1) occurrence at colder temperatures in January and February, (2) high ice occurrence from Cloudnet, (3) association with larger cloud particles, (4) association with times when the cloud residual and cloud particle measurements did not agree well. While all of these are indeed evidence for the presence of ice, they are also evidence for increasing importance of ice crystal shattering in the CVI, which is a well known issue with this type of cloud residual measurement. Indeed, these measurement artifacts are a partial motivation for developing ice selective inlets (e.g., https://www.atmos-meas-tech.net/8/3087/2015/ and https://www.atmos-meas-tech.net/9/3817/2016/). Further, Cluster 2 is the most frequent, but it shows the most resemblance to a residual distribution you would expect from shattering i.e., nearly uniform across all sizes, bearing little resemblance to the total particle distribution. For this reason, I strongly suggest (as described above) that the authors re-consider the scope of their manuscript to not focus entirely on these smallest cloud residuals.

We fully agree with this comment. We have restructured our manuscript and give more weight to the overall findings for all cloud types (liquid vs. mixed-phase/ice). We have expanded the discussion and include the possibility that clusters 1 and 2 are affected by artefacts, clearly stating that this cannot be ruled out. However, in contrast to what the reviewer suggests, we believe Cluster 2 is influenced to a much lesser extent than Cluster 1.

Further discussions among the coauthors, sparked by comments from both reviewers, have led us to believe that at least part of the Aitken mode cloud residuals in the winter may be the result of snowflake shattering artefacts, and this discussion has been added in the updated version of the manuscript. We suspect snow because particles need to reach a certain size for shattering to be likely, and this size ($\sim 70\,\mu$m) is far larger than the main mode seen in the FM-120. We

have added this in Sect. 3.1.3: "At air speeds of $100 \, \text{ms}^{-1}$, the critical diameter above which droplets may shatter into fragments is $\sim 76 \, \mu\text{m}$ (Twohy et al., 2003). Precipitating particles can exceed this size, and could thus produce fragments that are sampled if they are larger than the aerodynamic cut-size of the GCVI. However, it is likely that many of the fragments would not be aligned with the streamlines and therefore would not enter the GCVI."

There may be a few cloud droplets or ice crystals that are large enough to shatter but, as both reviewers have pointed out, such particles should be far outnumbered by the main droplet mode. Thus, they would have to be very large indeed to shatter into enough fragments to significantly affect the measured cloud residual distributions (bearing in mind that fragments still need to exceed the GCVI cut-size and be aligned with the streamlines in order to be sampled). We have also modified/added this in Sect. 3.1.3: "Droplet or ice crystal shattering is another potential source of small particles. Shattering could either happen in the wind tunnel or after the stagnation plane within the CVI inlet, and this could also cause an overestimation of the cloud residual number concentration. If the particles were to shatter after the stagnation plane, this should be clearly seen as spikes in the cloud residual concentrations measured by the total CPC, and this was not observed. Regarding shattering in the wind tunnel, as stated above, the cloud particles need to exceed a certain size for this process to be likely. While this can happen when there is precipitation, it needs to be borne in mind that in most non-precipitating clouds, the concentration of large cloud particles is much lower than the total cloud particle concentration, and therefore particles that do shatter may need to far exceed the critical break-up diameter to produce enough fragments to significantly increase the measured cloud residual concentration (Twohy et al., 2003). If the concentration differences we observe are caused by shattering artefacts, then, the magnitude of the difference suggests that precipitating particles (e.g. snow) are a more likely cause than large cloud droplets or ice crystals."

Cluster 1, which is linked to cold temperatures and has no clear droplet mode, is therefore likely to be influenced by snow shattering artefacts. Cluster 2, on the other hand, has a clear droplet mode and has a temperature distribution that is not appreciably different from those of clusters 3–5. In addition, its shape is not so much flat as bimodal with two broad modes. Within the new Fig. 11 one can observe that the average residual number size distribution consists of two broad modes (roughly around 30 and 150 nm), which are also present in the concurrent total size distribution measured behind the whole-air inlet. However, it is possible that the influence of artefacts gives the size distribution a flatter appearance — when the cluster analysis is repeated without the most suspicious data points (new supplement Fig. S12) the bimodality can be seen more clearly, which supports this hypothesis. We have added a supplementary figure with the cloud residual vs cloud particle concentration comparison separated by cluster to help quantify the influence of artefacts/suspicious data, and have added this information to the discussion about the clusters as well: "Large crystals are expected to be more prone to shattering, and indeed clusters 1 and 2 are related to larger cloud particles than the other clusters (Fig. 9a). Cloudnet does not distinguish between cloud ice and precipitating ice, so we could also be dealing with snowflakes. The average cloud particle size distribution associated with cluster 1 is rather flat with no obvious mode (Fig. 9a), and it is also associated with very low cloud particle concentrations (Fig. 10d). This could just be noisy measurements in the fog monitor during snowfall, and would indicate that cluster 1 is influenced by snow. Cluster 2, on the other hand, has a clear cloud particle size mode, although at a larger diameter compared to cluster 3–5 (Fig. 9a), and the cloud residuals are therefore much less likely to stem solely from precipitation.

The median cloud residual concentration is slightly larger than the median total particle concentration for cluster 1 around 20 nm (Fig. 12a), which would suggest that there is a risk of crystal shattering artefacts. As shown above, Figure 2b revealed two groups of data, where one showed

a discrepancy between measured cloud residual and cloud particle concentrations as would be expected with this type of artefact. The same figure separated by cluster (Fig. S9) shows that this group of data is overrepresented in clusters 1 and 2, which speaks in favour of the crystal shattering hypothesis as well. However, cloud residual size distributions with modal diameters similar to those of clusters 1 and 2 still appear in a cluster analysis where all the data outside the 10:1 and 1:10 lines in Fig. 2b are excluded (Fig S12a; note, results do not change if we are even stricter, i.e. within 1:2 and 2:1). Also note that a cloud particle mode then appears for the small particle cluster (Fig S12a), indicating a decreased relative influence from snow. This suggests that while ice crystal shattering is certainly a possibility, it is not necessarily the only explanation for the shape of the size distributions we observe."

After some paragraphs about secondary ice (see response to next comment), we then conclude the discussion of clusters 1 and 2 with the following: "In summary, it seems likely that cluster 1 is significantly influenced by snow and ice. It is difficult to say to what extent the signal is caused by crystal shattering artefacts as compared to other processes, but cold temperature outliers (cf. Sect. 3.1.3) make up roughly 45% of cluster 1 (cf. Fig. S9a). These data should be treated with caution, but there are some plausible physical explanations for the presence of small particles when the agreement with the fog monitor is better, e.g. secondary ice processes, yet further measurements would be needed to verify this. The possibility that such processes would show a signal similar to shattering artefacts is an important consideration when analysing GCVI data from ice or mixed-phase cloud conditions. However, from an aerosol activation perspective, it is irrelevant whether the snow or ice crystals shatter before or after they enter the inlet – in both cases, the resulting cloud residuals do not represent cloud nuclei.

Cluster 2 is also Aitken mode dominated, and occurs throughout the year (27% of the time, or 13% of the time if we only consider $T > 0°C$). Unlike cluster 1, cluster 2 is much less likely to be affected by snow artefacts. While the exact contribution is difficult to quantify, cold temperature outliers only make up about 13% of cluster 2 (cf. Fig. S9b), i.e. there is a significantly better agreement with the fog monitor than for cluster 1. Cluster 2 is also different from clusters 3–5 but, in contrast to cluster 1, it was observed more homogeneously throughout the year. In further contrast to cluster 1, the meteorological parameters related to cluster 2 are not distinctly different from those related to clusters 3–5 (cf. Fig. 10). This means that cluster 2 was also observed during sampling conditions when we can safely rule out the influence of mixed-phase clouds and ice crystals. Many of the caveats listed above related to cluster 1 thus do not apply to cluster 2 to the same extent. In addition, cluster 2 does not show the lack of accumulation mode particle activation that complicated the interpretation of cluster 1. Hence, the Aitken mode cloud residuals in cluster 2 very likely contain activated aerosol particles. Similar findings were reported in previous CVI measurements (Schwarzenboeck et al., 2000), although not in the Arctic. In the Arctic, activation of Aitken mode aerosol particles has been shown by indirect means and model studies (e.g., Leaitch et al., 2016; Koike et al., 2019; Korhonen et al., 2008). "

We hope that this, together with the overall refocusing of the manuscript that gives all the data more equal weight, has balanced the paper in a satisfactory way.

**2.** The authors explain their observation of Aitken mode cloud residuals with the possibility of secondary ice formation. However, two issues arise with this interpretation. First, the number of supercooled liquid droplets in a mixed phase cloud should far exceed the number of ice crystals. So, in addition to the small residuals from secondary ice formation there should also be accumulation mode residuals present from supercooled droplets. This does not appear to be the case. Second, INPs are generally not thought to be soluble and are generally larger than

a micron (e.g.: https://www.atmoschem-phys.net/16/1637/2016/ which includes some Arctic data) , and so INP material is unlikely to become fully distributed among secondary ice particles. Another possibility is coagulation scavenging of Aitken mode particles with ice crystals, which could lead to the observed residual size distributions upon shattering and/or evaporation in the CVI. Given these possibilities, the conclusion that Aitken mode particles driven cloud formation in Arctic winter is not supported.

We agree. We have added the caveats concerning the secondary ice hypothesis to the discussion of the cluster analysis. We have also expanded the discussion about artefacts; please refer to the previous comment for more details. As the reviewer correctly points out, residuals that are the result of scavenged particles, secondary ice processes or shattering artefacts cannot be said to be driving cloud formation since the particles were not acting as cloud nuclei. We have clarified this point in the discussion about the clusters as well: "At this point, it is important to point out that, even barring artefacts, a cloud residual does not necessarily respond directly to a CCN or INP. Cloud residuals can also be nuclei that have undergone processing inside the cloud (be that chemical or physical), and can contain material from e.g. riming or aerosol particles that have been scavenged by the cloud particles. Unfortunately, we have no way of distinguishing between these particle types, especially since the FM-120 cannot differentiate between cloud droplets and ice crystals. It could potentially be that the crystals we measure are the result of secondary ice formation processes (Field et al., 2016), which has been suggested in a model study to be important for Arctic stratocumulus clouds (Sotiropoulou et al., 2020). In other words, the cloud residuals we measure may be fragments of nuclei and/or scavenged particles, which would explain their small size.

Since secondary ice formation happens before the cloud particles enter our inlets, these particles should also be seen by the fog monitor. This is often not the case for cluster 1, as seen in Fig. S9, which means that secondary ice particles cannot be the only reason for the small residuals we observe (unless the ice crystals are undersampled by the fog monitor, see Sect. 3.1.3). Cloud observations at mountain-top stations such as Zeppelin Observatory may also be influenced by surface processes (e.g. blowing snow) that could increase the ice crystal concentrations (Beck et al., 2018), but this, too, should be seen by the fog monitor and as discussed earlier, no clear dependency on wind-speed has been observed. As stated earlier, it is not possible to translate the cloud residual data to CCN, INP, etc without further detailed information on cloud phase, structure and origin. The cloud phase is an important parameter and should as such be added in future studies. "

**3.** An intriguing observation is shown in Figure S7: the comparison between clustered cloud residual size distributions and the total particle size distributions. These demonstrate that when the smallest cloud residuals are present, the total particle size distribution resembles that of the cloud residuals, particularly for for cluster 1. This could be construed as evidence in favour of the authors hypothesis. But, are the total particle sizes measured coming from interstitial particles? i.e., those measured within a cloud? If so, are these valid particle size distributions, or are they impacted in some way by sampling of cloud particles into the whole air inlet? Do the authors get the same results if they sort the before-cloud and after-cloud size distributions based on the cloud residual clusters? If this result is robust, then I would expect something comparable. Overall, the out of cloud size distributions should be incorporated into this analysis to lend potential support to the conclusions. Further, the data shown in Figure S7 should be shown in the main paper, perhaps combined in some way with Figure 8.

The total particle number size distributions in Fig. S7 were measured behind the whole-air inlet

during the cloud event (measured concurrently with the cloud residual data), so it comes from both cloud particles and interstitial particles. As suggested by the reviewer, we have moved an updated version of this figure to the main part of the manuscript. The appearance or termination of the cloud event can also be caused by a change in air mass with differences in aerosol properties. As such, we refrain from an additional analysis of particle size distributions before and after the cloud event (cloud residual and whole-air) since we believe that the two-year (concurrent) dataset stands for itself.

**4.** Throughout the manuscript the authors appear to confuse the concepts of cloud residuals and cloud nuclei (e.g., L3: "cloud residuals, i.e. particles that were involved in cloud formation", L49-50, the paragraph beginning at L464, L532-535). Cloud residuals are a combination of cloud nuclei, particles that have been effectively scavenged in cloud by droplets and ice crystals, and scavenged particles or cloud nuclei that have been chemically processed within cloud. The one-to-one connection between residuals and nuclei cannot be made. This has direct bearing on the way in which the authors interpret their results.

We agree that the terms "cloud residual", "CCN", and "INP" were (unintentionally) somewhat muddled in the previous version of the manuscript. We have gone through the manuscript and done our best to correct this issue, as well as adding clarifying statements (see response to comment 2 above).

**5.** Related to (4) above: the authors also at times appear to misconstrue CCN and INP, which may come from the fact that they cannot distinguish the type of cloud particle they measure. This issue is most prevalent in the interpretation of the results. For example, at L479-480, the authors cite previous studies that have shown small particles can be important CCN in Arctic regions (i.e., when particle numbers are low and supersaturations are high). Given that their observations of small cloud residuals occur in winter when ice processes are important or even potentially dominant, it is unclear how these prior studies directly support their conclusions. Later at lines 486-490 the authors discuss the concept of a CCN-limited cloud-aerosol regime, which is not related to ice formation, and was originally proposed using summertime Arctic observations. While the authors acknowledge this fact in the following sentence, it is not entirely clear how this discussion of the prior literature supports their observations.

We have worked to fix this issue (see response to previous comment). We also believe that the new subdivision of the manuscript where cold and warm clouds are treated separately helps to avoid any confusion.

**6.** The authors make frequent assumptions in their analysis that do not appear to have been arrived at in a quantitative manner. For example on line 230 stating "no correction is preferable to an invalid correction," and the discussion around keeping all data in the analysis at L294-298. Some quantitative assessment of the uncertainty introduced in a correction, versus leaving out the correction, or the impact on the data interpretation when keeping a removing certain suspicious sections of data should be made to build a logical argument for making such decisions. I acknowledge here that making this type of measurement and accounting for all errors is very challenging, and at the same time I hope that the authors will consider this comment thoughtfully when refocusing this manuscript.

We agree with this comment and have done the following improvements:

- We present warm (non-suspicious) and cold (more suspicious) data separately, and also

derive correction factors (from the fog monitor comparison) separately for warm and cold clouds (new transmission efficiency figures Figs. 2–4).

- We have added a supplementary figure (Fig. S9) showing the comparison between cloud particles and cloud residuals separately for each cluster, to help quantify the influence of suspicious data/potential artefacts on each cluster.

- We have added a supplementary figure with a new cluster analysis without the most suspicious data (Fig. S12), to show that the results do not change.

- We removed this part of the text.

**7.** The monthly average total particle size distributions (orange curves in Figure S8) appear to have a larger dominance of Aitken mode particles in January and February than have been observed in previous multi-year measurements in Zeppelin (e.g., https://www.atmos-chem-phys.net/16/3665/2016/ and https://www.atmos-chemphys.net/17/8101/2017/). Do the out of cloud, or just before cloud, total particle size distributions look the same as these on a monthly basis? If not, this would suggest that using total particle size distributions measured during cloud events may not be representative of the actual ambient particle populations. How is this observation impacted by the particle loss corrections and density assumptions?

As mentioned above, the appearance and disappearance of clouds can also coincide with a change in air mass. As such we refrain from performing extra analysis in comparing size distributions before and after cloud events. Most importantly, as now shown in the revised manuscript, the comparison of accumulation mode concentration measured behind the GCVI and the whole-air (=total) inlet, showed a remarkably good agreement (factor around 2 for liquid clouds) with the comparison of cloud residual concentration and cloud particle concentration (as measured by the FM-120). This clearly shows that the whole-air inlet at Zeppelin Observatory (which follows the ACTRIS guidelines for aerosol sampling in high-altitude/cold environments) is capable of sampling the entire population of ambient aerosol and cloud particles. In order to be consistent with the new structure of our result section, we have replaced the previous Figure S8 by a new figure showing the monthly averaged cloud residual size distributions (separated by all, warm and cold cloud cases). For this figure (Fig. S8 in the supplement) we decided to only show the cloud residual size distribution due to the larger data coverage (see Tab. S1).

**8. Section 2.4:** What metrics were used to select the appropriate and physically meaningful number of clusters? Mean euclidean distance? Or any other objective way of looking at optimizing cluster number to explain variability in the data with the smallest number of possible clusters? Four clusters groups the two distributions that contain small particles – what evidence is there that these are physically distinct clusters? How was the total particle data incorporated? i.e., the grey size distributions in Figure S7, were they grouped based on the cloud residual clusters? Or clustered separately? If you cluster the out of cloud (I.e., before or after cloud) particle size distributions on their own, do the same five clusters come out?

We looked at various cluster metrics to determine the optimal number of clusters, but the results were inconclusive. Some metrics suggested 2 clusters (e.g. silhouette scores), some suggested 3 clusters (e.g. elbow method / sum of squared errors), while others suggested as many as 17 clusters as the optimum (Davies-Bouldin index). For this reason we decided to test and to show the result for a number of clusters of up to 6 clusters (see Fig. S3 in the SI). It was our intention to find the optimal number of clusters that would carve out the characteristic cluster

with a domination of Aitken mode particles (here cluster 1). We used physical reasoning to find the optimal number of clusters, and by looking at e.g. Fig. 10 (in the revised manuscript) it is clear that cluster 1 is physically distinct from cluster 2. Increasing the number of clusters to more than 5 only lead to an additional split of the accumulation mode dominated residual size distributions (see Fig. S3). Previous studies using cluster analysis on particle size distributions usually use many more clusters (e.g. Beddows et al., 2009), in addition to own judgements in combining the "over-clustered" size distributions to fewer clusters later on (e.g. Dall'Osto et al., 2017; Tunved and Ström, 2019).

The total particle data was then grouped based on the retrieved clusters. Interestingly, if we remove the data points with large disagreement in the FM-120 vs GCVI comparison (points outside the 10:1 and 1:10 lines; see Fig. 2b), similar clusters appear in the cluster analysis (new supplementary Fig. S12) which gives further confidence in the overall results.

**9.** The abstract and conclusions sections contain several statements that are not direct conclusions from this study. In particular, the conclusions at L 545-547 is not something that can be concluded from observations in this study.

We agree and have removed this part of the conclusions. Based on the all comments and the revisions done, we have completely rewritten the conclusions, see below:
"Results presented in this paper are the first direct long-term measurements of size-resolved cloud residual number concentrations of Arctic low-level clouds. It is also the first cloud residual data set that covers more than a full annual cycle, in the Arctic and globally.

[revised manuscript text omitted]

**1.2   Specific Comments:**

**L83-93:**   Observations from satellite are also relevant here, e.g.,:
https://agupubs.onlinelibrary.wiley.com/doi/full/10.1029/2012GL053385

We agree and have added to this particular paragraph the sentence:
"Satellite data show that Ny-Ålesund is located in a region with highest cloud cover in the Arctic (Cesana et al.; Mioche et al., 2015)."

**L135-140:** Is the CVI transmission experimentally determined here explicitly? If so, it should be shown as a function of size in the SI.

The CVI size-dependent transmission efficiency was determined by Shingler et al. (2012). The efficiency used in this work is from their Figure 4 (grey squares). For clarity, a figure with the used transmission efficiency from Shingler et al. (2012), the extrapolation, and the average cloud particle size distribution during our measurement period has been now added to the supplementary material (see Fig. S4 in revised SI) and is being referred to within the result section.

**L145-179:** How well do DMPS 2a and 2b agree in their overlapping size range? How sensitive is the particle loss calculation to the chosen density? A density of 1g/cm3 is likely much lower than the true value.

This information has been added to the manuscript: "In the overlapping size range, the size distributions from DMPS 2a and DMPS 2b were combined by using the data from DMPS 2a in all overlapping bins except the last three. DMPS 2a data were preferred because DMPS 2a is shorter than DMPS 2b, and therefore suffers fewer losses. The last three bins, however, were not corrected for multiple charges, and therefore we used the data from DMPS 2b for those bins instead." See also Fig. 1 below.

In our particle size range, the loss calculation is not very sensitive to the chosen density (hence our previous approximation). The other reviewer also commented on the chosen density being too low, and we have therefore redone the loss calculations assuming a density of $1.5\,\mathrm{g\,cm^{-3}}$ instead. There are some small differences compared to before, but they are only noticeable at the upper end of the size spectrum where sedimentation and impaction losses become more important with this higher density.

[Figure]

Figure 1: Average size distributions from DMPS 2a (teal) and DMPS 2b (red) for the entire data set, to illustrate how they compare in the overlapping size range. Solid and dotted lines show median and mean values, respectively, and shaded areas indicate the 25$^{\text{th}}$ to 75$^{\text{th}}$ percentile ranges. The bins in DMPS 2a that may sometimes be influenced by multiply charged particles (and therefore were not used) are circled.

**L165:** "Manual screen for outliers and contamination" should be elaborated

Overview figures of each single day with all relevant measured parameters (GCVI data, all CPC's & size distributions, meteorology, visibility, etc.) were produced and manually checked. In addition, all lab books were screened. Suspicious periods (e.g. with spikes in the CPC's) that happened at the same time with activities outside (close to inlet) or with other maintenance work that influenced our inlet lines were removed. Other suspicious periods included periods with the GCVI detecting a cloud but with almost no particles measured by the CPC. This happened occasionally when icicles formed on the visibility sensor, which was also manifested in a smooth and almost constant visibility signal. For all these periods we also checked the centrally saved webcam images to confirm that no cloud was present. These clarifications were added to the new Sect. 2.4 (data treatment), as follows:

"The DMPS and GCVI data were processed in several steps. The logbooks from Zeppelin Observatory – which detail dates and times for visits, maintenance, instrumental issues and other observations – were examined, and data were removed when the logbooks indicated that they may be affected by the activity at the station. Next, daily overview plots of all relevant parameters were made, and each daily plot was visually inspected. Outliers (e.g. sudden concentration spikes) and suspected pollution events (e.g. concentration peaks around mealtimes or flight times) were removed. Special attention was also given to data points around gaps in the time series, and if there appeared to be issues in the data leading up to the instrument failure or after reboot, the suspicious data points were removed. Finally, several numerical filters were applied to catch additional outliers that may have been overlooked during the visual inspection. These filters looked for DMPS scans where the integrated number concentration was much higher (¿500 cm$^{-3}$, e.g. caused by electrical sparking inside the DMA) than the concentration measured by the total CPC, data points that showed a much higher concentration than both neighbouring data points (¿1 500 cm$^{-3}$; kept this high so as not to accidentally cut out nucleation events), and scans where the majority of the concentration came from the highest or lowest size bin (indicating sparks in the DMA or possible pollution).

The GCVI system outputs status codes for the operation of each part. When switching on/off of the GCVI occurred during a DMPS 1 scan, that scan was removed (since it is neither in- or out of cloud, and the enrichment factor is not defined for this case). Occasionally, there were also issues with icing of the visibility sensor, which led to the GCVI turning on despite there not being a cloud at the station. These cases were found by comparing the visibility to the measured cloud residual concentration, and data points that seemed questionable (i.e. too low concentration with respect to the visibility) were further investigated. If no cloud was detected by looking at webcam images from the station, or if the visibility was suspiciously constant (indication of icing of the sensor), the DMPS scan for those times were removed.

After the data screening, 1 729 hours of cloud residual number size distribution measurements remained. All cloud residual data are used in Fig. S3; however, for the remaining figures we were limited by the availability of concurrent data from the other instruments (DMPS 2a, DMPS 2b, the fog monitor, the ultrasonic anemometer, and the Cloudnet retrieval). Thus, slightly different subsets of the cloud residual data are used in the different figures. Table S1 shows how many hours of simultaneous measurements we have for different instrument combinations, and which figures the combinations are relevant for.

We have not applied any standard temperature and pressure normalisation or particle shape correction to the data presented here, but multiple-charge corrections have been applied to all measured size distributions. They have also been corrected for particle losses due to diffusion,

impaction and sedimentation using the *Particle Loss Calculator* by von der Weiden et al. (2009), assuming a particle density of $1.5\,\mathrm{g\,cm^{-3}}$."

**L262-267:** Would these very low particle concentrations not be consistent with ice clouds in winter? Also, it should not be the absolute amount of particles that matters, but the difference between the two measurements, which is up to 2 orders of magnitude.

This may indeed be consistent with pure ice clouds. We thank the reviewer for this suggestion, and have added it as a hypothesis in the updated version of the manuscript. However, it is difficult to confirm since Cloudnet does not distinguish between cloud ice and falling snow. Due to this, as well as a comment from the other reviewer, we have now also discussed the possibility that the low particle concentrations also are consistent with shattering artefacts from snowfall (which would also explain the even lower concentrations in the fog monitor, since many snowflakes may be larger than the last channel of the FM-120). This explanation has also been added to the revised version of the manuscript (please refer to response to comment 1 above).

**L285-293:** This appears to be circular logic, and the inability to distinguish between droplets and ice crystals seems to cause a lot of problems here.

Both reviewers raised this issue, and we agree. We have opted to remove this part from the revised version of the manuscript.

**L324-327:** This could be the case, or could be indicative of a change in supersaturation.

We have expanded the explanation to include that a supersaturation change is necessary to activate smaller particles, and this change could be caused by either an increase in updraft velocity or a decrease in particle concentrations: "The apparent $D_{50\%}$ decreases with decreasing temperature, which indicates an increase in cloud supersaturation with decreasing temperature. If the meteorological conditions are otherwise the same, this could be caused by an increase in updraft velocity or by a decrease in particle concentration (less competition for water vapour allows smaller particles to be activated). The latter is consistent with the general decrease in particle concentrations with temperature seen in the first two panels of Fig. 6. "

**L330-332:** Do you expect values up to 2? Does this not give further evidence of shattering, especially at smaller particle sizes? It would be best to include shading around these means to show the range of uncertainty

When values close to 2 are happening in the range of larger particle sizes, the concentrations are very low as can be seen in panel b. Dividing small numbers by small numbers can easily lead to values above 1. At the smallest sizes, it could be an indication of shattering. The ratio fluctuations could also be the result of small uncertainties in sizing, concentration and losses of the two DMPS systems, causing the size modes to not be perfectly aligned. We have added these explanations to the revised manuscript: "The ratios are occasionally above 1 which, at the upper end of the particle size range, could be caused by small number statistics (i.e. ratios of small numbers). In the mid-size range, the ratio fluctuations could be the result of small uncertainties in sizing, concentration and losses of the two DMPS systems, causing the size modes to not be perfectly aligned. "

Adding shades of uncertainty would unfortunately make the panel unreadable, but we have

updated the other figure that includes activation ratios (Fig. 12 in the revised version) to include the ratio of both means and medians, in order to better illustrate the range of uncertainty. We have also added an additional supplementary figure that shows the mean, median, and interquartile range of the activated fraction of particles $> 100\,\mathrm{nm}$ as a function of temperature (similar to Fig. 7 in Verheggen et al. (2007); see Fig. 2 below).

**Figures 3a and 4a:** what do the error bars represent?

The whiskers on the boxplots extend to the last data point within 1.5 times the interquartile range from the nearest quartile (the Tukey original boxplot definition). Data points outside this range can be marked as individual points, or, as done here, excluded from the plot. We have added this information to the figure captions: "The whiskers extend to the farthest points that are within 1.5 times the interquartile range from the nearest quartile. Points that fall outside the whiskers are not shown."

**Figure 4c:** why do the authors think that the data for small cloud residuals is not apparent in this analysis?

The small residuals (cluster 1) is only present in around $8\,\%$ of the data and shows on average the same values of updraft as the other clusters (see Fig. 10b and 9b in the revised manuscript). As such, they are not apparent when calculating the mean residual size distribution binned by updraft.

**Figure 6, 7 and 8:** These figures are showing a lot of the same information in different ways. This could be focused in such a way to make clear what is most important for the reader to see.

We agree and have removed Figs. 6 and 7 from the main manuscript and just focus on Fig. 8 (now 9) to describe the shape of the cloud residual size distributions.

**2  Anonymous Referee #2**

**2.1  Main general comments:**

The authors present long term, ground-based measured cloud particle residual number concentration and size distribution results in combination with other aerosol and cloud particle and meteorological data. Like the authors point out, this is indeed a remarkable data set, helping to improve the knowledge about arctic cloud formation.

We thank reviewer 2 for their detailed and helpful comments.

But:

**A.** According to the diverse uncertainties (representativity of the measurement location, GCVI and FM-120 sampling efficiency, unassignable assumptions in certain parts of the data interpretation), which are honestly admitted in the manuscript, the conclusions are much too high-flown. Examples are "The reported measurements. . .provide a new basis. . .for developing robust parameterizations of mixed-phase clouds in Earth system models" (line 7-9) or "The direct

measurements. . .provide a valuable new perspective on Arctic CCN and INP" (line 551-552), although not one INP size distribution or INP concentration is presented in the manuscript. The ball should be kept low with regard to these concluding statements.

We agree. Within a major restructuring of our result, discussion and conclusion section, we have toned down our language. We removed the particular sentence from the abstract and also made sure to point out that our observations cannot distinguish between INP and CCN.

**B.** The main and crucial point of criticism is the discussion and evaluation of the small residual particles ($< 30$ nm) observed behind the GCVI. The authors try on one hand to find arguments for a cloud physical explanation but on the other hand they admit that artefacts during the measurements cannot ruled out. Unfortunately, they cannot quantitatively estimate the contribution of those artefacts, i.e. is this in the range of 10 or rather 80 %.

As stated in the manuscript, only $8\%$ of the data (cluster 1) clearly showed a different shape. We emphasise that artefacts cannot be ruled out, and have expanded the discussion (including additional figures) to try to quantify when artefacts are most likely, when they are less likely, and what bearing they have on the results. For example, we now show the comparison to the FM-120 for each cluster separately (also giving the percentage of data falling within the 10:1, 5:1 and 2:1 lines), and we have also included a supplementary figure where the cluster analysis is repeated without the most suspicious data points. Please refer to the answers to Reviewer 1, major comments 1 and 2). More details are also given in the answers below.

**C.** In the attempt to explain the small residuals scientifically, the authors provide the possibility of small CCN from droplets or "CCN and INP" material from secondary ice particles, without taking a decision for one of the two possibilities or at least provide supporting arguments from other observations. However, for both presented options there are counterarguments, which substantially reduces the chances of their occurrence. Indeed, CCN have been observed as cloud residuals at a diameter of 25 nm in the study of Schwarzenboeck et al. (2000), but the complete cloud residual size distribution and the size dependent activation fraction looked completely different compared to this study, because the larger CCN became activated to a much higher proportion. In general, the reasoning of the authors, that the sampling of secondary ice would result in small cloud residuals is straight forward. But, in a mixed-phase cloud, the supercooled droplets by far dominate the ice particles in terms of number density, i.e. beside the Aitken mode, many more accumulation mode residuals must be present. And why are these secondary ice particles only be observed as small residuals but not as cloud particles by the FM-120 (even if at wrong diameters). Both counterarguments are more indicative for large particle (most likely large ice particles) shattering at or in the GCVI inlet system and subsequent processing of the created fragments. Moreover, in case these small residuals are coming from secondary ice particles, these residuals are in no way involved in the cloud formation process as concluded by the authors.

We agree. We have added the caveats concerning the secondary ice hypothesis to the discussion of the cluster analysis. Further discussions among the coauthors, sparked by comments from both reviewers, have led us to believe that at least part of the Aitken mode cloud residuals in the winter may be the result of snowflake shattering artefacts, and this discussion has also been added in the updated version of the manuscript. As the reviewer correctly points out, residuals that are the result of scavenged particles, secondary ice processes or shattering artefacts cannot be said to be driving cloud formation since the particles were not acting as cloud nuclei. We have clarified this point in the discussion about the clusters as well. Please refer to answers to

Reviewer 1, major comments 1 and 2 where we have included text from the revised manuscript concerning these issues.

**D.** It is known that the results of many airborne CVI measurements in mixed-phase clouds suffer from artefacts by large cloud particle shattering and that beside this effect, the results are hard to interpret with respect to CCN and INP, because such a CVI exactly like the used GCVI cannot differentiate between droplets and ice particles. This was the motivation to develop at least ground-based CVI systems that are able to separate the liquid and iced cloud phases, like the Ice-CVI (Mertes et al., 2007; Kupiszewski et al., JGR, 2016), the Ice Selective Inlet ISI (Kupiszewski et al., ACP, 2015) or the ice selecting pumped CVI IS-PCVI (Hiranuma et al., AMT, 2016). With those systems the residuals of ice particles can be separated and measured in mixedphase clouds and would be thus needed for the Mt. Zeppelin measurements when mixed-phase clouds prevail.

We agree that it would be warranted to conduct a research project dedicated to a separate sampling of ice and liquid cloud particles at Zeppelin Observatory (which we have added to the conclusions). Continuous Ice-CVI measurements for longer periods are, however, difficult to perform. Also, one should bear in mind that most ground-based ice-CVI's suffer from the same issues as the here used GCVI system: Possible break-up of ice-crystals within the wind-tunnel or the impaction plate inside the ice-CVI (E. Weingartner, pers. comm., Nov. 2020). As discussed above, we have taken great care to evaluate the validity of our overall results and further analysis/discussion has been added to the revised manuscript. One important additional analysis and comparison of the temperature-dependence of the activation ratios of mixed-phase clouds to the observations of Verheggen et al. (2007) now also shows that the observations from Zeppelin Observatory (see Fig. 2 below and in SI) agree well with those from the high-alpine site Jungfraujoch (where the ice fraction within the cloud was directly measured and and ice-CVI was used). Although the temperature-dependent activation ration of mixed-phase clouds by Verheggen et al. (2007) was determined with a whole-air/interstitial inlet system set-up, we reach a similar result by using the GCVI/whole-air inlet set-up, which overall gives us strong confidence in the validity of our results.

**E.** Another drawback is the choice of the cloud particle sensor, which is the major reference instrument for the GCVI in this study. Due to its fixed position its detection efficiency is not clear and moreover it is also not able to distinguish between droplet and ice particles. But for the wintertime measurements and the discussion that the Aitken mode particles are related to ice formation in the cloud a ice particle sensor is crucially needed. It would also shed light into the question whether the measurements are real or mostly caused by artefacts. Concluding, the manuscript has to be refused in the actual form including the discussion of the wintertime, most likely mixed-phase clouds residual measurements, where the Aitken mode residuals are assumed to be involved in ice particle formation. Even major revisions would not help there. On the other hand, when the authors could force themselves to restrict their study to the presentation and discussion of pure liquid clouds, their work would indeed provide valuable progress in the understanding of cloud formation in the Arctic. So, the authors should feel encouraged to change their manuscript in this appropriate way. I do not know if this can be done at this stage of revision or needs a complete new submission. In hopes that the authors go in this direction, I made a complete review hereafter with minor general comments and specific remarks. Many points are related to the Aitken mode residuals in order to indicate in detail the large doubts of the actual interpretation, but this would then be irrelevant for a revised as requested manuscript.

We thank the reviewer for their really detailed and valuable comments, which helped to improve the revised version of our manuscript. As suggested also by reviewer 1, we have shifted the focus away from the Aitken mode particles being important for Arctic cloud formation and give the entire 2-year dataset more equal weight.

The reviewer suggested that we restrict our study to only focus on pure liquid clouds, i.e. discarding all data collected during the winter months. We disagree with this, but we have taken the concerns seriously and have majorly refocused and restructured the manuscript. Based also on the comments of reviewer # 1, we now separated the presentation of our results for warm and cold clouds and performed a major restructuring of our the result, discussion and conclusion section. We chose not to discard the cold cloud data for two main reasons:

1. We believe that our new manuscript structure makes it clear to the reader which data points are likely to be affected by artefacts and which are not, and that the toning down of the conclusions relating to those data points now makes the paper well balanced, thus resolving the issues raised by the reviewer.

2. Analysing and interpreting CVI data is not a trivial task, and we believe it is of great importance and value to the scientific community to clearly communicate the types of issues that may be encountered when carrying out observations like these. Refraining from showing and discussing "suspicious" data would mean withholding potentially valuable knowledge. By being transparent and openly and honestly discussing potential problems, we hope to make life easier for others who might wish to carry out similar observations.

We openly address and quantify the issue of potential artefacts and the need for further more detailed observations dedicated to the cloud phase. As discussed in more detail above (and Figure 2 below), we now show that our observations do generally agree well with observations of mixed-phase clouds by Verheggen et al. (2007).

All in-situ cloud observations (ground-based or airborne) are unavoidably affected by instrumental and set-up-specific choices. The FM-120 is indeed influenced by wind direction and particle shape. This is true for many other airborne and ground-based cloud and aerosol sensors (Baumgardner et al., 2017). The here used FM-120 was thoroughly evaluated by Koike et al. (2019) and by own analysis and no clear bias from the wind direction was observed at Zeppelin observatory. The sampling efficiency of the GCVI determined using FM-120 data and determined from comparing the accumulation mode concentration of the cloud residual and total size distributions for warm clouds delivered almost the same factor (see revised manuscript and details below). We have added a supplementary figure (Fig. S7 in revised SI) where the cloud residual and cloud particle number concentration comparison has been separated by wind direction. In general, no clear influence of the prevailing wind direction on the concentration comparison can be observed. As such we are convinced that the FM-120 measurements are trustworthy and valid for most parts. The here used CVI system has been thoroughly evaluated in various wind-tunnel studies and in over 30,000 in-flight droplet size distributions/residue concentration intercomparisons (e.g., Shingler et al., 2012, and references therein). Given the thorough evaluation of the GCVI within previous and the here presented work, and the now provided balanced and newly structured presentation and discussion of our results, we hope that we have convinced the reviewer about the validity and novelty of our work.

**2.2 Minor general comments:**

**1.** Many different number concentrations are elaborated in this study. The authors should think about to use an abbreviation for each of them to make life easier for the reader and themselves.

It is a good point, but with all the different correction factors and size cuts that occur in the different concentrations, it is difficult to device an abbreviation system that is not overly cumbersome. We hope that the restructuring and revisions of the manuscript will help make life easier for the reader.

**2.** To confirm rather important statements or analysis approaches only references are provided that do not really help the reader. Here the explicit plot in the reference should be given or a plot in the supplement would be highly appreciated. Moreover, some few studies are cited that did not really support the given statement or observation and thus needs intensively revised. These text passages are all mentioned in the "specific remarks".

We agree and have added the missing information to the text and the revised manuscript/SI (see comments below).

**3.** It is totally unclear why and how the sampling efficiency of an airborne, forward looking CVI are applied to an upward looking ground-based CVI. Since this the basis for the statement that the residual particle concentration agrees rather well with the cloud particle concentration, the applicability of this approach must be justified in the manuscript, which is totally missing.

Computational fluid dynamics modelling of the GCVI wind tunnel flow field demonstrated that droplets smaller than roughly 60-70 micron diameter followed the flow streamlines and reached the 100 m/sec velocity within the wind tunnel throat upstream of the CVI tip. Under these findings, the sampling efficiency of the CVI inlet that was previously measured in an aerosol/droplet wind tunnel and validated by over 30,000 in-flight droplet size distribution/residue concentration intercomparisons, as outlined by Shingler et al. (2012), applies. As part of a separate in-house GCVI characterization project by the manufacturer a small cloud chamber was constructed and the GCVI wind tunnel sampled droplets from the chamber in parallel with an FSSP. Agreement between the CVI enhancement factor-corrected FSSP droplet number concentrations above the CVI cut size and the residue particle concentrations measured downstream of the CVI by a MCPC were within experimental uncertainty, typically 25% (F. Brechtel, Oct. 2020, pers. comm.). We have summarised this information in the main manuscript as follows: "Shingler et al. (2012) measured the sampling efficiency of the CVI inlet in an aerosol/droplet wind tunnel, and it has been validated by over 30 000 in-flight droplet size distributions / cloud residual concentration intercomparisons. Computational fluid dynamics modelling and a separate GCVI characterisation project by the manufacturer in a small cloud chamber showed that the Shingler et al. (2012) sampling efficiency applies to the GCVI; agreement between the corrected droplet number concentrations above the GCVI cut-size and the cloud residual particle concentrations measured downstream of the CVI by an MCPC were within experimental uncertainty, typically 25% (F. Brechtel, Oct. 2020, pers. comm.)."

**4.** The order of plots in the supplement is not the same as their mention in the manuscript, which should be harmonized.

We have harmonised the order of figures in the SI with the order in the main manuscript.

**5.** It is unclear whether chapter 3.1 and 3.2 are not so much part of the assessment of the GCVI sampling efficiency but rather much more part of the scientific cloud analysis, maybe except the observation that the GCVI sampling efficiency is reduced at stronger upwind velocities. The authors should think about that.

We agree with this, and have taken it into consideration when restructuring the manuscript. The revised/remaining parts of these sections are now included as independent results rather than as part of the GCVI assessment.

**6.** In Addition, it is incomprehensible why the existing ambient aerosol particle data set (out of cloud) is not used at all in the analysis. Using short periods before and after cloud appearance, and in comparison with the total aerosol particle data set (inside cloud), the quantitative functionality of the whole air inlet inside cloud could be evaluated. When this is confirmed it would be another approach to derive a GCVI sampling efficiency by scaling the large part in the residual size distribution ($> 100$ nm) with the total aerosol particle size distribution assuming that those particles are activated with 100%. This result could be again compared with the approaches applied already in this study.

The appearance or termination of a cloud event can be caused by a change in air mass with differences in aerosol properties. As such, we refrain from an additional analysis of particle size distributions before and after the cloud event (cloud residual and whole-air) since we believe that the two-year (concurrent) dataset stands for itself (over 1300 hours of observations).

We thank the reviewer for the excellent suggestion of including an alternative way to determine the GCVI sampling efficiency for liquid clouds. This has now been incorporated into the section about the GCVI assessment (which, additionally, has been divided to consider warm and cold clouds separately in line with other remarks from both reviewers). This additional analysis gave similar results as the efficiency determination using the FM-120 which further increased confidence in our observations. We include the text relating to the reviewer's suggestion here for easy access, and ask the reviewer to refer to Fig. 4 in the revised manuscript): "For warm (liquid) clouds, the sampling efficiency could also be estimated by scaling the cloud residual size distribution to the total particle size distribution under the assumption that all accumulation mode particles activate into cloud droplets. We compared cloud residual and total particle concentrations (30 min mean values) integrated above different accumulation mode threshold diameters, $D_{\mathrm{cut}}$, in the range 41–505 nm. Figure 4a shows an example scatter plot for $D_{\mathrm{cut}} = 123$ nm, with the corresponding ODR fit parameters (grey data points correspond to cold clouds and are not included in the fit). Figure 4b–c shows ODR best fit slopes and coefficients of determination for all the different $D_{\mathrm{cut}}$ diameters (results from cold cloud data are included for completeness, but not used since we then cannot assume liquid droplet activation).

Above approximately 100 nm, the slope plateaus around 0.46. Given that this method does not take into account the cloud particle cut-size of the GCVI inlet, we can expect this estimate of the transmission efficiency to be a little lower than the one derived from the fog monitor data (0.51). A value of 0.46 is just within the standard deviation of the ratio from Fig. 3, so these two independent methods agree remarkably well. To be conservative, and to be consistent with the cold clouds (next subsection), we have corrected all (warm) cloud residual size distributions and concentrations by a factor of 2 (from Sect. 3.2.1 onward) assuming cloud residual size and cloud particle size are not correlated."

**2.3 Specific remarks**

**L.3:** This statement is not correct. There are aerosol particles contained in cloud particles due to impaction scavenging or secondary ice formation, which are not involved in cloud formation. So the authors must be more precise and need to reword their description here.

We agree and have added "...and cloud processes" following the "cloud formation". In addition, we clarified on the contribution and origin of cloud residuals in the introduction and method section.

**L.4:** There is a too drastic change from the research description to the used instrument. This should be done more smoothly, especially not every reader will know a "groundbased counter-flow virtual impactor inlet system" and its peculiarities.

We agree and have modified this part:
"To continuously sample cloud droplets and ice crystals and to separate them from non-activated aerosol, a ground-based counter-flow virtual impactor inlet system (GCVI) was used. A detailed evaluation of the here used GCVI system is also presented."

**L.6-7:** That not only liquid but also mixed-phase clouds exist in the Arctic is known and not only suggested by prior work. Hence, this statement is trivial and should be reworded.

We agree and have re-written the last part of the abstract that now also addresses the changes made during the review process. Please see response to reviewer 1 above (major comment 9).

**L.16:** The reference should be in brackets.

Changed accordingly.

**L.20-22:** It looks like that something is missing in this sentence. If not, the statement is not totally clear and should be formulated more clearly. What is the cause what is effect?

We have clarified this sentence. It now reads: "Crucially, the autumn and winter seasons are also when Arctic amplification is most pronounced (Serreze and Barry, 2011; Maturilli and Kayser, 2017). This, in combination with the low background particle concentrations, makes the Arctic autumn and winter seasons more likely to experience large relative changes in aerosol particle concentrations and, consequently, changes in cloud properties."

**L.29 and L. 33:** What should be the meaning of "typically"? Should be deleted.

Agreed and deleted. Since the size cut between Aitken and accumulation mode is not sharp, we used $\gtrsim$ and $\lesssim$ instead of $>$ and $<$, respectively.

**L.30:** Which scavenging is meant by the authors? Nucleation or impaction scavenging? This is an important aspect and should be mentioned.

Nucleation scavenging. We have updated the text.

**L.34:** Which particle sinks are meant? The needs to be explicitly listed here for a better understanding.

This relates mainly to an increase in accumulated precipitation. We have added to this sentence, which is also shown by Tunved et al. (2013) in their Figures 14-15 (reference is given at the end of the sentence): "...(i.e. precipitation)..."

**L.50-51:** delete "for example".

We have deleted it.

**L.55:** "performed" should be replaced by "carried out" or something else, since only actors "perform" on stage.

This has been changed.

**L. 56:** not only number concentrations but add also number size distributions.

True. We added "and size distributions".

**L.59:** it is obvious that the cloud particles are ambient, so this term should be omitted in the manuscript, especially since it could come to confusion with ambient particle size distributions where the term "ambient" is needed.

We agree and removed the word here and throughout the manuscript where appropriate.

**L.63-64:** This listing is redundant to the listing in L.58-59, so the authors should do it completely at one text passage and not two times.

We agree and have removed the redundant parts.

**L.116:** According to Fig.1 and chapter 2.3.1 only 2 L/min sample flow were indeed used to take the residual particle measurements. Thus, 13 L/min sample flow were not used at all. This is a pity, because when the sample flow would be reduced to the required 2 L/min, the CVI enrichment would be substantially enhanced, which would significantly improve the measurement statistics, which is especially important in the arctic clouds with less cloud particles. The increase in enrichment would increase from about 12 (L.133) to about 90! So, the authors should explain why they did not use this option or if this it is not possible to operate the commercial GCVI with this flexibility.

Parts of the remaining sample flow were used for other instrumentation from Stockholm University and collaborators which are not part of this manuscript. We added the following sentence to the instrument section: "It should be noted that other instrumentation besides the ones used in this study, and listed in Table 1, were operated during the years 2015 and 2018 behind the GCVI. That is why we used the overall high sample flow."

**L.121-122:** How save is this trap? Could it be that larger crystals and graupel could break-up in this trap, so that the smaller fragments become resuspended in the system and deliver residual particles in DMPS1? The authors should comment on this and best of all provide a short text passage in the manuscript.

The larger crystals and graupel that would be expected to break-up and shatter in the trap inside the CVI inlet would already have done so upstream of the CVI in the wind tunnel because they

are too large to follow the streamlines all the way to the CVI tip. Whether an impact can produce shattered crystals can be estimated using the non-dimensional Weber number (proportional to air density $\times$ velocity$^2$ $\times$ diameter). Sampling conditions with critical Weber number values between 10 and 12 are expected to produce fragments (Twohy et al., 2003). In the CVI these conditions occur at 100 m/s air speed and droplet diameters between 70 and 100 microns. If the shattered crystal fragments were larger than the CVI cut size they could be sampled by the CVI, however, a significant fraction of these fragments would likely not be aligned with the streamlines and would not enter the CVI. If particles would shatter inside the inlet, such events are typically clearly seen in timeline data of cloud residual particle concentration, where much higher concentrations of residual particles are observed by particle counters downstream of the CVI. Over the 11 years of deployment of over 20 CVI inlets on research aircraft and the last 7 years of deployment of the ground-based version of the CVI, artefact particle production relating to the particle trap has not been observed (F. Brechtel, Oct. 2020, pers. comm.).

**L.124:** For the WMO fog definition a solid reference should be given. Moreover, for a reader engaged in cloud physics it would be pretty much appreciated when this value would be transferred and provided as a minimum LWC, which gives a much better impression about the definition of a cloud in this study.

WMO's definition of fog can be found on the WMO's website (`https://cloudatlas.wmo.int/en/fog-compared-with-mist.html`). The current version of the "International Cloud Atlas: Manual on the Observation of Clouds and Other Meteors" from 2017 can only be found online. It can be found in various technical reports of WMO. The latest one is WMO (2008), where it e.g. states: "In climatological summaries, however, all occasions of visibility of less than 1 km are regarded as fog." We added this reference and the online version to the revised manuscript. In addition we now state the corresponding range of LWC as determined by the FM-120 by adding the following sentence: "This threshold corresponds to a large liquid water content (LWC) range of 0.0004 gm$^{-3}$ to 0.10 gm$^{-3}$ (5th and 95th percentile) with a median value at 0.01 gm$^{-3}$ as measured by the FM-120 (see below) for liquid clouds (temperatures above 0°C)"

**L. 158-159:** Why are two CPC used behind the whole air inlet to measure the total aerosol particle concentration? And the data of which of the two CPC are used in the study?

The second total CPC is mainly a back-up CPC in case the first total CPC is failing. For the number concentrations shown in the manuscript we used the integrated and loss corrected particle number size distributions. This was to make sure the data were comparable, since the total CPCs of the different DMPS systems are not all the same model and therefore have different cut sizes. However, when comparing the residual number concentrations to the cloud particle concentrations, the total CPC (behind the GCVI) was used.

**L. 164:** The used particle density is the one of liquid water, but the residual particles are measured at very dry conditions, so the effective particle density is 1.7 g/cm3 or even higher. The same argument holds for the total particles, but due to an expected higher RH in the sampling line, the effective particle density might be 1.5 g/cm-3. The particle density has a rather large influence in particle loss calculations, i.e. these correction calculations has to be repeated and applied to the presented data.

The choice of density has no significant influence (hence our previous approximation), but we agree that $1.5\,\mathrm{g\,cm^{-3}}$ is a more realistic value and have redone the loss corrections with this

density instead. The changes are only marginal and mainly affect the larger end of the DMPS size range.

**L.165:** What is meant by contaminations? The authors need to mention those sources of error explicitly to give the reader a better expression, which situations have been excluded from the study and which not.

Contamination means influence of particles from local activities such as maintenance on the roof of the station, the inlet system or other instruments. Our data screening involved that overview figures of each single day with all relevant measured parameters (GCVI data, all CPC's & size distributions, meteorology, visibility, etc.) were produced and manually checked. In addition, all lab books were screened. Suspicious periods (e.g. with spikes in the CPC's) that happened at the same time with activities outside (close to inlet) or with other maintenance work that influenced our inlet lines were removed. Other suspicious periods included periods with the GCVI detecting a cloud but with almost no particles measured by the CPC. This happened occasionally when icicles formed on the visibility sensor, which was also manifested in a smooth and almost constant visibility signal. For all these periods we also checked the centrally saved webcam images to confirm that no cloud was present. These clarifications were added to the new Sect. 2.4 (data treatment), see also response to Reviewer 1, specific comment L165, where we have included the text.

**L.175:** The consequence of the sentence is not quite clear. Does it mean the minor measurement efficiency of DMPS-1 for small diameters was not corrected at all or only not if the concentration in the respective bins was zero. In the presented data, the residual particle size distributions obtained by DMPS-1 often show non-zero concentration down to 10 nm, so that a diffusion loss correction could be in principle applied and it would be also possible to derive a correction from Fig.S2b, which would in both cases better than no correction. Therefore, the authors should say something about this.

We do apply the loss correction, perhaps the wording was unclear. What is meant is that, while the correction is applied, there are cases where it has no effect (namely when the concentration in the bin in question is zero). We recognise that this is a trivial statement that only causes confusion, and have therefore removed the sentence.

**L. 183:** From Fig.S1 it is obvious that about 2/3 of the cloud sampling was carried out at south wind and 1/3 at north wind conditions. This implies that for 2/3 of the cloud sampling the aspiration efficiency of the FM-120 was very good and for 1/3 of the cloud sampling the aspiration efficiency was pretty bad (since the cloud particles have to make a U-turn to get measured, which will be additionally modulated by the wind speed). Consequently, the first idea would be to do the analysis with the 2/3 south wind cloud cases. The authors should therefore try to go in this direction or present very good arguments, supported by a data evaluation in this respect, why the measurements from all wind directions are equally suited. The mention of only a reference (Koike et al., 2019) is definitely not sufficient.

Koike et al. (2019) performed thorough analysis of number concentration of cloud elements as a function of wind speed and direction (page 1802 and Appendix A). They did not observe any signature of enhanced particle losses and there is no clear difference between number of cloud elements, neither cloud water content for data observed in northerly and southerly winds. Following the reviewers suggestion, we have added a supplementary figure (Fig. S7 in revised SI) where the cloud residual and cloud particle number concentration comparison has been

separated by wind direction. In general, no clear influence of the prevailing wind direction on the concentration comparison can be observed. Together with the results from Koike et al. (2019), we believe this supports using data from all wind directions.

**L. 194:** It would be more meaningful to write the METEK user manual explicitly into the brackets and not treat it as a reference citation.

Changed accordingly.

**L. 197:** Why does the authors did not correct the temperature according to this noncorrect temperature difference?

We originally did not correct it because the reference sensor is at a different altitude compared to our ultrasonic anemometer and the temperature sensor in the GCVI was broken for most of the measurement period. However, we have now analysed and compared the GCVI temperature to the uSonic temperature during the period when the former was not broken, and found that the uSonic temperature was, on average, 2.6°C higher than the GCVI temperature. We have subtracted this offset from all the uSonic temperature data presented in the revised manuscript. We added the following sentences to the revised manuscript: "In the Arctic, this temperature difference was larger. The GCVI inlet has its own temperature sensor, but it was only working for a few months at the start of our measurement period. During the overlap period, the difference between the measured acoustic temperature and the ambient temperature measured by the GCVI temperature probe was around 2.6°C. Thus, we have subtracted 2.6°C from all temperatures measured by the ultrasonic anemometer."

**L. 220-222:** See comment for L. 183. It should be easy to carry out a FM-120 and GCVI comparison for the 2/3 south wind and the 1/3 north wind conditions to demonstrate whether there is a wind direction or not. This is much better than to "assume" that the fog monitor provides an accurate representation of the cloud particles entering the GCVI. This should be additionally restricted to droplets, since the FM-120 will not correctly measure ice particles.

Please see comment above (L.183) and new Fig. S7 in revised SI.

**L. 222-224:** The complete GCVI sampling efficiency correction procedure is a very important part of the paper, which the reader must follow more easily. Thus, the mentioned transmission efficiency has to be explicitly shown as a graph in the supplement as only provide the Shingler et al. (2012) reference, especially because the original size dependent efficiency was extended.

We agree. A new figure (see Fig. S4 in revised SI) with the used transmission efficiency from Shingler et al. (2012), the extrapolation, and the average cloud particle size distribution during our measurement period has been added to the supplementary material and is referenced within the text.

**L. 227:** How dry is the counterflow? This is not mentioned anywhere else.

We added to the method section (CVI description): "The dew point of the dry counterflow produced by the dry air generator was -40°C. "

**L. 230:** The expression "no correction is preferable to an invalid correction" is pretty unscientific. Should that be a general statement or only related to the concrete approach? If the latter

is the case, the authors should somehow quantify why the error is smaller with no correction or should simply remove this expression.

We agree and have removed this expression since it is not really needed.

**L. 231:** The expression "corrected cloud particle concentrations" is badly chosen and misleading. "The GCVI sampled cloud particle concentrations according to the transmission correction" or something similar would make it clearer.

We agree and have changed this sentence to: "The cloud particle concentrations, integrated above the GCVI cut-size and multiplied by the GCVI sampling efficiency, were compared ..."

**L. 236-238:** For the data points below the 1:10 line, it would be very important to know the measured FM-120 cloud particle concentration in order to evaluate how thick the clouds were during these measurements.

We added this information to the revised manuscript: "The second group [i.e. the points below the 1:10 line] is associated with very low cloud particle concentrations ($1 \pm 3\,\mathrm{cm}^{-3}$, total concentrations without cut-size and Shingler et al. (2012) correction) and the cloud particles are also fairly large in size ($11 \pm 4\,\mu\mathrm{m}$ effective radius)."

**L. 247-252:** At least the reference of Mertes et al. (2007) is used here in a wrong manner. When there is more than one aerosol particle in or on a ice crystal they will not be emerged as single particles during the drying process in a CVI. These particles will remain on the more and more shrinking ice particle until they will lump together and are released as one particle. Only if the original ice crystal breaks-up the scavenged aerosol particles will be counted individually. This could happen in the cloud (secondary ice) or by hitting any surface of the GCVI. The difference here is that secondary ice would have counted by the FM-120 too, whereas the latter would only be seen as increased concentration in the GCVI, as a shattering artefact, like it is described here.

Thank you for this comment. The other reviewer said "Another possibility is coagulation scavenging of Aitken mode particles with ice crystals, which could lead to the observed residual size distributions upon shattering and/or evaporation in the CVI." In their view, it seems that shattering is not necessary for more than one residual to be released. We have not been able to find any references that describe how rimed or scavenged particles would behave on an evaporating ice crystal. In the revised version of the manuscript, we have removed the Mertes et al. (2007) reference and weakened our statements to be more speculative in nature. However, if the reviewer has any references in mind that would help elucidate this issue, we would be very happy to receive them and add them to the final manuscript. The part about secondary ice particles also being seen in the fog monitor has also been clarified in the new version of the text.

The revised paragraph now reads: "Riming or impaction scavenging of interstitial aerosol particles onto an ice crystal may be able to result in more than one cloud residual emerging from the crystal as it dries inside the GCVI inlet. If more than one residual could be released through this process without the need for crystal break-up, it could be an alternative explanation that is consistent with the difference in cloud residual and cloud particle number concentrations. However, this is speculative since no strong experimental evidence exists on how rimed particles would behave inside the CVI flow regime."

**L. 253:** Again a citation, this time Lauber et al. (2018) is used in a wrong sense, because it does not show or even treat the topic of a INP break-up, which indeed is not possible in nature.

Thank you for spotting this. The Lauber-reference referred to the second part of the sentence about liquid droplets that can eject material when freezing as shown by Lauber et al. (2018). This sentence has been removed from the revised version.

**L. 263:** What is the quantity of "sometimes"? Better change "that sometimes occurs" into "that was observed".

Changed accordingly.

**L. 264:** The acceleration and deceleration zones are not the only possible locations. Since larger cloud particles could not follow the streamlines, they will simply hit inlet surfaces and walls where the shattering occurs. This needs to be added here.

We agree. In addition, one should keep in mind that the cloud particles that break within the wind tunnel of the GCVI need to produce large artefacts (above the cut-diameter of the CVI) in order to be sampled. Break-up after the stagnation plane of the CVI is less likely and has not been observed. We have looked again at periods of our data when the cloud residual size distributions were dominated by cluster 1 (which we attributed to clouds with contributions of ice and snow). The total CPC data behind the CVI did not show any suspicious spikes which could be an indication of crystal shattering within the CVI (or the following sampling line). As such, if crystal shattering occurred, it most likely happened within the wind tunnel above the CVI inlet. Also in reference to comments by reviewer 1, we have clarified these aspects and changed this paragraph (please refer to the response to Reviewer 1, Major comment 1).

**L. 265-267:** Not the concentration but the size of the particles is the important parameter.

Yes, good point. We have added this.

**L 279-280:** The precipitating particles could easily explain the difference in concentrations, since they could also easily shatter at the GCVI surfaces and their fragments are entering the GCVI and their residuals are counted.

This is true, and we have added discussion about the potential influence of snowflake shattering artefacts in the revised manuscript. See response to Reviewer 1, major comment 1. We agree with the reviewer that it is a likely explanation. That said, also bear in mind that the fragments need to be larger than the GCVI cut-size and aligned with the streamlines in order to be able to enter the inlet and be sampled. The wind tunnel also has a rain cover to avoid rain droplets entering the inlet.

**L. 284:** Spiegel et al. (2012) only studied liquid droplets and not ice crystals, i.e. this citation is incorrectly used here.

We agree and removed the reference (although the same loss mechanisms for droplets are also relevant for ice crystals).

**L. 285-293:** What is this about now? First the FM-120 was supposed to be correct in order to evaluate the GCVI sampling, now the GCVI concentration results are used to scale FM-120

concentration measurements, and this only when the first is higher than the latter, to account for a FM-120 undercounting. This is dubious and a bit helpless. The much better approach would be to separate the cases with and w/o the presence of cloud ice by an appropriate sensor to prove all these assumptions. Since the amount of ice particles in a mixed-phase cloud is much smaller than the amount of droplets the ice undercounting of the FM-120 would not have much effect.

Both reviewers raised this issue, and we agree. We have opted to remove this part from the revised version of the manuscript. A sensor that discriminates the cloud phase was unfortunately not available and it is clear that these measurements would be really valuable.

**L. 294-298:** This is really a funny conclusion: I cannot really quantify the sources of error in my instruments, therefore I use all the data!

We agree that our wording sounded too insecure at this point. With the new analysis and the re-structuring of this section (warm and cold clouds separately), we improved this part.

**L. 303:** The size-dependent transmission efficiency must be shown in the supplement or at least the exact plot in Shingler et al. (2012) must be provided.

A figure with the used transmission efficiency from Shingler et al. (2012), the extrapolation, and the average cloud particle size distribution during our measurement period has been added to the supplementary material (see Fig. S4 in revised SI).

**L. 309:** I would call it "for the sake of convenience".

We really appreciate the high detail the reviewer has commented which clearly helped to improve our manuscript. However, statements like "for the sake of convenience", "funny conclusion" or "dubious and a bit helpless" are neither encouraging nor constructive.

**L. 324:** All particles measured behind the GCVI are residuals. Most likely the authors wanted to say "are in fact CCN".

Yes, we agree. This has been fixed.

**L. 324-328:** The decisive parameter for this observation at unaltered background aerosol is an increase in cloud supersaturation. This needs to be explicitly mentioned here. A decrease in particle concentration is only a possible cause, just like for example a higher updraft velocity.

We have expanded the explanation to include that a supersaturation change is necessary to activate smaller particles, and this change could be caused by either an increase in updraft velocity or a decrease in particle concentrations: "The apparent $D_{50\%}$ decreases with decreasing temperature, which indicates an increase in cloud supersaturation with decreasing temperature. If the meteorological conditions are otherwise the same, this could be caused by an increase in updraft velocity or by a decrease in particle concentration (less competition for water vapour allows smaller particles to be activated). The latter is consistent with the general decrease in particle concentrations with temperature seen in the first two panels of Fig. 6. "

**L.330-333:** a) there should be no uncertainty in the CVI enrichment factor. If so, the authors need to explain this in the manuscript here or before when the working principle of the GCVI is

explained. b) The uncertainty of the sampling efficiency would explain a level of the ratio below or above 1 but could not explain the variations of the ratio. Maybe the counting statistics is not sufficient. The authors, should check for the real reasons for these variations.

We have removed the part about the EF uncertainties as we agree that there should be none (unless you count uncertainties in flow velocities). We have added some alternative explanations, so the text now reads: "The ratios are occasionally above 1 which, at the upper end of the particle size range, could be caused by small number statistics (i.e. ratios of small numbers). In the mid-size range, the ratio fluctuations could be the result of small uncertainties in sizing, concentration and losses of the two DMPS systems, causing the size modes to not be perfectly aligned."

**L. 339-340:** This statement is not true, because when the same size modes would be present in the whole air inlet there would be no peak at 20 nm in Fig.3c. So, Fig.S7 is not a good proof for the given statement, in addition because Fig.S7a also shows this peak.

Here the reviewer most likely misunderstood our arguments. The whole air inlet is very different from the GCVI inlet. Also the flows, velocities and the way particles and clouds are moving inside these two inlets are very different. Thus it is very unlikely that both inlets will produce the same artefacts with a similar frequency and size distributions. We have clarified this statement and moved it to the discussion concerning the clusters (where we have consolidated most of the discussion about artefacts and alternative origins of the small residuals). The statement is not meant to be understood as "there are no artefacts", but rather as "we see the same size particles in a different inlet that should not suffer from the same types of artefacts, therefore there may be other explanations for the small particles (in addition to artefacts)."

**L.340-341:** The statement in L.339-340 (beside the point that it is not really the truth) is not an argument that there are no reasons to expect droplet and even more ice crystal shattering here.

Please see the answer to the previous comment.

**L.343-362:** It is totally surprising that there are no residual particle size distributions and ratios residual/total present that show this "ice present effect" of small residuals observed in chapter 3.1. Why is that? These "cold cases" should also be found when the size distributions are sorted by updraft. An explanation by the authors is definitely needed here.

Cluster 1 (the small residuals) is only present in around 8 % of the data and shows on average the same values of updraft as the other clusters (see Fig. 10b and 9b in the revised manuscript). As such, the "cold cases" are not apparent when calculating the mean residual size distribution binned by updraft, and therefore not apparent in the calculated ratios.

**L. 367:** Why is no 25th percentile given here, like it is two lines before for the residual number concentration?

This has now been added.

**L. 375-376:** This sentence is not comprehensible. What is meant by "in terms of when peak saturation occurs"? And what is meant by "some differences to previous measurements" in detail?

The reviewer probably has misread the text. In the original text in lines 375-376 it is written "peak concentration" and not "peak saturation". The differences related e.g. to the monthly maximum in peak concentration, which other studies have observed in other months. These differences come natural due to the natural annual variability in aerosol properties. For clarification, we have changed this sentence to: "There are some differences to previous measurements at Zeppelin Observatory related to the natural variability of aerosols, for example in terms of when peak concentration were observed."

**L.381:** The residual particle annual cycle cannot be "confirmed" by the cloud particle measurements. But one could say, that it is "closely related to the cloud particle annual cycle.

Agreed, we wanted to indicate that there is relatively good agreement between the GCVI and the fog monitor. We have reworded the sentence to make it clearer: "As seen in Fig. 2a, the shape and the magnitude of the cloud residual annual cycle agree nicely with the cloud particle annual cycle."

**L. 393:** It is not convincingly proven that this "clear seasonality" is not caused by the artefacts of ice particle shattering. Therefore, I would not use this expression here.

We have removed this expression in the revised version of the manuscript.

**L. 430:** What is meant with "total number concentration" in Fig.S11?

We have clarified this sentence, so that it now reads "number concentration of both cloud residuals, cloud particles, and total particles".

**L. 436-440:** This argumentation supports to some extend the option of large cloud particle shattering as an explanation for the occurrence of the Aitken mode residuals. It is really a pity that no ice particle sensor was operated to bring more light in this problem.

We agree that it is a pity, and hopefully future studies can shed more light on this. We have included this line of argument in the discussion of artefacts in clusters 1 and 2, with the sentence "Large crystals are expected to be more prone to shattering, and indeed clusters 1 and 2 are related to larger cloud particles than the other clusters (Fig. 9a)." See also response to Reviewer 1, major comment 1 for further changes to the text.

**L. 441:** Do the authors indeed mean "ambient" particle size distributions (w/o cloud) or is it more likely "total" particle size distributions (within cloud)?

Both measurements are concurrent and in the text we have changed "ambient" for "total" now (which includes interstitial particles and residual particles resulting from dried cloud particles).

**L. 434-448:** In contrast to the discussion in the manuscript both Cluster 1 and 2 are most likely an indication for an artefact sampling due to ice particle and large droplet shattering. Cluster 1 are very small Aitken mode particles, but related to a broad size range of cloud particles. The larger cloud particles most likely experience shattering at the GCVI and the very small ones would release only one residual. Maybe this very small fraction might be due to secondary ice formation, but this is not further discussed in detail in this study. The flat shape of Cluster 2 is a clear indication of large cloud particle shattering, realistically indicating

that in this way all particle sizes between 20 and 200 nm occur with more or less the same frequency. At least there is no other mechanism known that could account for such a shape. The existence of artefact measurement by shattering is strongly supported by the fact that this cluster is related to the largest cloud particles measured. The connection of both clusters with low temperatures and the existence of ice particles further support the occurrence of ice particle shattering artefacts.

We have expanded the discussion and include the possibility that these clusters are affected by artefacts. However, in contrast to what the reviewer suggests, we believe Cluster 2 is influenced to a much lesser extent than Cluster 1.

Further discussions among the coauthors, sparked by comments from both reviewers, have led us to believe that at least part of the Aitken mode cloud residuals in the winter may be the result of snowflake shattering artefacts (most likely in the wind tunnel of the GCVI), and this discussion has been added in the updated version of the manuscript. We suspect snow because particles need to reach a certain size for shattering to be likely, and this size ($\sim 70\,\mu$m) is far larger than the main mode seen in the FM-120. There may be a few cloud droplets or ice crystals that are large enough to shatter but, as the reviewer has pointed out in several comments, such particles should be far outnumbered by the main droplet mode. Thus, they would have to be very large indeed to shatter into enough fragments to significantly affect the measured cloud residual distributions (bearing in mind that fragments still need to exceed the GCVI cut-size and be aligned with the streamlines in order to be sampled). Cluster 1, which is linked to cold temperatures has no clear clear droplet mode, is therefore likely to be influenced by snow shattering artefacts. Cluster 2, on the other hand, has a clear droplet mode and has a temperature distribution (see Fig. 10f in revised manuscript) that is not appreciably different from those of clusters 3–5. In addition, its shape is not so much flat as bimodal with two broad modes. However, it is possible that the influence of artefacts gives the size distribution a flatter appearance — when the cluster analysis is repeated without the most suspicious data points (new supplement Fig. S12) the bimodality can be seen more clearly, which supports this hypothesis. For all clusters, we now provide separate scatterplots of number concentration of residuals versus cloud particles. In this way, the reader can easily assess how much data of the individual clusters is potentially affected by artefacts. See also response to Reviewer 1, major comment 1 (for text from the revised manuscript relating to these issues).

**L. 464-466:** It is very good to point out the difference of CCN and residual particle measurements, but this make only sense when the residuals of liquid clouds are subject of the discussion. Once ice particles occur there is of course a difference since no INPs can be measured in a CCN counter.

Agreed. With the restructuring of the manuscript, we now show the annual cycle of number concentrations separately for liquid-dominated clouds and cold clouds. We have modified the text to point out that the comparison to the CCN counter studies can only be made for liquid clouds: "Since these other techniques cannot measure INP, we will only compare them to our liquid cloud-dominated data. "

**L. 467:** it is clearer to write "long-term CCN data sets".

Agreed, we have changed the text accordingly.

**L.479-482:** Despite all the caution the authors has exercised in the data analysis by leaving it open whether the occurrence of Aitken mode is an artefact or not, they now claim that the observed Aitken mode particles play an important role for Arctic clouds. But the data situation does not allow for this.

This statement has been removed.

**L.486-490:** This explanation is incomplete. It explains why smaller particles could become activated, but it does not mention that still the larger ones become activated first. Thus, the accumulation mode particles should be seen much more pronounced, especially because they are activated with 100% and, the 20-30 nm particles only with few % (e.g. Schwarzenboeck et al., 2000).

We agree that the explanation was incomplete as far as the CCN-limited regime goes. We have completely rewritten the conclusions and this text has been removed. Schwarzenboeck et al. (2000) analysed data where the aerosol size distribution was dominated by accumulation mode aerosol or accumulation mode aerosol contributed significantly to total aerosol number concentration. For these conditions the reviewer is correct with this statement. However, in our case we observe approximately one order of magnitude lower aerosol concentrations and relatively larger contribution of Aitken mode aerosol. Unfortunately we do not have information about the chemical composition, but a plausible explanation can be also that chemistry related hygroscopic properties can play much more important role than in the case of observations from Puy de Dome reported by Schwarzenboeck et al. (2000), and in this context the argument that only few percent of Aitken mode can be activated may not hold.

**L. 495-499:** Again, the argumentation her is only partly true. Findings from mixedphase and liquid clouds are mentioned in one sentence, so it is not clear to which cloud type the authors attribute the small Aitken mode particles to. Mertes et al. (2007) indeed measured similar residual size distribution in mixed-phase clouds but with an ice selective CVI, which is not the case in this study. There should be a difference in the measurements since the droplet residuals should be involved here but not are not in the cited study. Consequently, the number of residuals is much less in Mertes et al. (2007), because the CVI was designed to remove large cloud particles that are mostly responsible for artefact measurements. Moreover, the activated fraction showed a strong increase with size, whereas in this study this is not seen (cf. Fig.S7a, Fig.3c). Schwarzenboeck et al. (2000) found droplet activation down to 25 nm particles but at the same time an increase of the activation fraction to 1 above 100 nm, which is totally different to the Aitken mode particle observation in this study and is thus not a confirmation of the findings here.

In the revised version of the manuscript, we discuss liquid clouds separately from mixed-phase and ice clouds, so the citations no longer appear together in the same sentence. The conditions with respect to aerosol number density, shape of size distribution and environmental conditions (ambient T close to or above 0°C) in Schwarzenboeck et al. (2000) are comparable with our spring and summer conditions. Here our data and analysis agree well with their work. They did not present any observations for cases with ambient temperatures well below zero, nor at aerosol concentrations as low as those we observed during periods when the Aitken mode dominated the cloud residual size distribution. With regard to the observations of Mertes et al. (2007) using an ice-CVI, it is true that this instrument removes all larger cloud droplet. However, it is interesting to note that our cluster 1 does not show a droplet mode and as such we suspect the cloud to consist of pure ice (and snow) and as such might be comparable to the observations of Mertes

et al. (2007). It is therefore still valid to make the link of cluster 1 to previous observations of residual size distributions resulting from ice particles.

**L. 500-507:** It is interesting that the authors do not find an explanation for the occurrence of Aitken mode particles in the Arctic winter. So how should these particles then play an important role in wintertime cloud formation?

We observed Aitken mode particles, and do provide a discussion on some plausible sources (e.g. see next comment). In addition, we highlight that future work on the origin of these particles is needed (e.g. using more sophisticated aerosol mass spectrometry measurements, see next comment). Studying the origin of these small particles is not within the scope of this paper.

**L. 508-513:** Secondary ice might be possible and would create small residuals, but the number concentration of ice particles with respect to cloud droplets in mixed-phase clouds is still very low (at least 1 order of magnitude), so that these residuals would not dominate the size distribution. And the Fm-120 would see these particles too, which seemed not to be the case as it was pointed out by the authors. Once more, only CCN but not the insoluble INPs would undergo fragmentation.

This has been addressed in the revised version of the manuscript. See earlier answer to major general comment C, as well as response to Reviewer 1, major comments 1 and 2.

**L.513-524:** Many speculations which are not supported by additional measurements that would have been essential for such a study or using one of the ice selective CVI inlets.

This is part of the discussion and various hypotheses are presented. The shape of cluster 1 has been observed by CVI measurements within cirrus clouds (Seifert et al., 2003) and by ice-selective inlets (Mertes et al., 2007). It should be clear to the reader that this discussion of the various possible scenarios is done in the subjunctive form. As mentioned above, we have added further analysis and changed the discussion. The increased contribution of ice is also supported by the independent Cloudnet observations (see now Fig. 11). In addition, we have now compared our observations the study of Verheggen et al. (2007), which we missed in the previous version of our manuscript. Indeed, our observations of the activated fraction of mixed-phase clouds agrees surprisingly well with the observations of mixed-phase clouds at the high-alpine site Jungfraujoch (see their Fig. 7 and Fig. 2 below) giving further support to our findings. Verheggen et al. (2007) derived similar activation ratios although using observations from an interstitial and whole-air inlet, while our findings are based on a more complex sampling via the GCVI inlet. Still, it is surprising that we find a similar shape of the activated fraction with a similar maximum at 0 °C. Verheggen et al. (2007) had more sophisticated cloud probes available and could link this decreasing activation ratio with decreasing temperature to the increased contribution of ice crystals which is consistent with the Wegener-Bergeron-Findeisen process taking place inside mixed-phase clouds. This is consistent with a decreased amount of activated accumulation mode particles seen in cluster 1. For our study, it would have been favourable to perform more detailed measurements e.g. with ice-selective inlets and especially with more advanced cloud probes that are capable of distinguishing between cloud droplets and ice crystals. The findings of this work were also one motivation to deploy more sophisticated cloud probes within our current NASCENT campaign (https://www.aces.su.se/research/projects/the-ny-alesund-aerosol-cloud-experiment-nascent-2019-2020/). Here, collaborators from ETH Zürich have deployed holographic cloud probes (e.g., Henneberger et al., 2013) at Zeppelin Observatory and on a balloon to investigate the physical properties of mixed-phase clouds.

[Figure]

Figure 2: **Dependence of activated fraction on temperature.** The activated fraction is here defined as the ratio of cloud residual and total particle concentrations integrated above 100 nm diameter. **Left:** Our observations (ratios are based on 30 min average concentrations). Solid and dotted lines show median and mean values, respectively, and shaded areas indicate the 25[th] to 75[th] percentile ranges. **Right:** Figure 7 from Verheggen et al. (2007). Note that Verheggen et al. (2007) determined the activated fraction using observations behind a whole-air and interstitial inlet, while our observations are based on measurements behind the GCVI and whole-air inlet. The left panel has been added to the SI of the revised manuscript.

**L. 525-528:** What are the conditions that are denoted as GCVI malfunctioning and how are they recognized?

We have added a new section describing the data treatment in detail. See clarifications above.

**L. 529-530:** A more scientific procedure would be to remove all data points that could have been biased by potential artefacts and use the "unsuspicious" measurements/results only. Much stronger and first of all justified statements would have been possible.

First of all, clear outliers due to instrument malfunctioning and local pollution were removed (see data treatment section). In accordance with this and the previous comments, the revised version now addresses the previous mentioned shortcomings. We now also perform the analysis for cold and warm clouds separately. In addition, the cluster analysis quantifies the contribution of the different shapes of the residual size distribution. Cluster 1 and 2, which might be the most suspicious ones to the reviewer, are most likely linked to ice. As recommended by the reviewer, we repeated the cluster analysis by removing the data points with largest disagreement between FM-120 and GCVI (new Fig. S12 in supplementary material), but still the main messages almost stay unchanged. Nevertheless, we still address the issue of possible artefacts more clearly in the revised version (as they can happen with almost all cloud in-situ instrumentation). Ice crystals need to reach a certain size for shattering to be likely, and we thus believe that some of the small particles we observe may be the result of snowflake shattering in the wind tunnel of the GCVI (as the reviewer also suggested as a possibility in an earlier comment). This discussion has been added to the revised manuscript. It would also explain why the concentrations measured by the FM-120 are much lower, since the snowflakes should be larger than the last channel in the fog monitor.

**L.533:** Even if one would assume that the small residuals stem from non-shattered cloud particles, the authors always leave it open if these residuals are released from droplets or ice particles. If they are originating from secondary ice (which is to some extend the most likely explanation), these residuals do definitely not contribute to the formation of mixed-phase clouds. The used expression is even in this case a total exaggeration.

We have rewritten the conclusions, so the phrase referred to by the reviewer no longer occurs (see response to Reviewer 1, major comment 9 for text). In the revised version of the manuscript we have addressed caveats relating to the secondary ice hypothesis more clearly, and clearly stated that the residuals from secondary ice particles do not represent cloud nuclei (see response to Reviewer 1, major comment 2).

**L. 538-539:** This is a trivial statement that is valid for all clouds around the globe and could be therefore deleted.

We have deleted this sentence in the revised manuscript.

**L. 540:** Before in this study, the separation of Aitken mode and accumulation mode was defined at a diameter of 60 nm, now the authors introduce a size of 100 nm as the lower limit for accumulation mode particles. This is not consistent and has to be harmonized. Since Cluster 1 and 2 are more suspected to present artefact sampling, the remaining Clusters 3,4,5 are accumulation mode particles according to the first definition, although they are closely to the left edge of the size distribution of the accumulation mode. Hence, it might be advantageous to speak of particles smaller 100 nm instead of declaring that these are Aitken or accumulation mode particles.

We did not intend to say that 100 nm was the lower size limit for the accumulation mode — we simply mentioned that the use of a fixed diameter (e.g. 100 nm) to determine which aerosol particles act as CCN is a common approach used in a number of aerosol–cloud interactions studies, and we wanted to highlight that this approach is not suitable in the Arctic.

In the revised version of the manuscript, we do not introduce the 100 nm discussion, since the figure that was related to this has been removed in favour of figures relating to the cluster analysis, based on the recommendations from Reviewer 1.

**L. 543:** It puzzles me that the authors claim that the may see the features of a CCN limited regime in their dark period data. Looking at Fig.S7a or Fig.4b it is obvious that the are many particles present in the total aerosol (interstitial + residual) that are not activated. But CCN limited means that there are no more or very few particles to activate, i.e. that there are hardly any interstitial particles left, which is not the case here.

Given the previous comments by both reviewers, we have removed this part and rewritten the conclusions (see response to Reviewer 1, major comment 9 for text).

**L.544-547:** This description is a possible scenario but certainly not a conclusion of this study. Thus, it should be removed or considerably reworded to bring it in the right context to the own conclusions.

We agree and as such have removed this part. As summarised above, we have completely rewritten the conclusions (see response to Reviewer 1, major comment 9 for text).

**Fig.2:** Here mean and median are the solid and dotted lines whereas it is vice versa in all other figures. This should be made consistent.

This was a mistake in the caption, thanks for pointing it out! It has now been fixed.

**Fig.2:** according to the text it is not the measured FM-120 concentration but the GCVI sampled cloud particle concentration derived from the GCVI transmission efficiency, correct? At least that is the explanation in the corresponding text. If so, the figure caption needs to be corrected

The original caption already included the line "The transmission efficiency of the GCVI inlet (Shingler et al., 2012) has been included in the calculation of the cloud particle number concentration in all panels." We have replaced the phrase "cloud particle number concentrations as measured by the FM-120 fog monitor" with "cloud particle number concentrations derived from the FM-120 fog monitor measurements" to make it more clear.

**Fig. 3 and Fig.4:** brackets are use in this way "( ]". In case this has no special meaning, this should be made consistent. Furthermore: The mean, median, percentiles are not mentioned in the figure caption which should be added.

"( ]" is standard interval notation indicating whether the endpoint is excluded (soft bracket) or included (hard bracket) in the interval. In addition, we added to the caption of Fig. 3 and 4 (see also comment by reviewer 1): "The whiskers extend to the farthest points that are within 1.5 times the interquartile range from the nearest quartile. Points that fall outside the whiskers are not shown." The mean is not shown since this standard box plot shows the median and 25th and 75th percentiles (quartiles), which should be obvious from the revised figure caption.

**Fig.S4:** figure caption: meet instead of meed.

This comment refers to a figure panel which has been removed in the revised version of the manuscript.

**Figures in supplement:** Several times the straight lines are named "full", whereas "solid" is the more common term and used in all other figures (supplement and manuscript). This should be made consistent.

This has been fixed.

**3 Further changes**

- We now use the total CPC (behind GCVI) for the comparison with the FM-120 (before the integrated number concentration from the DMPS scan was used). The results have improved slightly.

- We moved Figure S11 from the SI to the main text (now Figure S10). We have added the corresponding average cloud particle concentration to panel d.

- As suggested by reviewer 1, we also moved Figure S7 from the SI to the main part (now Fig. 12). To be consistent with the other size distribution plots, we show the distributions on a log-y-scale. We also added median and percentile values.

- In addition, we have removed Figures 6, S8 and S9. These figures were replaced with the new Figure S8, which shows the monthly averages of cloud residual size distributions for all, warm and cold clouds separately.

- We removed the box plots (Figure 7 and S10) showing the contribution of small particles (box plots).

---

## Referee Report (RR1)

The revised manuscript "A long-term study of cloud residuals from low-level Arctic clouds" has largely addressed most of the comments of the reviewers. However, a few comments remain not be thoroughly addressed. And I have some additional comments. I recommend the publication of this manuscript in ACP if these comments can be addressed.

- 1. Both reviewers mentioned making comparison for the particle number distribution in cloud and outside cloud to ensure the measurement quality from whole-air inlet (Reviewer 1 comment 3 and 7, Reviewer 2 minor general comment 6). Cloud particles may also experience possible additional losses in the whole-air inlet too due to e.g. the different size distribution compared with ambient aerosol particles. But the authors decided to not follow the comments because the authors argue that "appearance or termination of the cloud event can also be caused by a change in air mass with differences in aerosol properties." In my opinion, the information on the particle loss of whole-air inlet for cloud droplet sampling is necessary because if there were significant difference, it would affect the ratio of cloud residuals to total particle concentrations shown in Fig. 6 and Fig. 7. Although the authors showed a new comparison of cloud residual concentration vs. total particle concentration has no artefacts/free of loss and of ensured quality.
- 2. A related comment to the comment 1 is that the description of the whole-air inlet is lack, especially the important role of the data for this study. One cannot expect readers to follow the setup just stating "fulfils the World Meteorological Organization guideline". For example, how is the inlet heated, and to what temperature? Has this setup been validated previously regarding the particle loss? If so, references could be cited. More details would be helpful here.
- 3. In the new structure, for Section "3.2.1 Warm clouds" and Fig. 6, are all these data for temperature < 0 C? If so, it would helpful to clarify explicitly in both the text of 3.2.1 and caption of Fig. 6.
- 4. Section 3.2.2 heading "Cold clouds" seems not to be consistent with the scope of discussion that it covers. It mainly discusses the influence of temperature rather than cold clouds. Maybe a different heading would work better.
- Some responses to reviewers should be incorporated to the revised manuscript as they will be helpful to other readers. For example, the response to the comment D (Pg 17 of the authors' response file), comment "L. 158-159" (Pg 23 of the authors' response), comment "L121-122" (Pg22), comment.
- 6. Some "new" statements are introduced in the "conclusions" part while how they are drawn is not straight and completely clear in the main text. For example, L663, "The cluster analysis of cloud residual size distributions showed a D50% dependence on updraft for liquid clouds (clusters 3–5)." This finding is not explicitly drawn, although the authors show that D50% depends on updraft velocity and updraft velocity of clusters 3–5 are different. A direct way to draw such a finding would be a figure showing the D50% dependence on updraft for data belonging to clusters 3–5.

L666-667, the finding "A clear relationship between a decreasing total particle number concentration and a decrease in D50% was also observed" is also not directly drawn in the main text. In L537-538, a figure of D50% vs. particle number concentration and

updraft velocity is needed as the current presentation of the trend e.g. D50% values is hard to follow.

L668-670, it is not clear which data and discussion these statements are based on since the data were not discussed by April-October or November-March.

7. The abstract has not fully reflected the changes in the focus of the revised manuscript as much of the findings in the main text and conclusion are missed in the abstract.

Minor comments:

- 1. L523 and Fig. S9, why different clusters (2-5) show different slopes? Shouldn't it be that all cluster have similar slopes if the transmission efficiency of GCVI (size-dependent if I understand correctly) are correctly applied and the GVI is free of artefacts?
- The caption of Fig. 2 can still be misleading if one reads separately from the main text. I suggest rewriting "cloud particle number concentrations derived from the FM-120 fog monitor measurements (red)" as "*corresponding* cloud particle number concentrations derived from the FM-120 fog monitor measurements *and transmission efficiency of GCVI*'.
- 3. L445, Fig. 6 or Fig. 7?

---

## Author Response (AR2)

**Reply to reviewers of the manuscript "A long-term study of cloud residuals from low-level Arctic clouds"**

Karlsson et al.

March 22, 2021

We thank both reviewers for their helpful and constructive comments. As suggested by the reviewers, we have implemented an individual correction factor of the CVI sampling efficiency based on the concurrent cloud particle measurements. As expected, the overall results were not significantly affected1 except for the few cases at very cold temperatures when the fog monitor did not sufficiently measure cloud particles. In addition, we added two case studies to the appendix of the manuscript as requested by the reviewer. With these last changes, we hope to have sufficiently fulfilled all requirements and requests. More detailed replies are given below.

**1 Anonymous Referee #2**

General comments:

With respect to the first version, the actual manuscript has substantially improved. Especially the reduced importance of small residuals with regard to all residual sizes is very good for the work.

However, there remain some critical comments, which are a summary of the main issues of the "specific remarks" below. These specific remarks are more or less only related to content, since language/wording and formal points are very good implemented.

1. Of course, the statistical treatment of the data is unique and very helpful for in-cloud CVI measurements. Nevertheless, two short case studies, one for warm one for cold clouds, where the real physical closure of cloud particle, aerosol particle and residual number concentration are presented would very much increase the confidence in the empirical approach and methodology. Moreover, at least some of the many, many open questions about small residuals in mixed-phase (cluster 1) and warm clouds (cluster 2) could be answered this way or at least some speculations could be ruled out.

The strength of our presented work lies in the availability of a large data set (>1700 hours of in-cloud data), which can be studied using various statistical methods. To address the reviewer's request and thus convey our confidence in our analysis, we have added two case studies in the appendix to the main manuscript. The first case is a straight-forward to analyse liquid cloud case (cluster 5), while the second case is a cluster 1 case that illustrates the difficulties in interpreting data from snowy, ice or mixed-phase clouds. These case studies complement the already

<sup>1Most figures (very slightly) changed due to the individual-based correction, they are marked with a purple frame in the revised version with track-changes. New figures are marked with a blue box.

existing detailed discussion of our overall observations (with all pros and cons).

As mentioned in the last reply letter, first results from the NASCENT 2019-2020 campaign (see website here) give us now further evidence that clouds at Zeppelin with cluster 1 occurrence (measured by the CVI) are linked to secondary ice processes (Pasquier et al, in prep.). In addition, we would like to mention that the reader can access the data from the Bolin Centre Database and can perform their own individual desired analysis later on.

2. I am still not a fan of the conclusion that say that the measured data could be an artefact or a real physical cloud process or a mixture of both like it still is in the discussion of the small residual particles. In my understanding of research one should avoid or at least quantify artefacts to be sure that they do not exist or at least do not significantly impact my measurements so that the data can then be scientifically interpreted. If this is not the case such data should not be submitted for publication. In this special case it is even more surprising and disappointing, since it is well-known that measurements with "standard", i. e. non-phase segregating, CVI inlets are subject to large artefacts and that therefore several highly sophisticated phase segregating inlets have been developed exactly due to that reason. It is my hope that the authors further reduce their claim that the observed small residuals are related to secondary ice processes in the cloud, because their inlet system is unusable make such a statement.

We need to disagree with the reviewer here. We are grateful for the many constructive suggestions the reviewer has given us (see detailed replies below), but we consider it good scientific practice to discuss potential hypotheses that result from evidence-based findings. We present evidence, both by own analysis and by comparing to literature, that our findings are indeed plausible. At the same time we make the reader aware that certain artefacts can not be excluded and identify/discuss those in great detail. Showing doubts and negative results is an important part of scientific progress. We present and discuss our data and findings in a clear and very balanced way, and no unjustified conclusions are being drawn. Nevertheless, to be more precise, we removed "frequently" from the abstract and are now stating that secondary ice processes is a "potential" explanation for our observations, while clearly stating that artefacts are potentially involved as well. We also added to the abstract and to the introduction the following sentences: "... and the potential contribution of sampling artefacts is discussed in detail." and "Since this is the first long-term deployment of a GCVI inlet, globally and in the Arctic, emphasis will be put on the evaluation of the GCVI inlet sampling efficiency and a detailed discussion of the potential contribution of artefacts during mixed-phase cloud conditions."

We are of course aware that other types of inlets exist that can e.g. be used to uniquely sample ice crystals, which we also address in our manuscript. However, these were not operated or available for our project. Moreover it is unclear if they can be operated on a long-term basis and under the harsh environmental conditions found in the Arctic. In addition, those inlets are unfortunately also not free of artefacts. Last but not least, we would like to state that the here used ground-based CVI is now a commercially available instrument, and as such we find it especially important to present all observations and drawbacks to the reader in a comprehensive and balanced way.

**3.** I do not know of course what kind of measurement results the authors expected during the long operation period of the GCVI, but it is a pity that the whole set of instruments was not better selected. Beside a phase segregating CVI system, this mainly means a dedicated sensor for droplets (concentration, size distribution, LWC) and ice particles (concentration, size

distribution, IWC). Especially, sensors that are able to measure LWC and IWC would have been much better to clarify the immense amount of speculations and would have been much better as a cloud detector compared to the visibility measurement. In the conclusion, the authors promise to repeat this kind of measurements with an improved set of instruments. This is often done in conclusion chapters without any future action but I hope this time the authors take it serious due to our scientific curiosity about clouds.

Long-term observations in the Arctic at this complexity are a complicated and expensive endeavour which is often dependent on international collaborations. Indeed, our Japanese partners have had installed cloud probes that could have delivered more information about cloudphase. Unfortunately this particular instrument was not operational during our period. We cannot go back in time and change the instrument set-up, so the best we can do is expand the set of instruments we have and hope that future studies can shed light on the issues where our results were inconclusive. Within the course of this work, we have clearly identified the need for more detailed cloud microphysics observations. Within the recent NASCENT campaign (https://www.aces.su.se/research/projects/the-ny-alesund-aerosol-cloud-experimentnascent-2019-2020/), several additional instruments have already been installed at Zeppelin Observatory that included a cloud holographic imaging probe (Henneberger et al., 2013). Some of the co-authors are already involved in studies that plan to expand on the analysis presented in this paper. As mentioned above, first preliminary findings seem to indicate the link of cluster 1 to the presence of secondary ice but the detailed analysis is still ongoing.

4. Beside listing all the advantages of the possibility long-term ground-based residual measurements in contrast to short-term measurements with aircrafts, it would be fair to mention the disadvantage in contrast to airborne measurements as well. This means the restriction to clouds with soil contact and orographic effects (which is a general problem and not one at Mt. Zeppelin only) whereas "unbiased" clouds can be reached only by aircrafts. This should be a second point about the representativeness of the measurements together with the one in the site description concerning the representative location of Mt. Zeppelin/Ny Alesund for the Arctic.

We agree and have added to the introduction the following: "Aircraft measurements using CVI inlets have the advantage of recording profiles of undisturbed and elevated clouds but are very expensive and limited in time, while ground-based CVI observations can cover longer time periods (seasons to years) but are potentially affected by the surrounding orography."

5. It is known that the sampling efficiency of CVI systems, and particularly ground-based CVIs, changes with wind and cloud microphysical properties, i.e. it could even change within one cloud event. Therefore, it is nor clear why the authors used only one value to correct for all cloud events included in this study. In principal, each measured residual particle size distribution should be scaled/corrected with the actual CVI sampling efficiency. This is not meant as a harassment, but could indeed lead to different shapes of the averaged residual particle size distributions for the temperature or month intervals (Fig.6b, Fig.7b, and Fig.S8). Consequently, that may also change the complete cluster analysis and the respective discussion. Fig.3a and Fig.5a are not real counter arguments to that because the frequency distribution should be much broader when plotting the ratio value for each residual particle size distribution. Thus, the authors are called to address this issue.

We have reevaluated this part of the analysis and have decided to follow the reviewer's suggestion and implement an individual correction factor for each cloud residual size distribution. We have updated Sect. 3.1.1 and 3.1.2 (now combined into 3.1.1) and other discussion accordingly. When doing this, we also decided to try to correct for particles lost due to the GCVI cut-size (before we only derived the correction factor above the cut-size) and would like to note here that the average of the new (individual) correction factors is now around 2.2. To our own surprise, this agrees perfectly with the accumulation mode comparison for liquid clouds and gives further confidence in our approach and overall results.

As expected, due to the statistical approach, most figures are dominated by the average correction factor and therefore do not change much when we use individual correction factors (see updated figures). Figures 7,8, S8 (now 6, 7, S9), i.e. the figures where data are segregated based on temperature, are the only figures that show a significant change for specific lines at cold temperatures. Since an individual correction depends on how well the fog monitor works, it may introduce an additional bias to the results during conditions when the fog monitor might have issues (e.g. sampling of large cloud particles or ice, or at high wind speeds). We still apply the individual factor, but bring up this potential drawback of that approach in the new appendix (Case II discussion).

Concerning the cluster analysis, we would also like to clear up a misunderstanding by the reviewer. The cluster analysis is not influenced by the choice of correction factor at all, since it is based on normalised size distributions. The correction does of course play a role in the non-normalised size distributions, but because the correction factor (individual or otherwise) is just a scalar applied evenly to each cloud residual size distribution bin, it has no effect on the shape of the cloud residual size distribution.

6. Three times the authors refer to other studies to affirm similar results with their own study (L.569: Verheggen et al. (2007), L. 575-578: Seifert et al. (2003), Mertes et al. (2007)). On the other hand, these studies also show clear differences to the actual study which are not addressed at all (details about that are given in the specific remarks). This is surely not intentional, but needs to be included for a complete scientific discussion.

We have added a couple more sentences that mention how these studies differ from ours (e.g. different background aerosol size distributions and activation ratios). See responses to the specific remarks below.

7. The size distribution of cluster 2 is in this revised manuscript attributed to droplet residuals. But its broad shape is very unusual for a CCN size distribution. This needs to be related to the shape of the total aerosol size distribution simultaneously present at the site. This should be examined during the text passage where cluster 2 is discussed. Moreover, the maximum possible supersaturation should be estimated in order to prove that those small particles can be indeed activated. In principal this should be related to a large updraft velocity, but this seems not to be the case according Fig.10 b.

Cluster 2 is indeed a special cluster that can not be solely described by pure liquid activation. The overall averaged size distributions as such deviate from what would be expected by assuming Koehler theory (see Fig. 12 and the discussion within the manuscript). As described in more detail below, there are several studies now that give clear evidence that 25-80 nm particles can be activated for liquid cloud cases (see e.g., Bulatovic et al., 2021) and as such we see no need to perform own calculations. However, we have added a few more sentences to the revised manuscript that describe the total particle size distributions of cluster 2: "The cluster is bimodal, but relatively broad and flat. While the same approximate size modes are present

in the total particle size distribution, the average cluster 2 cloud residual size distribution has a less pronounced minimum and slightly lower concentration of accumulation mode particles than the total particle distribution. The shape of cluster 2 could perhaps also be influenced by ice processes (i.e. not all cloud residuals correspond to CCN), or the shape might be affected by evaporation of volatile compounds from the accumulation mode particles. However, this cannot be confirmed without size-resolved chemical composition or volatility measurements, which were not available for our period."

I would have preferred to completely forego the presentation and discussion of the residual particle size distribution measured during mixed-phase conditions (mainly the cluster 1 discussion with 8 % of the time) and I still have some doubts about the actual existence and interpretation of cluster 2. However, this is certainly not enough to reject this manuscript presenting a unique long-term cloud particle residual data set at an important place on earth with respect to the role of clouds to arctic amplification. The authors seem to stick to these points, which is acceptable in the way it is implemented, although it does not strengthen the manuscript to my opinion. However, the issues raised in the general comments and the specific remarks in this review must be responded and where appropriate included in the manuscript for publication.

**Specific remarks**

**L.59:** beside secondary ice one should at least also mention impaction scavenging as an in-cloud process that could result in residuals that are not identical to the original CCN or INP.

We agree, we have added this to the sentence so it reads "[...](e.g. impaction scavenging or secondary ice [...])".

**L.60-64:** Already here it would be fair to mention the disadvantage of ground-based or advantage of aircraft in-situ cloud measurements as already mentioned in my general comments.

We added the following sentence here: "Aircraft measurements using CVI inlets have the advantage of recording profiles of undisturbed and elevated clouds but are very expensive and limited in time, while ground-based CVI observations can cover longer time periods (seasons to years) but are potentially affected by the surrounding orography."

Figure 2b,c; Fig.3a,b; Fig.4a; Fig5a,b: Maybe it is given overseen by me, but it is important to indicate the averaging time of the data points.

In almost all cases, data were averaged to the resolution of our main instrument (i.e. the DMPS behind the GCVI). We apologise that this was not stated more clearly. We have added the following in the introduction of the Results section "Unless otherwise stated, all data presented in this section have been averaged to match the time resolution of the cloud residual size distributions measured by DMPS 1 (i.e. 5–7 min averaging time, cf. Tab. 1). When making simple comparisons to DMPS 2, which has a lower resolution than DMPS 1, we used all simultaneously measured data (i.e. overlapping DMPS scans, without repetition of data points). In the cases where a one-to-one data point comparison was necessary, both DMPS data sets were downsampled (usually to 30 min averages).".

As for Fig. 4, it uses 30 min averages. This is stated in the text but had been forgotten in the caption, so we added "30 min mean values of DMPS 1 and DMPS 2b data were used for this analysis" to the caption.

**L.383-389:** Here the argumentation of the authors is not correct. At this text passage clouds are discussed with a cloud particle concentration of 1 cm-3 and where the cloud particle concentration is below the cloud residual concentration. These must be more or less totally glaciated clouds, where most cloud particles (ice crystals) are rather large. This means, the very likely shattering of these ice particles in the GCVI wind tunnel would substantially increase the cloud residual but not the cloud particle concentration. This is supported by many airborne CVI measurements in ice clouds.

This is a good point. As discussed in the text, cloud particles would need to exceed ~ 70  $\mu$ m to be likely to shatter in the wind tunnel. We believe the reasoning still stands that for mixed-phase clouds, where such large crystals would be far outnumbered by droplets, potential shattering would not significantly influence the cloud residual concentrations. We added "mixed-phase" into the sentence to clarify ("[...]in most mixed-phase non-precipitating clouds, the concentration of large cloud particles is much lower than the total cloud particle concentration[...]").

If the clouds are fully glaciated, we agree with the reviewer that artefacts may come from cloud ice as well as precipitating ice. We changed the end of the paragraph to: "Therefore, the magnitude of the concentration difference we observe suggests that precipitating particles (e.g. snow) are a more likely cause than large cloud droplets or ice crystals. If, on the other hand, we are sampling a fully glaciated cloud consisting of large ice crystals, then shattering artefacts could come from cloud ice as well as precipitating ice."

Having said that, the only instrument we have to try to verify the cloud particle size and concentration is the FM-120, which has an upper size limit of 50  $\mu$ m. In other words, if the pure ice clouds should consist of very large crystals, those crystals might not be detected by the fog monitor. The visibility comparison, while not perfectly valid for ice clouds, did show that the fog monitor may have issues detecting the particles. However, cloud particles that are so large would perhaps also be too large to enter the GCVI (without artefacts). All in all, with our instrumental set-up we cannot clearly disentangle these effects.

**L.413-416:** This analysis is very crude. The match of the total aerosol and cloud residual size distribution, which is more or less the exact and individual knowledge of the GCVI sampling efficiency for each cloud included in the respective data set, has to be taken into account and not only a correction factor of 2 for all clouds. This is important, because the value of the D50% is very sensitive to the correct quantitative relation of total particle to residual particle size distribution. For a first guess that can be done by normalizing the plateau between 100 and 300 nm to 1. Doing so, one would see a trend in Fig.6c, i.e. a decrease of D50% for higher updraft velocities. This would be the expected behaviour, because this means higher supersaturation with the capability to activate smaller particles. Thus, this approach leads to contrary conclusion with regard to the actual text and should be at least commented by the authors, since this is a crucial point for the description of the properties of warm arctic clouds.

Because we are deriving the  $D_{50\%}$  values from average size distributions, the behaviour is dominated by the average correction factor. Using an individual correction factor does not significantly change the relative behaviour of the curves in this figure. Nevertheless, we have decided to follow the reviewer's suggestion and implement an individual correction factor (see response to comment 5 above and comment L.425-426 below for more details).

We agree with the rest of the comment, and have made a figure where the ratios have been normalised to 1 at 200 nm, see Fig. 1a. Figure 1b shows the interpolated  $D_{50\%}$  values from the normalised ratio for each updraft interval (together with the size range of the diameter bin in

which the value falls). This figure is included in the supplementary material and discussed in the main text of the revised manuscript as follows: "The  $D_{50\%}$ , defined as the diameter where the ratio is 0.5, ranges between approximately 57 and 75 nm in Fig. 5c. One can attempt to account for the aforementioned sampling issues by normalising the plateau of the ratios to 1. If this is done, the  $D_{50\%}$  ranges from 50 to 78 nm and shows a decreasing trend with increasing updraft (Fig. S8). This behaviour is expected from a cloud physics point of view, where higher updraft velocities can produce higher supersaturation levels which, in turn, allows smaller particles to activate."

Figure 1:  $D_{50\%}$  dependence on updraft. a Ratio of mean size distributions, i.e. cloud residual concentrations divided by total particle concentrations. The coloured ratio curves have been normalised to 1 at 200 nm. The grey dotted curves represent the position of the nonnormalised ratio curves (Fig. 6 in main manuscript). b Interpolated  $D_{50\%}$  for each updraft interval (i.e. the diameter at which the ratio in **a** is 0.5). The "error bars" represent the width of the diameter bin within which each interpolated value falls.

**L.415:** All observed updraft velocities are influenced by the orography independent of the amount of the updraft. Without the orography the updraft velocity would be different for an arctic cloud. Regarding in addition the point before, one would find an orographic influence on the D50%, which is again in contrast to the conclusion in the manuscript and needs to be further considered by the authors.

This is correct; however, updrafts close to  $1 \text{ ms}^{-1}$  are plausible for stratiform and stratocumulus clouds (and what physical forcing causes the updraft velocity is not really important). We have removed the sentence from the revised manuscript, but added the following where we discuss the  $D_{50\%}$ : "Updraft velocities in marine stratiform clouds are typically below  $1 \text{ ms}^{-1}$  (Zheng et al., 2016) and hence higher updraft velocities could be indicative of local orographic effects and may not be representative for Arctic clouds in other areas. Excluding the bins with updraft >  $1 \text{ ms}^{-1}$  gives a  $D_{50\%}$  range of 58–78 nm (Fig. S8)". See also the response to the comments above and below with regards to the  $D_{50\%}$ .

**L.418:** Does it mean the enrichment factor is uncertain and only assumed? This should be clarified and made clear in the manuscript.

The enrichment factor is not only assumed; it is calculated based on the flows and geometry of the inlet (this is already mentioned in Sect. 2.2.2). There are technically always small uncertainties in e.g. flow measurements, which is why "and enrichment factor" was included in this sentence to begin with, but in the grand scheme of things such uncertainties are likely to be negligible. We realise now that mentioning them only adds confusion, so to address this comment we simply removed the phrase "and enrichment factor" from the sentence in question.

**L.425-426:** This is exactly what is brought up in the last two points and emphasizes that the GCVI sampling efficiency has to be determined for individual cloud events.

Following comment 5 above, we have implemented an individual correction factor for each cloud residual size distribution scan. This, however, does not significantly change the behaviour this comment refers to. The Shingler et al. (2012) transmission efficiency we use for the corrections can only account for what happens when the cloud particles enter the inlet/wind tunnel, but it could be that high updrafts or wind speeds prevent some cloud particles from actually entering the wind tunnel, and this is the kind of sampling issue that is meant in the text. We have added a parenthesis "(e.g. if the winds make it more difficult for the cloud particles to enter the wind tunnel)" to clarify this.

**Fig.7:** The first temperature bin from -8 to -21 C is much too broad. In this temperature range the cloud phase could change from supercooled to totally ice including all stages of mixed-phase conditions. The reason for this broad range is most likely a statistical one. It is not known whether most included cloud events are closer to -8 or -21 C and dominate the presentation and interpretation in and of Fig.7b and 7c. This should be presented in a more differentiated way.

It was indeed chosen for statistical reasons; however, we agree with the reviewer and have therefore split this bin to mitigate the issue. As can be seen in Fig. 2 below, the two "parts" of the original bin show very similar behaviour. We chose to put the additional bin boundary at  $-12^{\circ}$ C because of the distribution of data points in the original bin (see Fig. 3 below).

---

## Author Response (AR3)

28 April 2021

Dear editor,

We thank the referees for their last round of comments. Concerning the whole-air inlet we have made it clearer that the inlet was exactly built according to the technical drawings given by the World Calibration Centre for Aerosol Physics (WCCAP) and that it is technically identical to the inlet by Weingartner et al (1999). We refrain from adding the sentence "the inlet has not been experimentally validated for particle loss" to our manuscript because this is an uncommon procedure in the world of ambient aerosol sampling and technically difficult to perform. We have no reason to believe that significant particle losses have occurred (see previous review comments). Therefore, adding such a sentence would cast unnecessary doubt to our work. However, we have clarified that we have accounted for losses using the particle loss calculator.

The modified text now reads as:

"The standard aerosol inlet is heated to a temperature of around 5--10°C to prevent freezing and fulfils the World Meteorological Organization (WMO)/Global Atmosphere Watch programme guidelines for aerosol sampling of whole-air in extreme environments (WMO/GAW 2016, Wiedensohler et al., 2014). The inlet was built in collaboration with the World Calibration Centre for Aerosol Physics (WCCAP) at the Leibniz Institute for Tropospheric Research, Germany and is technically identical to the inlet described by (Weingartner et al., 1999), which can sample cloud droplets up to 40 µm at wind speeds up to 20 ms$^{-1}$. Particle losses within the inlet lines were accounted for using the *Particle Loss Calculator* by (von der Weiden et al, 2009) (see Sect. 2.4  below). "

Concerning the data submission, we are in the process to receive a DOI from the Bolin Centre Database and will add the exact address during the proof-reading stage.

Thanks and kind regards,

Paul Zieger on behalf of all co-authors.